# Utility Boundary of Dataset Distillation: Scaling and Coverage Laws

**Zhengquan Luo** [1] **Zhiqiang Xu** [1]

## Abstract

Dataset distillation (DD) aims to replace a full training set with a tiny synthetic one, yet current theories neither explain why heterogeneous matching objectives (gradient, distribution, trajectory) work nor provide a quantitative boundary for robustness under configuration changes (optimizer, architecture, augmentation). We propose *configuration-dynamics-error (CDE) analysis* for a broad class of matching-based DD methods, a unified generalization framework that treats a training configuration as an update operator inducing optimization dynamics and measures distillation robustness by the test-risk gap between models trained on distilled versus full data. Within this framework, gradient, distribution, and trajectory matching reduce the same dynamics-induced risk gap, explaining why these heterogeneous objectives can all support dataset distillation. CDE yields two predictive laws: within a fixed configuration, the gap decays as $\mathcal{O}(k^{-1/2})$ with the distilled set size $k$ until a configuration-dependent floor, explaining IPC saturation and indicating when reducing the floor is more valuable than enlarging $k$. Across configurations, an order-tight coverage law formalizes the utility boundary: the required $k$ grows linearly with the configuration diversity captured by covering-number complexity. Experiments with representative DD methods and configuration changes exhibit predictive behavior consistent with both laws.

## 1. Introduction

*Dataset distillation* (Wang et al., 2018; Sucholutsky & Schonlau, 2021), also known as dataset condensation (DC) (Zhao et al., 2020; Wang et al., 2022), seeks to synthesize a compact dataset that enables models to approach the accuracy of full-data training while greatly reducing storage and compute costs. In recent years, three main matching-based categories have emerged. *Gradient matching (GM)* aligns gradients between real and synthetic data through bilevel optimization, extended by augmentation consistency (Zhao & Bilen, 2021), diversity regularization (Cazenavette et al., 2023), and reverse matching (Ye et al., 2024). *Distribution matching (DM)* matches feature statistics, from early MMD-based formulations (Li et al., 2017) to higher-order or quantile-based variants (Wang et al., 2022; Zhang et al., 2024; Wei et al., 2024). *Trajectory matching (TM)* aligns full optimization dynamics, introduced in MTT (Cazenavette et al., 2022) and later extended to self-supervised DD and detection tasks (Lee et al., 2023; Qi et al., 2024) (a comprehensive review of related work is provided in Appendix B).

Despite empirical advances, the theoretical foundation of DD remains fragmented. Existing theories are confined to paradigm-specific assumptions: the theory for GM is largely restricted to first-order gradient matching (Zhao & Bilen, 2021; Deng & Russakovsky, 2022); the theory for DM relies on kernel- or moment-based statistics (Li et al., 2017), thus neglecting optimization dynamics; and the theory for TM, while empirically strong (Cazenavette et al., 2022), lacks rigorous convergence guarantees beyond heuristic approximations. These paradigm-specific limitations highlight the absence of a unified theoretical view to relate different DD approaches. It is still unclear why all three categories can yield near full-data utility and it is difficult to explain recurring empirical patterns such as the saturation of accuracy gains with larger distilled sample sizes (Cazenavette et al., 2022). A further challenge is the robustness to *training configuration shifts*, e.g., changes in optimizer, architecture, and augmentation between distillation and downstream training. Since DD is expected to substitute for the full dataset in practice, distilled data must remain effective under these shifts. Current evaluations often restrict to fixed setups or mild parameter perturbations (Nguyen et al., 2021; Zhao & Bilen, 2023), and reported results reveal instability: sensitivity to random seeds (Wang et al., 2018), reliance on augmentation (Zhao & Bilen, 2021), and weak cross-architecture transfer (Liu et al., 2022).

To this end, we introduce the notion of a *utility boundary*: the relationship between the distilled sample size and the

---

[1]Department of Machine Learning, Mohamed bin Zayed University of Artificial Intelligence, Abu Dhabi, United Arab Emirates. Correspondence to: Zhiqiang Xu <zhiqiang.xu@mbzuai.ac.ae>.

*Proceedings of the 43rd International Conference on Machine Learning*, Seoul, South Korea. PMLR 306, 2026. Copyright 2026 by the author(s).

diversity of configurations within which the distilled dataset can still match the performance of full-data training. We address these challenges by *instantiating* standard tools from optimization and generalization analysis, yielding a unified *configuration-dynamics-error* (CDE) framework. A configuration specifies the update operator (e.g., optimizer, architecture, augmentation) that governs parameter changes, and together with the training distribution induce the optimization dynamics. We analyze the resulting *test-risk gap* between training on distilled versus real data on a common test distribution.

Building on this framework, we first analyze the single-configuration case, yielding the *scaling-to-floor law*: as the distilled sample size $k$ increases, the generalization error decreases until it reaches a configuration-dependent irreducible bound $\epsilon_{\text{bound}}$, following the statistical rate $\Delta \leq \mathcal{O}\left(1/\sqrt{k}\right) + \epsilon_{\text{bound}}$. This explains the commonly observed images-per-class (IPC) saturation: when $k$ is sufficiently large, the error is dominated by the irreducible floor and accuracy no longer improves by increasing $k$; in this regime, reducing $\epsilon_{\text{bound}}$ is more valuable than blindly enlarging $k$.

To account for generalization across configurations, we then extend the analysis from a single configuration to a family of configurations. In this setting, $\mathcal{H}_{\text{cov}}(\mathcal{A}, r)$ measures configuration diversity, yielding the *configuration-coverage law*: $\Delta = \Theta\left(\sqrt{\mathcal{H}_{\text{cov}}(\mathcal{A}, r)/k}\right)$. This provides a formal *utility boundary* for dataset distillation, showing how the required sample size must grow with configuration diversity to maintain generalization.

Within this framework, GM, DM, and TM are not independent heuristics but instances of a common objective: minimizing a *matching discrepancy* that measures how differently real and distilled data drive training dynamics under a configuration. While different DD methods align gradients, trajectories, or feature-level statistics, our analysis and experiments indicate that these are surrogate choices within a shared outer-inner (bi-level) mechanism. Their generalization is therefore governed by the same scaling law and coverage law, providing guidance for designing distilled datasets that achieve both sample efficiency and robustness. To summarize, we make the following contributions:

- *Utility across configurations.* We formalize dataset distillation as minimizing a configuration-dependent risk gap and argue that practical DD should be evaluated by its utility across configurations, not a single training protocol.

- *Scaling law.* Under mild trajectory-stability conditions, we show the distillation error decreases as $\mathcal{O}(1/\sqrt{k})$ before saturating at a configuration-dependent irreducible floor, explaining IPC saturation.

- *Coverage law.* We quantify configuration diversity via a covering complexity and show that maintaining a target error across configurations requires $k$ to scale proportionally with this complexity; we also provide matching lower bounds (order-tight).

- *Unify DD.* We place GM, DM, and TM under the proposed framework and empirically validate the theoretical findings across representative methods, datasets, and configuration shifts.

## 2. Preliminaries and Problem Setup

**Dataset distillation (DD).** Given a real dataset $\mathcal{D}_\tau = \{(x_i, y_i)\}_{i=1}^n$ with empirical distribution $\hat{\mu}_\tau = \frac{1}{n} \sum_{i=1}^n \iota_{(x_i, y_i)}$[1], the goal of DD is to construct a compact synthetic dataset $\mathcal{D}_s = \{(x'_j, y'_j)\}_{j=1}^k$ with empirical distribution $\hat{\mu}_s = \frac{1}{k} \sum_{j=1}^k \iota_{(x'_j, y'_j)}$ for $k \ll n$, such that training on $\hat{\mu}_s$ matches training on $\hat{\mu}_\tau$ in test performance (Wang et al., 2018; Sucholutsky & Schonlau, 2021; Zhao et al., 2020). Let $\nu$ denote a test distribution with empirical counterpart $\hat{\nu} = \frac{1}{m} \sum_{l=1}^m \iota_{(x_l^{\text{te}}, y_l^{\text{te}})}$. When population-level training distributions are needed in the proofs, we write $\mu_\tau$ for the distribution underlying the real data and $\mu_s$ for the ideal or prototype distribution represented by the distilled data; hats always denote empirical measures. We evaluate empirical and population risks:

$$\hat{R}(\theta) = \mathbb{E}_{\hat{\nu}}[\ell(\theta; z)], \qquad R_\nu(\theta) = \mathbb{E}_\nu[\ell(\theta; z)], \quad (1)$$

where $z$ denotes a data point, and compare models trained on $\hat{\mu}_s$ versus $\hat{\mu}_\tau$.

**Single training configuration.** Most theories are established under a single fixed training configuration (e.g., optimizer, hyperparameters, augmentation, architecture) and show that particular matching strategies (gradient, distribution, or trajectory) improve accuracy in the single setting (Zhao et al., 2020; Cazenavette et al., 2022; Zhao & Bilen, 2023). Formally, one training step under a (fixed) configuration can be written as:

$$\theta_{t+1} = \Phi(\theta_t; \mu) = \theta_t - \eta \, P(\theta_t) \, \mathbb{E}_\mu[g(\theta_t; z)], \quad (2)$$

where $P(\theta)$ denotes a (possibly adaptive) preconditioner and $g$ the per-sample update. We refer to the configuration used to generate the distilled dataset as the *source configuration*. The configuration under which we evaluate and compare the utility of real versus distilled data is called the *target configuration*.

---

[1] $\iota$ denotes the Dirac measure that assigns unit mass to the sample point.

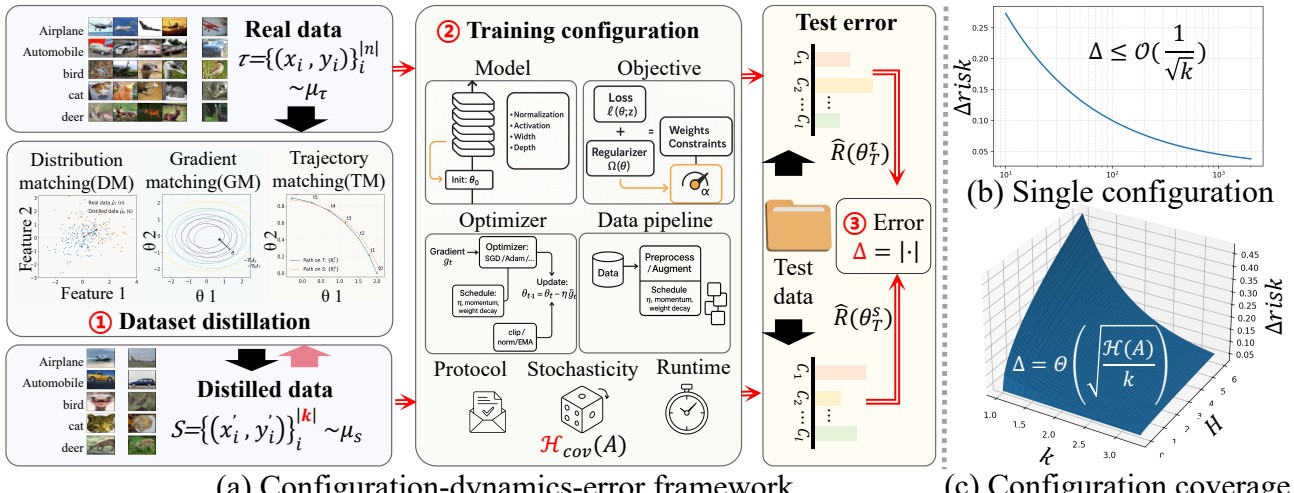

*Figure 1.* Configuration-dynamics-error framework: a configuration (optimizer, architecture, augmentation, etc.) together with the training distribution (either the real dataset or its distilled dataset) induces optimization dynamics, whose risk is evaluated through generalization error bounds; this yields the scaling law for a single configuration and the coverage law across configuration families.

## 3. Unified Configuration-Dynamics-Error

In practice, the ultimate goal of a distilled dataset is to replace the real dataset across diverse applications. Since the target configurations used for deployment may differ from those assumed during distillation, it is necessary to extend the analysis from a single setup to a *family of configurations*. Doing so highlights three key aspects that jointly determine the effectiveness of distilled data: the *diversity* of configurations it must cover, the *alignment* it maintains with real data, and the *stability* with which training dynamics transfer across configurations.

**Space of configurations.** A configuration $a$ specifies optimizer, hyperparameters, augmentation, and architecture. The training under configuration $a$ on distribution $\mu$ induces parameter iterate:

$$\theta_{t+1} = \Phi_a(\theta_t; \mu) = \theta_t - \eta\, P_a(\theta_t)\, \mathbb{E}_\mu\, g_a(\theta_t; z) \in \Gamma_a, \quad (3)$$

with feasible set $\Gamma_a$ for parameters. The way we update parameters here generalizes classical stochastic approximation and adaptive methods (Robbins & Monro, 1951; Bottou et al., 2018). We consider a family of target configurations, $\mathcal{A} \subseteq \mathcal{C}$, that reflects the intended deployment setting (Shalev-Shwartz & Ben-David, 2014).

**Diversity.** Each configuration $a \in \mathcal{A}$ induces its own feasible parameter set $\Gamma_a \subseteq \mathbb{R}^d$. To compare two configurations $a, a' \in \mathcal{A}$ on the same real data $\hat{\mu}_\tau$, we define the configuration-distance

$$d_{\mathcal{A}}(a, a') = \sup_{\theta \in \Gamma_a \cap \Gamma_{a'}} \left\| P_a(\theta)\, \mathbb{E}_{\hat{\mu}_\tau} g_a(\theta; z) \right.$$
$$\left. - P_{a'}(\theta)\, \mathbb{E}_{\hat{\mu}_\tau} g_{a'}(\theta; z) \right\|_2. \quad (4)$$

This metric is inspired by the stability and uniform convergence analyses of algorithmic dynamics (Hardt et al., 2016; Raginsky et al., 2017). The *coverage complexity* of $\mathcal{A}$ at radius $r > 0$ under $d_{\mathcal{A}}$ is $\mathcal{H}_{\text{cov}}(\mathcal{A}, r) = \log N(\mathcal{A}, d_{\mathcal{A}}, r)$, where $N(\mathcal{A}, d_{\mathcal{A}}, r)$ is the minimal number of $d_{\mathcal{A}}$-balls of radius $r$ needed to cover $\mathcal{A}$. Here $r$ is mathematically the covering *radius*, which determines the *resolution* of configuration distinctions: smaller $r$ resolves finer differences and thus increases $\mathcal{H}_{\text{cov}}(\mathcal{A}, r)$.

**Alignment.** For measures $\mu, \nu$ and configuration $a$, we define the *matching discrepancy*

$$\Delta_a(\mu, \nu) := \sup_{\theta \in \Gamma_a} \left\| P_a(\theta) \big( \mathbb{E}_\mu g_a(\theta; z) - \mathbb{E}_\nu g_a(\theta; z) \big) \right\|_2. \quad (5)$$

This notion unifies classical discrepancy measures used in dataset distillation and domain adaptation (Zhao et al., 2020; Cazenavette et al., 2022; Zhao & Bilen, 2023; Ben-David et al., 2006). Specializing to empirical real and empirical synthetic distributions gives $\Delta_a(\hat{\mu}_\tau, \hat{\mu}_s)$.

**Stability.** The final stage of our configuration–dynamics–error framework asks how discrepancies in training dynamics translate into generalization error. Throughout the paper, we build on stability- and information-theoretic perspectives on generalization (Bousquet & Elisseeff, 2002; Russo & Zou, 2016; Xu & Raginsky, 2017), and use a *decomposition view* to separate the contributions of (i) optimization progress (controlled by the number of steps), (ii) statistical fluctuations from finite samples, and (iii) the configuration-dependent alignment mismatch between synthetic and real data. We defer the formal statements and the resulting bound decomposition to Sections 4-5.

At this point we have a complete *configuration dynamics error* framework as Figure 1. Configurations specify the update operators that drive parameter changes; their diversity is captured through covering complexity under a configuration distance (*Diversity*); the gap between synthetic and real data within each configuration is measured by the matching discrepancy (*Alignment*); and the transfer from dynamics to generalization error is governed by generalization error bounds decomposition (*Stability*). Together, these elements form a coherent CDE analysing chain.

# 4. Single-Configuration Generalization Bound

We instantiate the configuration-dynamics-error framework with a fixed configuration $a$ and derive a finite–sample bound that reveals the scaling law of dataset distillation.

**Notations** The quantity $\Delta_a$ measures the update mismatch between real and distilled data under configuration $a$; $C_{2,a}^{\text{traj}}$ measures how this local mismatch accumulates along the realized training trajectory; $e_g$ and $e_{\text{te}}$ are finite-sample fluctuations from distillation/training data and test data, respectively; and $\epsilon_{\text{bound}}$ denotes the residual floor after the $k$-dependent terms decay. A full notation table is given in Appendix A.

**Assumption 4.1** (Regularity). On the feasible parameter domain $\Gamma_a$, we assume: (i) bounded per–sample update and preconditioner, i.e., $\|g_a(\theta; z)\| \leq B_g$, $\|P_a(\theta)\| \leq \kappa_a$; (ii) bounded loss $|\ell(\theta; z)| \leq B_\ell$ and $L_R$–Lipschitz test risk $\hat{R}$; (iii) trajectory-local stable dynamics: along the training trajectories induced by real and synthetic data, there exist $\{\rho_{a,t}\}_{t=0}^{T-1}$ with $\rho_{a,t} \in (0,1)$ such that

$$\|\theta_{t+1}^{(s)} - \theta_{t+1}^{(\tau)}\| \leq \rho_{a,t}\|\theta_t^{(s)} - \theta_t^{(\tau)}\| + \eta\,\Delta_a(\mu_\tau, \mu_s),$$

for all $t = 0, \ldots, T-1$. We define the *trajectory effective constant* $C_{2,a}^{\text{traj}} := \sum_{t=0}^{T-1} \prod_{s=t+1}^{T-1} \rho_{a,s}$, which is finite whenever $\rho_{a,t} < 1$ along the trajectory.

Assumption 4.1(iii) is weaker than a global PL condition: it only requires local stability along the realized training trajectory under configuration $a$. In practice, $\prod_t \rho_{a,t}$ and $C_{2,a}^{\text{traj}}$ can be estimated by measuring the sensitivity of the update map to small perturbations around $\{\theta_t\}_{t=0}^T$.

**Definition 4.2** (Intrinsic generalization error). For configuration $a$, let $\mathcal{P}_k$ denote the class of distributions supported on at most $k$ prototypes. This class induces an irreducible generalization error $\Delta_a^\star := \inf_{\mu \in \mathcal{P}_k} \Delta_a(\mu_\tau, \mu)$. It measures the best possible population-level matching between $k$ prototypes and the real training distribution under configuration $a$; replacing $\mu_\tau$ by $\hat{\mu}_\tau$ gives the empirical analogue used in finite-sample statements and experiments.

**Theorem 4.3** (Single–configuration risk bound). *Let $\theta_T^{(s)}$ and $\theta_T^{(\tau)}$ denote the parameters after $T$ steps trained on*

synthetic and real data, respectively, with initialization gap $\delta_0 = \theta_0^{(s)} - \theta_0^{(\tau)}$. *Then with probability at least $1 - \varepsilon$,*

$$|R_\nu(\theta_T^{(s)}) - R_\nu(\theta_T^{(\tau)})| \leq L_R\Big(\prod_{t=0}^{T-1} \rho_{a,t}\Big)\|\delta_0\| \tag{6}$$
$$+\eta L_R\, C_{2,a}^{\text{traj}}(\Delta_a^\star + \kappa_a e_g) + e_{\text{te}},$$

*where $e_g = \mathcal{O}(1/\sqrt{k} + 1/\sqrt{n})$ is the fluctuation from distillation and training samples, and $e_{\text{te}} = \mathcal{O}(1/\sqrt{m})$ is the test concentration error. If the distilled dataset $\mathcal{D}_s$ is generated from the real dataset $\mathcal{D}_\tau$, then $e_g$ further incurs an information–theoretic penalty $\mathcal{O}(\sqrt{I(\mathcal{D}_s; \mathcal{D}_\tau)/k})$ (Russo & Zou, 2016) (see Appendix C for proof details).*

*Remark* 4.4. When $T, n, m$ are sufficiently large, optimization and statistical terms vanish. We then arrive at the single–configuration *scaling law*:

$$\left|R_\nu(\theta_T^{(s)}) - R_\nu(\theta_T^{(\tau)})\right| \approx \eta L_R\, C_{2,a}^{\text{traj}}\, \Delta_a^\star + \mathcal{O}(1/\sqrt{k}). \tag{7}$$

**Corollary 4.5.** *For a target error $\epsilon_0 > \eta L_R C_{2,a}^{\text{traj}}\Delta_a^\star$, distilled sample size $k$ must satisfy $k = \Omega\Big((\epsilon_0 - \eta L_R C_{2,a}^{\text{traj}}\Delta_a^\star)^{-2}\Big)$.*

*Remark* 4.6. i) The generalization error could decrease with $k$ until saturation at the irreducible error bound $\eta L_R C_{2,a}^{\text{traj}}\Delta_a^\star$, which accounts for the commonly observed IPC saturation; ii) With finite training and test samples $n, m$, their sizes may limit accuracy, but the distilled sample size $k$ remains the fundamental bottleneck of distillation since $k \ll n, m$.

# 5. Coverage-Aware Bounds

The preceding remark establishes the local scaling behavior under a fixed configuration, highlighting the role of $k$ as the fundamental bottleneck. In practice, however, distilled data are expected to remain effective across not just one but a family of configurations $\mathcal{A} \subseteq \mathcal{C}$ (optimizers, architectures, augmentations). In what follows, we analyze how the generalization error scales with the configuration diversity of $\mathcal{A}$, derive the corresponding upper and lower bounds, and ultimately arrive at a tight *coverage law*.

With concepts introduced in Sections 2–4, ranging from configuration–distance $d_\mathcal{A}$, coverage diversity $\mathcal{H}_{\text{cov}}(\mathcal{A}, r) = \log N(\mathcal{A}, d_\mathcal{A}, r)$, matching discrepancy $\Delta_a$, irreducible generalization error $\Delta_a^\star$, to trajectory dynamics constants $\{\rho_{a,t}\}_{t=0}^{T-1}$ and $C_{2,a}^{\text{traj}}$, we further introduce Rademacher constants $C_G^+, \tilde{C}_G^{+2}$, and extend the Lipschitz assumption:

---

[2]$C_G^+$ denotes the supremum Rademacher complexity constant across configurations. When finite-sample or information-theoretic corrections are present, we denote the corrected version by $\tilde{C}_G^+$; both share the same order.

**Assumption 5.1** (Lipschitz transfer across configurations and parameters). There exist $L_{\text{conf}}, L_\theta > 0$ such that for all $a, a' \in \mathcal{C}$, all $\theta, \theta' \in \Gamma_a \cap \Gamma_{a'}$, and $\mu \in \{\hat{\mu}_\tau, \hat{\mu}_s\}$,

$$\left\| \begin{aligned} &P_a(\theta)\mathbb{E}_\mu g_a(\theta; z) \\ &- P_{a'}(\theta)\mathbb{E}_\mu g_{a'}(\theta; z) \end{aligned} \right\|_2 \leq L_{\text{conf}} d_{\mathcal{A}}(a, a'),$$

$$\left\| \begin{aligned} &P_a(\theta)\mathbb{E}_\mu g_a(\theta; z) \\ &- P_a(\theta')\mathbb{E}_\mu g_a(\theta'; z) \end{aligned} \right\|_2 \leq L_\theta \|\theta - \theta'\|_2,$$

which requires smooth variation across configurations and parameter–Lipschitz continuity along the relevant trajectories; this is compatible with standard bounded-learning-rate implementations of optimizers such as SGD and Adam when the iterates remain in the studied configuration family.

Based on the mild extension of $d_{\mathcal{A}}$, which only requires the uniform Lipschitz continuity over $\{\hat{\mu}_\tau, \hat{\mu}_s\}$ and all $\theta \in \Gamma$, the $d_{\mathcal{A}}$–based coverage argument extends consistently from cover centers to configuration family $\mathcal{A}$, and the configuration-dynamic-risk framework can extend from single-configuration trajectory-stable dynamics to all configurations.

**Theorem 5.2** (Uniform cross–configuration bound). *For any $\varepsilon \in (0, 1)$, with probability at least $1 - \varepsilon$ over the draws of $\hat{\mu}_\tau, \hat{\mu}_s, \hat{\nu}$, it holds for any configuration prior $\Pi$ supported on $\mathcal{A}$ that*

$$\mathbb{E}_{a \sim \Pi} \left| R_\nu(\theta_T^{(s,a)}) - R_\nu(\theta_T^{(\tau,a)}) \right| \leq \epsilon_{\text{bound}}^{\text{upper}} + A_1 \frac{\mathcal{H}_{\text{cov}}(\mathcal{A}, r)}{k} + A_2 \sqrt{\frac{\mathcal{H}_{\text{cov}}(\mathcal{A}, r)}{k}},$$

$$\sup_{a \in \mathcal{A}} \left| R_\nu(\theta_T^{(s,a)}) - R_\nu(\theta_T^{(\tau,a)}) \right| \leq \epsilon_{\text{bound}}^{\text{upper}} + \frac{C_{\text{cov}}(\mathcal{A})}{\sqrt{k}},$$

$$\epsilon_{\text{bound}}^{\text{upper}} = \mathcal{O}\left( \rho_{\max}^T \|\delta_0\| + \sup_{a \in \mathcal{A}} \Delta_a^\star + \frac{1}{\sqrt{n}} + \frac{\sqrt{\mathcal{H}_{\text{cov}}(\mathcal{A}, r)}}{\sqrt{m}} \right),$$

$$C_{\text{cov}}(\mathcal{A}) = \mathcal{O}\left( \sqrt{\mathcal{H}_{\text{cov}}(\mathcal{A}, r)} \right),$$

*where $\rho_{\max}^T := \sup_{a \in \mathcal{A}} \prod_{t=0}^{T-1} \rho_{a,t}$ denotes the worst-case trajectory contraction product over $\mathcal{A}$. Throughout, the $\mathcal{O}(\cdot)$ notation hides constants that may depend on $\sup_{a \in \mathcal{A}} \kappa_a$ and $\sup_{a \in \mathcal{A}} C_{2,a}^{\text{traj}}$ (as well as other fixed regularity and Lipschitz constants), but are independent of $k, n, m$. If $\mathcal{D}_s$ depends on $\mathcal{D}_\tau$, an additional information-theoretic correction $\mathcal{O}\left( \sqrt{I(\mathcal{D}_s; \mathcal{D}_\tau)/k} \right)$ is added to both bounds (see Appendix D.1 and D.2 for proof details).*

The above upper bound indicates that the required distilled size increases linearly with *the configuration diversity*

$\mathcal{H}_{\text{cov}}(\mathcal{A})$. A natural further question then is: *is this dependence optimal?* The next theorem answers it in the affirmative by giving a matching lower bound and thus justifying the optimal dependence of the distilled sample size on the configuration diversity.

**Theorem 5.3** (Coverage lower bound). *Suppose Assumption 4.1 holds (single–configuration regularity). Assume further an* identifiability condition*: there exists $\lambda > 0$ such that, for all $\theta \in \Gamma$ and any two distinct configurations $a, a' \in \mathcal{A}$, $\left\| P_a(\theta) \mathbb{E}_{\hat{\mu}_\tau} g_a(\theta; z) - P_{a'}(\theta) \mathbb{E}_{\hat{\mu}_\tau} g_{a'}(\theta; z) \right\|_2 \geq \lambda d_{\mathcal{A}}(a, a')$. That is, update dynamics corresponding to different configurations are uniformly separated. Assume also the standard testing regularity used in Appendix D.4: (i) risk gaps are locally observable in the alignment metric, and (ii) a $k$-prototype distilled representation has Fano capacity at most order $k(\epsilon/(\lambda \varrho))^2$ for resolving a $\varrho$-packing. Let $\mathcal{A}$ admit a $\varrho$–packing with $M$ elements (so that the packing entropy $\log M$ is comparable to $\mathcal{H}_{\text{cov}}(\mathcal{A}, \varrho)$ up to standard packing-covering constants). Then, for any distillation algorithm producing $k$ synthetic samples, there exists a distribution over this packed family such that*

$$\mathbb{E}_a \left| R_\nu(\theta_T^{(s,a)}) - R_\nu(\theta_T^{(\tau,a)}) \right|$$
$$\geq \epsilon_{\text{bound}}^{\text{lower}} + c_{\text{lb}} \, \varrho \lambda \sqrt{\frac{\mathcal{H}_{\text{cov}}(\mathcal{A}, \varrho)}{k}},$$

*where $c_{\text{lb}} \in (0, 1)$ is a universal constant (see Appendix D.4 for proof details).*

**Corollary 5.4** (Coverage law). *For any generalization error $\epsilon_0 > \epsilon_{\text{bound}}$, if the distilled sample size satisfies $k \geq K_{\min}(\epsilon_0, \mathcal{A}) = \left( \frac{C_{\text{cov}}(\mathcal{A})}{\epsilon_0 - \epsilon_{\text{bound}}} \right)^2 = \Theta(\mathcal{H}_{\text{cov}}(\mathcal{A}, r))$, then it holds that*

$$\sup_{a \in \mathcal{A}} \left| R_\nu(\theta_T^{(s,a)}) - R_\nu(\theta_T^{(\tau,a)}) \right| \leq \epsilon_0.$$

*Remark* 5.5 (Coverage law). (i) The upper bound separates two errors: an approximation error $\mathcal{H}_{\text{cov}}/k$, since a size-$k$ set cannot fully cover $\mathcal{A}$, and a concentration error $\sqrt{\mathcal{H}_{\text{cov}}/k}$ from uniform guarantees. In typical regimes, the latter dominates, as it decays more slowly and thus sets the critical rate. (ii) The residual $\epsilon_{\text{bound}}$ collects optimization error, irreducible matching discrepancy, and sampling noise, yielding a non-vanishing floor. (iii) $\mathbb{E}_{a \sim \Pi}$ gives an *average-case* guarantee under a prior $\Pi$, while $\sup_{a \in \mathcal{A}}$ strengthens it to a *worst-case* guarantee across all configurations, reducing to the $\sqrt{\mathcal{H}_{\text{cov}}/k}$ rate. (iv) The lower bound shows that any target error $\epsilon_0 > \epsilon_{\text{bound}}$ requires $k = \Omega(\mathcal{H}_{\text{cov}}(\mathcal{A}, r))$. This matches the upper bound $k = \mathcal{O}(\mathcal{H}_{\text{cov}}(\mathcal{A}, r))$ up to constants. Taken together, these results establish the tightness of the coverage law: as $\mathcal{H}_{\text{cov}}$ grows, $k$ must scale proportionally to maintain accuracy, and no distillation algorithm can avoid the $\sqrt{\mathcal{H}_{\text{cov}}/k}$ barrier.

**Practical estimation of coverage entropy.** Our coverage law depends on $\mathcal{H}_{\mathrm{cov}}(\mathcal{A}, r) = \log N_r$ over $(\mathcal{A}, d_{\mathcal{A}})$, where $N_r$ is the minimal $r$-covering number. Since exact covering is NP-hard, we estimate $\mathcal{H}_{\mathrm{cov}}$ from *one-step training dynamics*. We fix a few random initializations $\{\theta_0^{(s)}\}_{s=1}^S$ and mini-batches $\{\mathcal{B}_b\}_{b=1}^B$. For each configuration $a$, define the normalized (preconditioned) one-step update

$$u(a; s, b) = \frac{P_a(\theta_0^{(s)})\, g_a(\theta_0^{(s)}; \mathcal{B}_b)}{\|P_a(\theta_0^{(s)})\, g_a(\theta_0^{(s)}; \mathcal{B}_b)\|_2 + \varepsilon},$$

where $\varepsilon > 0$ is a small numerical constant to avoid division by zero, and the configuration distance $d_{\mathcal{A}}(a, a') = \frac{1}{SB} \sum_{s=1}^S \sum_{b=1}^B \|u(a; s, b) - u(a'; s, b)\|_2$. Given sampled candidates $\mathcal{A}_M \subset \mathcal{A}$, we apply a farthest-first greedy $r$–cover under $d_{\mathcal{A}}$ to obtain $\widehat{N}_r$ and set

$$\widehat{\mathcal{H}_{\mathrm{cov}}}(\mathcal{A}, r) = \log \widehat{N}_r. \tag{8}$$

We choose $r$ by noise calibration: $r = c \cdot \mathrm{median}_{a \in \mathcal{A}_M} \sigma_a$ with $c \in [1, 2]$, where $\mathrm{median}(\cdot)$ denotes the sample median, $\sigma_a = \mathrm{median}_{s,b} \|u(a; s, b) - \bar{u}(a)\|_2$, and $\bar{u}(a) = \frac{1}{SB} \sum_{s,b} u(a; s, b)$ (Algorithmic details are provided in Appendix F.2).

*Remark* 5.6. New insights into dataset distillation emerge from the coverage law and its practical estimation: (i) the number of distilled samples $k$ must grow in proportion to the coverage complexity $\mathcal{H}_{\mathrm{cov}}(\mathcal{A}, r)$ in order to maintain a fixed generalization error; and (ii) estimating $\mathcal{H}_{\mathrm{cov}}(\mathcal{A}, r)$, or using a proxy such as $\log M$, provides a principled way to determine how many distilled samples are needed and which source configurations should be prioritized so that the synthetic dataset preserves the utility of the real dataset across a target configuration family. These insights connect the theoretical limits with practical guidelines for designing robust dataset distillation methods.

# 6. Unifying Categories of Dataset Distillation

From Sections 4-5, the matching discrepancy $\Delta_a(\hat{\mu}_\tau, \hat{\mu}_s)$ turns out to be the key to the generalization errors. We are now in a position to scrutinize why the three major distillation methods, i.e., DM, GM, and TM, albeit in different forms, all reduce the same $\Delta_a$ via the bi-level optimization mechanism.[3]

---

[3]We use "bi-level" here in the operational sense of separating synthetic-data optimization from model training on the synthetic data: $\xi$ parameterizes the synthetic data, while $\theta_t$ denotes model parameters trained on them. Under this convention, DM also fits the template: the distilled data are optimized by a distribution-matching objective, while the downstream model is trained on the distilled data in the inner loop. Standard DM, however, does not use the resulting inner-loop model parameters, gradients, or trajectories to guide the synthetic-data update; GM and TM do so explicitly through gradient or trajectory information.

**Unified bi-level optimization.** Let the distilled dataset be parameterized by $\xi$ with distribution $\mu(\xi)$, and $\Theta_j$ the inner states queried at outer iteration $j$. Each method minimizes a surrogate $\mathcal{M}_\phi(\mu(\xi); \hat{\mu}_\tau, b, \Theta_j)$ with $\phi \in \{\mathrm{DM}, \mathrm{GM}, \mathrm{TM}\}$, where $b$ denotes the *source configuration* under which distillation is performed (to distinguish it from the target configuration $a$ used for evaluation):

Inner loop (training under $b$):

$$\theta_{t+1} = \theta_t - \eta\, P_b(\theta_t)\, \mathbb{E}_{z \sim \mu(\xi)} g_b(\theta_t; z),$$

Outer loop (updating $\xi$):

$$\xi^{(j+1)} = \xi^{(j)} - \eta_j\, \nabla_\xi\, \mathcal{M}_\phi(\mu(\xi^{(j)}); \hat{\mu}_\tau, b, \Theta_j).$$

The concrete choice of $\mathcal{M}_\phi$ differs across branches (distribution-, gradient-, or trajectory-based), but its role is uniform: it offers a tractable objective whose decrease can be translated into a decrease of the alignment discrepancy $\Delta_a$. Moreover, under standard smoothness and estimator assumptions, the outer solver drives the surrogate toward approximate stationarity, up to an estimator-induced floor (formalized in Appendix E.2). To make this connection explicit, we next establish a surrogate-to-alignment bridge that upper-bounds $\Delta_a$ by each branch surrogate and shows these controls are comparable up to constants, thereby formalizing the exchangeability of DM/GM/TM.

**Lemma 6.1** (Exchangeability of surrogates). *Fix configuration $a = b$. Under the smoothness, Lipschitz, and contraction conditions in Assumption 4.1, the matching discrepancy admits the bounds*

$$\Delta_a(\hat{\mu}_\tau, \hat{\mu}_s) \leq \underbrace{\kappa_a L_{z,a}\, W_1 \text{ or } \kappa_a C_k\, \mathrm{MMD}_k}_{\mathfrak{B}_{\mathrm{DM}}},$$

$$\Delta_a(\hat{\mu}_\tau, \hat{\mu}_s) \leq \underbrace{\kappa_a |\Theta_j|\, \mathcal{M}_{\mathrm{GM}}}_{\mathfrak{B}_{\mathrm{GM}}}, \tag{9}$$

$$\Delta_a(\hat{\mu}_\tau, \hat{\mu}_s) \leq \underbrace{\frac{L_\theta + 2/\eta}{\omega_{\min}} \mathcal{M}_{\mathrm{TM}} + 2 L_\theta \varepsilon_{\mathrm{path}}}_{\mathfrak{B}_{\mathrm{TM}}},$$

*where $\mathfrak{B}_{\mathrm{DM}}$, $\mathfrak{B}_{\mathrm{GM}}$, and $\mathfrak{B}_{\mathrm{TM}}$ are the distribution-, gradient-, and trajectory-based surrogate bounds on $\Delta_a$. Moreover, these bounds are equivalent up to constant factors, i.e.,*

$$\mathfrak{B}_{\mathrm{TM}} = \mathcal{O}(\mathfrak{B}_{\mathrm{GM}}), \quad \mathfrak{B}_{\mathrm{GM}} = \mathcal{O}(\mathfrak{B}_{\mathrm{DM}}).$$

With Lemma 6.1 in place, we can plug the surrogate-to-alignment controls into the CDE analysis, and obtain a unified single-configuration and configuration coverage generalization bound that applies to all DD branches through the choice of $\mathcal{M}_\phi$.

*Table 1.* Unified practical comparison and surrogate-to-alignment bridge. Left: how each branch is optimized; Middle: how its surrogate controls $\Delta_a$; Right: what drives the outer rate.

|  | Outer obj. | Inner $\Theta_j$ | Robust | Compute | Bridge to $\Delta_a$ |
|---|---|---|---|---|---|
| DM | $W_1$ MMD | none | **High** | Low | $\mathfrak{B}_{\mathrm{DM}}^{W_1} = \kappa_a L_{z,a} W_1$ $\mathfrak{B}_{\mathrm{DM}}^{MMD} = \kappa_a C_k \mathrm{MMD}_k$ |
| GM | Grad gap | 1-few $\theta$ | Mid | Mid | $\mathfrak{B}_{\mathrm{GM}} = \kappa_a |\Theta_j| \mathcal{M}_{\mathrm{GM}}$ |
| TM | Path gap | unroll $L_b$ | Low | **High** | $\mathfrak{B}_{\mathrm{TM}} \simeq \frac{L_\theta + 2/\eta}{\omega_{\min}} \mathcal{M}_{\mathrm{TM}}$ |

**Theorem 6.2** (Dynamic single–configuration bound for unifying DD methods)**.** *Fix configuration $a = b$ and run $J$ outer steps with surrogate $\mathcal{M}_\phi$. Under Assumption 4.1, with probability at least $1 - \varepsilon$,*

$$\left| R_\nu(\theta_T^{(s,a)}) - R_\nu(\theta_T^{(\tau,a)}) \right| \leq \epsilon_{\mathrm{bound}}^{\mathrm{distillation}} + \epsilon_{\mathrm{method}}^{(\phi)} + \epsilon_k^{(\phi)},$$

*where $\epsilon_{\mathrm{bound}}^{\mathrm{distillation}} = L_R \left( \prod_{t=0}^{T-1} \rho_{a,t} \right) \|\delta_0\| + \mathcal{O}(1/\sqrt{m})$,*
*$\epsilon_{\mathrm{method}}^{(\phi)} = \mathcal{O}\left( C_{\phi,a} L_R C_{2,a}^{\mathrm{traj}} \left[ \mathcal{M}_\phi(\xi^{(J)}) + \epsilon_{\mathrm{outer}}^{(\phi)}(J,\eta) \right] \right)$,*
*and $\epsilon_k^{(\phi)} = \tilde{\mathcal{O}}(1/\sqrt{k})$. The detailed proof are deferred to Appendix E.5.*

**Theorem 6.3** (Coverage–aware bound with dynamic outer progress)**.** *Let $\mathcal{A}$ denote the family of target configurations, with prior $\Pi$ supported on a $\varrho$–packing, and $a, b \in \mathcal{A}$ (here $\varrho$ is a packing radius, distinct from the contraction factors $\rho_{a,t}$). Under Assumption 5.1, with probability at least $1 - \varepsilon$,*

$$\sup_{a \in \mathcal{A}} |R_\nu(\theta_T^{(s,a)}) - R_\nu(\theta_T^{(\tau,a)})|$$
$$\leq \epsilon_{\mathrm{bound}}^{\mathrm{distillation}} + \epsilon_{\mathrm{method}}^{(\phi)} + \epsilon_{k,\mathrm{cov}}^{(\phi)}.$$

*Compared to Theorem 6.2, there are only two changes. First, the method term is controlled by the worst–case constant $C_{\phi,\max} = \sup_{a \in \mathcal{A}} C_{\phi,a}$ instead of the per–configuration factor $C_{\phi,a}$. Second, the sampling term has an explicit dependence on the coverage complexity, $\epsilon_{k,\mathrm{cov}}^{(\phi)} = \tilde{\mathcal{O}}(\sqrt{\mathcal{H}_{\mathrm{cov}}(\mathcal{A},r)/k})$, in place of the $\tilde{\mathcal{O}}(1/\sqrt{k})$ rate for a single configuration (see Appendix E.6 for proof details).*

*Remark* 6.4 (Interpretation and practical guidance). Theorems 6.2 and 6.3 decompose the generalization gap into (i) an irreducible floor, (ii) a surrogate-/method-dependent term (controlled by $\mathcal{M}_\phi(\xi^{(J)})$ and $\epsilon_{\mathrm{outer}}^{(\phi)}$), and (iii) a sampling term that decreases with $k$; moving from a single configuration to a configuration family affects only the sampling term by introducing the coverage factor, i.e., $\tilde{\mathcal{O}}(1/\sqrt{k}) \to \tilde{\mathcal{O}}(\sqrt{\mathcal{H}_{\mathrm{cov}}(\mathcal{A},r)/k})$.

Crucially, despite their technical differences, GM, DM, and TM are therefore governed by the same scaling law in single configurations and the same coverage law across configuration families, yielding a unified set of design principles. Practically, this suggests improving performance along

two orthogonal axes: increase $k$ (and, under configuration shift, scale $k$ proportionally to $\mathcal{H}_{\mathrm{cov}}(\mathcal{A}, r)$ or reduce $\mathcal{H}_{\mathrm{cov}}$ by restricting the target configuration space), and choose the branch $\phi$ to optimize the method-dependent constant in $\epsilon_{\mathrm{method}}^{(\phi)}$ (Table 1) under compute constraints: DM is typically cheaper and more robust, TM can yield tighter alignment at higher compute, and GM often offers a middle ground.

# 7. Experiments

**Dataset and baselines.** We verify our theoretical results on MNIST (LeCun et al., 2002), CIFAR-10/100 (Krizhevsky et al., 2009), ImageNette (Deng et al., 2009), and additionally report results on Tiny-ImageNet (Le & Yang, 2015) and ImageNet-1K (Deng et al., 2009) to test robustness at larger scales. We evaluate a broad set of distillation paradigms, including four canonical families: *Gradient Matching* (DC (Zhao et al., 2020), DSA (Zhao & Bilen, 2021)), *Distribution Matching* (DM (Zhao & Bilen, 2023)), *Trajectory Matching* (MTT (Cazenavette et al., 2022)), and a recent *diffusion-based* method (MGD[3] (Chan-Santiago et al., 2025)); we further include representative strong baselines from recent literature (TESLA (Cui et al., 2023), NCFM (Wang et al., 2025), DATM (Guo et al., 2023), and SRe2L (Jiang et al., 2025)) when their reported IPC settings are available (details in Appendix F.1).

**Experimental setup.** For DC/DSA/DM/MTT and MGD[3], the distilled dataset is generated in a fixed source configuration (ConvNet+SGD) using the official open-source implementations released by the corresponding papers to ensure reproducibility; for TESLA/NCFM/DATM/SRe2L, we directly extract the IPC results reported in the original papers, which typically provide only a sparse set of IPC values. We vary the number of distilled samples $k \in \{2, 4, 6, 8, 12, 18, 28, 51, 100, 200\}$ whenever the IPC grid is available, and report the generalization error $\Delta_a(\hat{\mu}_\tau, \hat{\mu}_s) = \left| \hat{R}(\theta_T^{(s,a)}) - \hat{R}(\theta_T^{(\tau,a)}) \right|$, where $a$ indexes the target configuration. Each target configuration $a$ is instantiated within a ConvNet-family design space that factors architecture into (i) depth $D \in \{1, 2, 3, 4\}$, (ii) width $W \in \{32, 64, 128, 256\}$, (iii) activation $\in \{\texttt{sigmoid}, \texttt{relu}, \texttt{leakyrelu}, \texttt{swish}\}$, and (iv) nor-

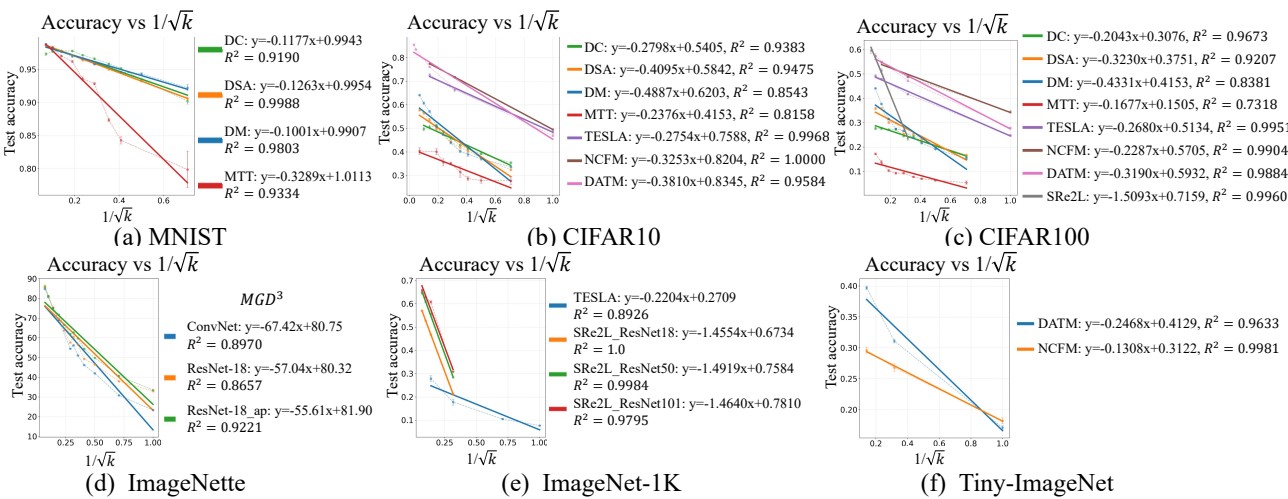

*Figure 2.* Single-configuration scaling law. On MNIST, CIFAR-10/100, ImageNette, ImageNet-1K, and Tiny-ImageNet. The curves of generalization error $\Delta$ against $1/\sqrt{k}$ for GM, DM, and TM shows linear decay at small $k$ followed by saturation at a positive generalization error bound. Regression intercepts give $\epsilon_{\text{bound}}(a)$, consistent with Theorems 4.3 and 6.2.

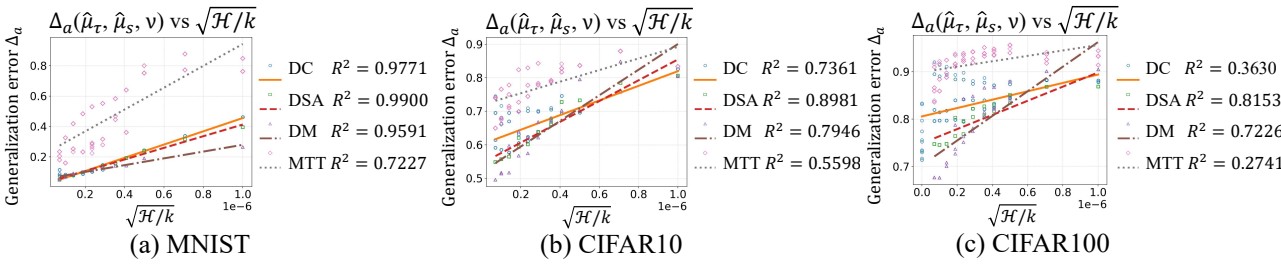

*Figure 3.* Configuration coverage law. For subsets of $m$ configurations, plotting $Y = \Delta(k, M)$ against $X = \sqrt{\log \widehat{N}_r}/\sqrt{k}$ yields near-linear trends under random sampling. Results remain consistent with Theorems 5.2 and 6.3.

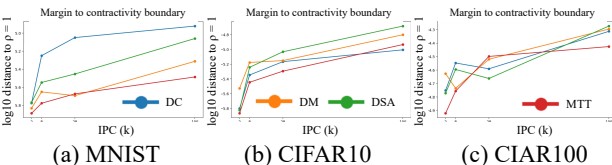

*Figure 4.* Local PL margin along the optimization trajectory.

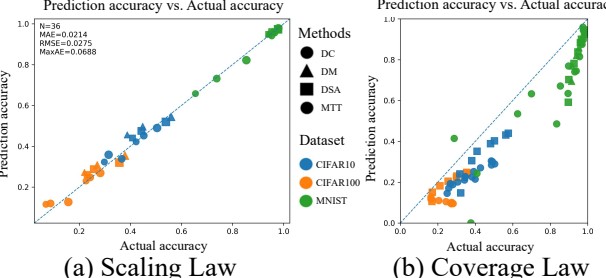

*Figure 5.* Predictive validation of the scaling and coverage laws.

malization $\in \{layernorm, instancenorm, groupnorm\}$, together with the optimizer (SGD/Adam) and augmentation switch (DSA on/off). In practice, we estimate the coverage complexity $\mathcal{H}_{\text{cov}}(\mathcal{A}, r)$ using the update-induced metric and greedy $r$-cover procedure defined in Eq.( 8); pseudo-code is given in Algorithm 2, Appendix F.2.

**Single-configuration regime.** We test the $1/\sqrt{k}$ scaling law (Theorem 6.2) under a fixed configuration $a$ (ConvNet+SGD). Figure 2 shows that across datasets and distillation families, $\Delta_a$ decreases monotonically with $1/\sqrt{k}$ and is typically well captured by a single-line model ($R^2 \gtrsim 0.85$), corroborating the predicted scaling. Specifically, on *MNIST*,

matching-based baselines (DC/DSA/DM) are essentially linear ($R^2$ near 1), whereas MTT is steeper but less stable, consistent with variance from trajectory unrolling. Moving to *CIFAR-10/100*, the scaling trend remains strong, but dataset complexity amplifies behavioral differences: DC/DSA preserve high linearity (up to $R^2 = 0.97$), while DM and especially MTT deviate more on CIFAR-100 ($R^2 \approx 0.84$ and $0.73$), reflecting faster diminishing re-

turns and stronger sensitivity to trajectory mismatch. For TESLA/NCFM/DATM/SRe2L methods, we directly use the IPC results reported in the corresponding papers (typically only a few IPC values); despite this sparse grid, the extracted points remain strictly monotone, indicating that the $1/\sqrt{k}$ law is not tied to a specific matching mechanism. This robustness further carries to *ImageNette* across backbones (MGD³ with ConvNet/ResNet variants, $R^2 \approx 0.87$–$0.92$) and to larger-scale settings (*Tiny-ImageNet* and *ImageNet-1K*). Detailed regression statistics and additional analyses are provided in Appendix F.3.

**Cross-configuration coverage.** We test the coverage law $\Delta \propto \sqrt{\mathcal{H}_{\text{cov}}(\mathcal{A})/k}$ (Theorems 5.2–5.3). Figure 3 shows that across *MNIST/CIFAR-10/CIFAR-100*, $\Delta$ increases monotonically with the proxy $\sqrt{\mathcal{H}_{\text{cov}}(\mathcal{A})/k}$ and is well approximated by a linear relation, providing direct empirical support for the coverage law. Quantitatively, the linear summary is tight on MNIST ($R^2$ up to 0.99), remains informative on CIFAR-10 ($R^2$ up to 0.90), and degrades on CIFAR-100 (down to $R^2 \approx 0.3$ for some methods), reflecting increased variance from class diversity rather than a change in scaling. Across datasets, DC/DSA consistently achieve higher explanatory power under coverage shifts (e.g., $R^2 \gtrsim 0.7$ on CIFAR-10/100), whereas DM and especially MTT exhibit weaker fits on the harder datasets, indicating greater sensitivity to configuration mismatch. Overall, cross-configuration coverage makes explicit the scale-robustness tradeoff and provides a principled guideline for selecting the distilled dataset size when robustness across configurations is required. We further validate the operational value of this viewpoint through coverage-guided source-selection reruns in Appendix F.7. Additional analyses, robustness to alternative coverage proxies, and further cross-backbone evaluations (e.g. ConvNet $\rightarrow$ ResNet) are provided in Appendix F.4, F.5.

**Predictive validation of laws.** We evaluate the two laws via out-of-sample prediction on three datasets (MNIST/CIFAR-10/CIFAR-100) and four distillation methods (DC/DM/DSA/MTT). For the *scaling law*, we fit on a subset of observed budgets $k$ and predict test accuracy at held-out (unseen) $k$; for the *coverage law*, we use observed training configurations to predict accuracy under unseen configurations and compare against the actual accuracy obtained by training under those unseen settings. Figure 5 shows parity plots (predicted vs. actual): scaling predictions tightly follow the identity line (e.g., MAE = 0.0214), while coverage predictions are systematically conservative (below the identity line) because our coverage estimator targets the *worst-case* performance loss over configurations within a covered neighborhood, yielding a risk-aware lower bound on unseen-configuration accuracy. Despite this conservativeness, predictions remain strongly aligned with actual accuracies across unseen configurations, providing direct

empirical support for both laws under genuine distribution shifts. Detailed analyses are provided in Appendix F.6.

**Trajectory-local contractivity.** Our theory relies on a trajectory-local stability condition rather than global convexity or a global PL inequality. We empirically check this condition by directly probing the local contractivity coefficient $\hat{\rho}_t$ during training. As shown in Figure 4, across datasets, distillation methods, and distilled set sizes $k$, post-burn-in updates satisfy $\hat{\rho}_t < 1$ with a stable positive margin, typically $10^{-5}$–$10^{-6}$.

## 8. Conclusion

We propose a configuration-dynamics-error framework that unifies gradient-, distribution-, and trajectory-matching distillation. Our analysis reveals two dataset distillation laws: (i) a scaling law, under which the generalization gap decreases at rate $1/\sqrt{k}$ with the distilled sample size before reaching a configuration-dependent floor, and (ii) a coverage law, under which the required distilled sample size scales linearly with configuration diversity. Together, these laws characterize a utility boundary for dataset distillation and provide guidance for theory-driven, configuration-robust method design.

**Limitations.** While we study a representative ConvNet-based target configuration family and use $H_{\text{cov}}$ as a practical coverage proxy, the extension to broader protocols and more systematically comparing proxies remains future work (details in Appendix G).

## Impact Statement

This paper presents work whose goal is to advance the theoretical understanding of dataset distillation within machine learning. The broader societal and ethical implications are those commonly associated with improved efficiency and generalization in machine learning systems, and no specific additional impacts are highlighted here.

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

# Appendix Contents

# A. Notations

# B. Related Work

## B.1. Methodological Advances in Dataset Distillation

**Gradient Matching (GM).** Dataset distillation was first formulated via gradient matching, where synthetic data are optimized to align the training gradients of real data (Zhao et al., 2020). Differentiable Siamese augmentation improved stability and generalization across pipelines (Zhao & Bilen, 2021). Later works explored diversity regularization (Cazenavette et al., 2022), representative matching (DREAM) (Liu et al., 2023), and feature regression (FRePo) (Zhou et al., 2022). Engineering efforts such as DC-BENCH (Cui et al., 2022) provided standardized evaluation. Despite progress, GM often struggles with cross-architecture transfer.

**Distribution Matching (DM).** Distribution-based methods align feature or embedding distributions rather than raw gradients. Zhao and Bilen (Zhao & Bilen, 2023) proposed matching distributions through MMD in intermediate feature space. CAFe introduced hierarchical alignment (Wang et al., 2022), while M3D (Zhang et al., 2024) and latent quantile matching (LQM) (Wei et al., 2024) further improved statistical robustness. DM methods emphasize scalability and robustness, with strong performance under augmentation shifts.

**Trajectory Matching (TM).** Trajectory matching aligns the full optimization dynamics rather than single-step gradients. The MTT framework (Cazenavette et al., 2022) established this principle, later improved by truncated backpropagation (Kim et al., 2022) and automated trajectory design (Liu et al., 2024a). Extensions include self-supervised distillation (Zhou et al., 2024). TM captures optimization pathways faithfully but is computationally demanding.

**Beyond GM/DM/TM.** Generative priors constrain synthetic samples in pretrained generative latent spaces, e.g., GLaD (Cazenavette et al., 2023) and diffusion-based methods (Su et al., 2024). Large-scale efforts like SRe$^2$L (Yin et al., 2023) and TESLA (Cui et al., 2023) scale distillation to ImageNet-1K. Beyond images, condensation has been extended to graphs (Liu et al., 2024b; Zheng & Li, 2023) and adversarially robust regimes (Sun et al., 2024). Label-centric approaches argue that soft labels often dominate performance gains (Sucholutsky & Schonlau, 2021; Chen et al., 2023). These directions illustrate the versatility of dataset distillation across domains and modalities. *We do not directly evaluate generative-based methods in our experiments, since our focus is on canonical optimization-driven procedures (GM/DM/TM). Nevertheless, our theoretical framework is general and applies equally to settings where synthetic data are produced by generative priors or large-scale latent models, as the alignment–risk perspective only depends on the induced empirical distributions rather than the mechanism of synthesis.*

## B.2. Theoretical Explorations in Dataset Distillation

**Kernel and NTK perspectives.** Several works analyze distillation under kernel ridge regression or NTK approximations. KIP (Nguyen et al., 2020) and its infinite-width extension (Nguyen et al., 2021) provided conditions for exact recovery.

*Table 2.* Notations for preliminaries 2, configuration-dynamic-risk framework 3, single–configuration 4, and configurations coverage 5.

| Symbol | Meaning |
| --- | --- |
| **Data & Distributions** | |
| $\mathcal{D}_\tau, \mu_\tau$ | Real training dataset and its population distribution. |
| $\hat{\mu}_\tau$ | Empirical measure of $\mathcal{D}_\tau$. |
| $\mathcal{D}_s, \mu_s, \hat{\mu}_s$ | Distilled dataset, its ideal/prototype distribution, and its empirical measure (support size $k$). |
| $\mathcal{P}_k$ | Class of prototype measures supported on at most $k$ atoms. |
| $\nu, \hat{\nu}$ | Test distribution and its empirical counterpart of size $m$. |
| $z = (x, y)$ | A data point used in the loss $\ell(\theta; z)$. |
| **configurations & Geometry** | |
| $\mathcal{C}, \mathcal{A} \subseteq \mathcal{C}$ | Universe of training configurations; target subfamily for generalization. |
| $a \in \mathcal{A}$ | A concrete configuration (optimizer, hyperparameters, augmentation, architecture). |
| $\Gamma, \Gamma_a$ | Common feasible set; feasible set associated with configuration $a$. |
| $d_{\mathcal{A}}(a, a')$ | configuration-distance (update-field discrepancy between $a$ and $a'$). |
| $\mathcal{H}_{\text{cov}}(\mathcal{A}, r)$ | Coverage complexity: $\log N(\mathcal{A}, d_{\mathcal{A}}, r)$. |
| $\Pi$ | Prior over configurations (e.g., uniform over a $\varrho$-cover/packing). |
| **Losses & Risks** | |
| $\ell(\theta; z)$ | Per-sample loss. |
| $R_\nu(\theta)$ | Population risk under distribution $q$: $R_\nu = \mathbb{E}_\nu \ell(\theta; z)$. |
| $\hat{R}(\theta)$ | Empirical risk under $\hat{\nu}$. |
| **Dynamics & Regularity** | |
| $\theta_t, \theta_T$ | Parameters at step $t$ and after $T$ steps. |
| $\Phi_a(\theta; \mu)$ | Update map under configuration $a$ and data $\mu$: $\theta_{t+1} = \Phi_a(\theta_t; \mu)$. |
| $P_a(\theta)$ | Preconditioner/metric of configuration $a$ (e.g., adaptive gradient metric). |
| $g_a(\theta; z)$ | Per-sample update field used by configuration $a$. |
| $\eta$ | Inner-loop step size (learning rate). |
| $\rho_{a,t} \in (0, 1)$ | Trajectory-local contraction factor at step $t$ under configuration $a$. |
| $C_{2,a}^{\text{traj}}$ | Trajectory accumulation constant $\sum_{t=0}^{T-1} \prod_{s=t+1}^{T-1} \rho_{a,s}$. |
| $\kappa_a$ | Uniform bound on $\|P_a(\theta)\|$ over $\Gamma$. |
| $B_g, B_\ell$ | Bounds on $\|g_a(\theta; z)\|$ and $|\ell(\theta; z)|$, respectively. |
| $\delta_0$ | Initialization mismatch: $\delta_0 = \theta_0^{(s)} - \theta_0^{(\tau)}$. |
| **Alignment & Aggregation** | |
| $\Delta_a(\mu, \nu)$ | Preconditioned alignment discrepancy between data $\mu$ and $\nu$ under $a$. |
| $\Delta_a^\star$ | Irreducible alignment error at budget $k$: $\inf_{\mu \in \mathcal{P}_k} \Delta_a(\mu_\tau, \mu)$; an empirical analogue replaces $\mu_\tau$ by $\hat{\mu}_\tau$. |
| $\Delta_\sharp^\star$ | Aggregated irreducible error: $\inf_{\mu \in \mathcal{P}_k} \mathbb{E}_{a \sim \Pi} \Delta_a(\mu_\tau, \mu)$. |
| **Complexities & Rates** | |
| $R_k(\mathcal{G}_a), R_n(\mathcal{G}_a)$ | Rademacher complexities of gradient classes at sizes $k$ and $n$. |
| $R_m(\mathcal{L}_a)$ | Rademacher complexity of the test-loss class at size $m$. |
| $e_g(n, k, \varepsilon), e_{\text{te}}(m, \varepsilon)$ | Estimation terms on training/test sides (confidence $\varepsilon$). |
| $\epsilon_{\text{bound}}$ | $k$-independent error floor (optimization + irreducible alignment + statistics). |
| $C_{\text{cov}}(\mathcal{A})$ | Coverage slope controlling the $\sqrt{H_{\text{cov}}/k}$ term. |
| **Information Terms** | |
| $I(D_s; D_\tau)$ | Mutual information between distilled and real training data. |
| $C_I, C_I'$ | Problem-dependent constants in MI-based corrections (avg./uniform forms). |

Random feature approximation (RFAD) improved scalability (Loo et al., 2022). Recent works extend these guarantees to provable bounds (Chen et al., 2024).

**Generalization and stability.** Classical results on uniform stability (Hardt et al., 2016), information-theoretic generalization (Xu & Raginsky, 2017; Bu et al., 2020), and Rademacher complexity (Bartlett & Mendelson, 2002) have been

*Table 3.* Notations for unifying various branches of dataset distillation 6.

| Symbol | Meaning |
| --- | --- |
| **Distribution-matching (DM) branch** | |
| $L_{z,a}$ | Lipschitz constant of $g_a(\theta; z)$ with respect to the data metric $d_Z$. |
| $W_1$ | 1-Wasserstein distance $W_1(\hat{\mu}_s, \hat{\mu}_\tau)$. |
| $C_k$ | RKHS aggregate norm bound, $\sup_\theta \left( \sum_j \|g_{a,j}(\theta; \cdot)\|_{\mathcal{H}_k}^2 \right)^{1/2}$. |
| $\mathrm{MMD}_k$ | Maximum mean discrepancy with kernel $k$ (kernel subscript, not the distilled-set size), measuring distributional distance in RKHS. |
| **Gradient-matching (GM) branch** | |
| $|\Theta_j|$ | Cardinality of the GM anchor set $\Theta_j$. |
| $\mathcal{M}_{\mathrm{GM}}$ | GM surrogate: anchor-averaged field gap $\frac{1}{|\Theta_j|} \sum_{\theta \in \Theta_j} \|\mathbb{E}_{\hat{\mu}_s} g_a - \mathbb{E}_{\hat{\mu}_\tau} g_a\|_2$. |
| **Trajectory-matching (TM) branch** | |
| $\omega_{\min}$ | Minimum TM weight, $\omega_{\min} := \min_t \omega_t > 0$. |
| $\mathcal{M}_{\mathrm{TM}}$ | TM surrogate, $\mathcal{M}_{\mathrm{TM}} = \sum_{t=0}^{L_b} \omega_t \|\theta_t^{(s,a)} - \theta_t^{(\tau,a)}\|_2$. |
| $\varepsilon_{\mathrm{path}}$ | Path coverage radius: maximum distance from an optimal point to the unrolled path set. |
| **Outer-loop / Branch-agnostic** | |
| $\phi \in \{\mathrm{DM}, \mathrm{GM}, \mathrm{TM}\}$ | Distillation surrogate branch (distribution-, gradient-, or trajectory-matching). |
| $C_{\phi,a}$ | Bridge constant for branch $\phi$ under configuration $a$. |
| $J$ | Number of outer-loop (bilevel) iterations. |
| $M_\phi(\xi^{(0)})$ | Initial surrogate misfit at outer-loop initialization $\xi^{(0)}$. |
| $\epsilon_{\mathrm{est}}^{(\phi)}$ | Estimation/statistical error floor for branch $\phi$. |
| $\xi$ | Outer-loop optimization variables (e.g., prototypes, labels, or weights). |
| **Abbreviations** | |
| "DM/GM/TM" | Three surrogate types: distribution-matching, gradient-matching, and trajectory-matching. |

adapted to study synthetic datasets. Convexified implicit gradient methods (Wai et al., 2020) provide insights into bilevel optimization stability, directly relevant to GM and TM.

**Scaling laws and soft labels.** Empirical analyses consistently show saturation as the number of distilled samples (IPC) increases (Cazenavette et al., 2022; Zhao & Bilen, 2023). Soft-label studies (Sucholutsky & Schonlau, 2021; Chen et al., 2023) demonstrate Pareto frontiers relating IPC and generalization accuracy, suggesting the existence of an irreducible *alignment floor*.

**Configuration and transfer.** Cross-architecture transferability remains a challenge: distilled datasets often fail under mismatched architectures or augmentations (Liu et al., 2023; Cazenavette et al., 2023). Large-scale efforts like SRe²L (Yin et al., 2023) and TESLA (Cui et al., 2023), as well as generative priors (Cazenavette et al., 2023; Su et al., 2024), can be interpreted as attempts to reduce alignment floors or expand coverage. Design space analyses (Shao et al., 2024) and diversity–realism tradeoff studies (Sun et al., 2024) further quantify configurational robustness.

**Connection to our work.** Our configuration-dynamics–risk framework unifies GM, DM, and TM as alignment instances, and establishes both the *single-configuration scaling law* and the *coverage law*. This explains the observed saturation in IPC scaling and the degradation under configuration shifts, bridging empirical phenomena with provable guarantees.

## C. Proofs for the Single-Configuration Bound

Recall the update rule for fixed training configuration $a$ with dataset distribution $\mu$ Eq. (3):

$$\theta_{t+1} = \Phi_a(\theta_t; \mu) = \theta_t - \eta \, P_a(\theta_t) \, \mathbb{E}_\mu \, g_a(\theta_t; z),$$

the empirical risks Eq. (1), and Assumption 4.1. Throughout, $\|\cdot\|$ denotes the Euclidean norm and the associated operator norm.

The distribution-aware discrepancy is defined are Eq. (5) as:

$$\Delta_a(\mu, \nu) := \sup_{\theta \in \Gamma_a} \left\| P_a(\theta)\big(\mathbb{E}_\mu g_a(\theta; z) - \mathbb{E}_\nu g_a(\theta; z)\big) \right\|_2.$$

Given a $k$-prototype class $\mathcal{P}_k$ and the real population distribution $\mu_\tau$, the *intrinsic alignment error* is

$$\Delta_a^\star = \inf_{\mu \in \mathcal{P}_k} \Delta_a(\mu_\tau, \mu).$$

## C.1. Technical Lemmas Used in the Proofs

**Lemma C.1** (Risk Lipschitz reduction to parameter mismatch). *Under Assumption 4.1, for any $\theta, \theta' \in \Gamma_a$ and any empirical test $\hat{\nu}$,*

$$\left| \hat{R}(\theta) - \hat{R}(\theta') \right| = \left| \mathbb{E}_{\hat{\nu}}\, \ell(\theta; z) - \mathbb{E}_{\hat{\nu}}\, \ell(\theta'; z) \right| \leq L_R \left\| \theta - \theta' \right\|.$$

*Proof.* Assumption 4.1 states that the test risk $R_\nu(\theta) = \mathbb{E}_\nu \ell(\theta; z)$ is $L_R$–Lipschitz on $\Gamma_a$. To obtain the same Lipschitz constant for *any* empirical measure $\hat{\nu}$ (including finite-support measures), we rely on the standard pointwise Lipschitz condition on the loss:

$$\forall z, \; \forall \theta, \theta' \in \Gamma_a: \quad |\ell(\theta; z) - \ell(\theta'; z)| \leq L_R \left\| \theta - \theta' \right\|. \tag{C.1.1}$$

Condition Eq. (C.1.1) is implied, e.g., by a uniform gradient bound $\sup_{\xi \in \Gamma_a} \left\| \nabla_\theta \ell(\xi; z) \right\| \leq L_R$ for all $z$ via the mean value theorem (e.g., Bubeck et al. 2015, Prop. B.10; Nesterov 2013, Chap. 2). It also immediately implies that every risk functional $R_\nu(\theta) = \mathbb{E}_\nu \ell(\theta; z)$—for any probability measure $\nu$ on $z$—is $L_R$–Lipschitz, since expectations preserve Lipschitz moduli (see Step 2 below). Thus, Assumption 4.1 plus Eq. (C.1.1) yields Lipschitzness uniformly over $\nu$, including $\nu = \hat{\nu}$.

By the definition $\hat{R}(\theta) = \mathbb{E}_{\hat{\nu}} \ell(\theta; z) = \frac{1}{m} \sum_{\ell=1}^{m} \ell(\theta; z_\ell^{\text{te}})$,

$$\left| \hat{R}(\theta) - \hat{R}(\theta') \right| = \left| \frac{1}{m} \sum_{\ell=1}^{m} \big( \ell(\theta; z_\ell^{\text{te}}) - \ell(\theta'; z_\ell^{\text{te}}) \big) \right|.$$

Apply triangle inequality to the finite sum (equivalently, linearity of expectation and Jensen's inequality for $|\cdot|$):

$$\left| \frac{1}{m} \sum_{\ell=1}^{m} \big( \ell(\theta; z_\ell^{\text{te}}) - \ell(\theta'; z_\ell^{\text{te}}) \big) \right| \leq \frac{1}{m} \sum_{\ell=1}^{m} \left| \ell(\theta; z_\ell^{\text{te}}) - \ell(\theta'; z_\ell^{\text{te}}) \right| = \mathbb{E}_{\hat{\nu}} \left| \ell(\theta; z) - \ell(\theta'; z) \right|.$$

Using Eq. (C.1.1) inside the expectation,

$$\mathbb{E}_{\hat{\nu}} \left| \ell(\theta; z) - \ell(\theta'; z) \right| \leq \mathbb{E}_{\hat{\nu}} \big( L_R \left\| \theta - \theta' \right\| \big) = L_R \left\| \theta - \theta' \right\|,$$

since $\left\| \theta - \theta' \right\|$ does not depend on $z$.

Combining all above equtions,

$$\left| \hat{R}(\theta) - \hat{R}(\theta') \right| \leq L_R \left\| \theta - \theta' \right\|,$$

which is the desired inequality. $\qquad\square$

**Lemma C.2** (One-step decomposition: trajectory-local contraction + two-sample drift). *Fix a configuration $a$ and a step size $\eta \in (0, 2/L)$. Let $\delta_t := \theta_t^{(s)} - \theta_t^{(\tau)}$ denote the parameter gap after $t$ iterations when training respectively on $\hat{\mu}_s$ and $\hat{\mu}_\tau$. Under Assumption 4.1, we have*

$$\left\| \delta_{t+1} \right\| \leq \rho_{a,t} \left\| \delta_t \right\| + \eta \, \Delta_a(\hat{\mu}_s, \hat{\mu}_\tau). \tag{10}$$

*Proof.* By definition of the update map Eq. (3),

$$\theta_{t+1}^{(s)} = \Phi_a(\theta_t^{(s)}; \hat{\mu}_s), \qquad \theta_{t+1}^{(\tau)} = \Phi_a(\theta_t^{(\tau)}; \hat{\mu}_\tau).$$

Hence

$$\delta_{t+1} = \Phi_a(\theta_t^{(s)}; \hat{\mu}_s) - \Phi_a(\theta_t^{(\tau)}; \hat{\mu}_\tau).$$

Insert and subtract $\Phi_a(\theta_t^{(\tau)}; \hat{\mu}_s)$, then apply the triangle inequality:

$$\|\delta_{t+1}\| \leq \underbrace{\left\|\Phi_a(\theta_t^{(s)}; \hat{\mu}_s) - \Phi_a(\theta_t^{(\tau)}; \hat{\mu}_s)\right\|}_{\text{same data distribution}} + \underbrace{\left\|\Phi_a(\theta_t^{(\tau)}; \hat{\mu}_s) - \Phi_a(\theta_t^{(\tau)}; \hat{\mu}_\tau)\right\|}_{\text{same parameter}}. \tag{C.2.1}$$

For the first term in Eq. (C.2.1), Assumption 4.1(iii) (trajectory-local stable dynamics) implies that along the realized training trajectory, the update map with fixed $\mu = \hat{\mu}_s$ is locally contractive:

$$\|\Phi_a(\theta_t^{(s)}; \hat{\mu}_s) - \Phi_a(\theta_t^{(\tau)}; \hat{\mu}_s)\| \leq \rho_{a,t} \|\theta_t^{(s)} - \theta_t^{(\tau)}\| = \rho_{a,t} \|\delta_t\|.$$

For the second term,

$$\Phi_a(\theta_t^{(\tau)}; \hat{\mu}_s) - \Phi_a(\theta_t^{(\tau)}; \hat{\mu}_\tau) = -\eta\, P_a(\theta_t^{(\tau)})\Big(\mathbb{E}_{\hat{\mu}_s} g_a(\theta_t^{(\tau)}; z) - \mathbb{E}_{\hat{\mu}_\tau} g_a(\theta_t^{(\tau)}; z)\Big).$$

Taking norms and using the definition of $\Delta_a$, yields

$$\|\Phi_a(\theta_t^{(\tau)}; \hat{\mu}_s) - \Phi_a(\theta_t^{(\tau)}; \hat{\mu}_\tau)\| \leq \eta\, \Delta_a(\hat{\mu}_s, \hat{\mu}_\tau).$$

Substituting the two estimates back into the decomposition from Eq (C.2.1) gives

$$\|\delta_{t+1}\| \leq \rho_{a,t} \|\delta_t\| + \eta\, \Delta_a(\hat{\mu}_s, \hat{\mu}_\tau),$$

which is exactly the claimed bound. □

**Lemma C.3** (Unrolling a time-varying contraction recursion). *Let $u_{t+1} \leq \rho_t\, u_t + c$ with $\rho_t \in (0,1)$, $u_0 \geq 0$, $c \geq 0$. Then for any $T \geq 1$,*

$$u_T \leq \Big(\prod_{t=0}^{T-1} \rho_t\Big) u_0 + c \sum_{t=0}^{T-1} \prod_{s=t+1}^{T-1} \rho_s.$$

*Proof.* Repeatedly apply the recursion: $u_1 \leq \rho_0 u_0 + c$, $u_2 \leq \rho_1 \rho_0 u_0 + c(\rho_1 + 1)$, ..., which yields the stated bound. □

**Lemma C.4** (Two-sample uniform deviation via Rademacher complexity). *Let $\mathcal{G}_a := \{ z \mapsto g_a(\theta; z) : \theta \in \Gamma_a \}$ be a vector-valued class with $\|g_a(\theta; z)\| \leq B_g$ for all $z$ and $\theta$. Let $\hat{\mu}_s$ and $\hat{\mu}_\tau$ be empirical measures based on $k$ synthetic samples and $n$ real samples, with population counterparts $\mu_s \in \mathcal{P}_k$ and $\mu_\tau$. Then, with probability at least $1 - \varepsilon$,*

$$\Delta_a(\hat{\mu}_s, \hat{\mu}_\tau) \leq \Delta_a^\star + \kappa_a\left[2(\mathfrak{R}_k(\mathcal{G}_a) + \mathfrak{R}_n(\mathcal{G}_a)) + B_g\sqrt{2\log\tfrac{4}{\varepsilon}}\Big(\tfrac{1}{\sqrt{k}} + \tfrac{1}{\sqrt{n}}\Big)\right]$$

*Proof.*

**Step C.4.0 Scalarization and Rademacher conventions.** Let $\|\cdot\|_*$ be the dual norm of $\|\cdot\|$. For any signed measure $\nu$ and any $\theta$,

$$\left\|\mathbb{E}_\nu g_a(\theta; z)\right\| = \sup_{\|v\|_* \leq 1} \big\langle v, \mathbb{E}_\nu g_a(\theta; z)\big\rangle = \sup_{\|v\|_* \leq 1} \mathbb{E}_\nu \big\langle v, g_a(\theta; z)\big\rangle.$$

Hence all vector deviations can be reduced to a scalar class

$$\mathcal{F}_a := \big\{ f_{\theta,v}(z) := \langle v, g_a(\theta; z)\rangle \,\big|\, \theta \in \Gamma_a,\ \|v\|_* \leq 1\big\}, \qquad |f_{\theta,v}(z)| \leq B_g.$$

We use the (empirical) Rademacher complexity with the factor "2" built in (Mohri et al. 2018, Def. 3.1): for a sample $U = (z_1, \ldots, z_m)$,

$$\widehat{\mathfrak{R}}_U(\mathcal{F}_a) := \mathbb{E}_\sigma\Big[\frac{2}{m} \sup_{f \in \mathcal{F}_a} \sum_{i=1}^m \sigma_i f(z_i)\Big], \qquad \mathfrak{R}_m(\mathcal{F}_a) := \mathbb{E}_U \widehat{\mathfrak{R}}_U(\mathcal{F}_a).$$

Under this normalization, the standard uniform deviation bound (Mohri et al. 2018, theorem 3.3; see also Bartlett & Mendelson 2002; Xu et al. 2016) reads: for any $\varepsilon \in (0,1)$, with probability at least $1 - \varepsilon$,

$$\sup_{f \in \mathcal{F}_a} \left| \mathbb{E}f - \mathbb{E}_{\hat{U}} f \right| \leq \widehat{\mathfrak{R}}_U(\mathcal{F}_a) + B_g \sqrt{\frac{\log(2/\varepsilon)}{2m}}. \tag{11}$$

(The $\widehat{\mathfrak{R}}_U$ term can be further upper bounded by $\mathfrak{R}_m(\mathcal{F}_a)$ in expectation; we will retain $\mathfrak{R}_m(\cdot)$ notation below. A direct treatment of the vector norm via the vector-contraction inequality Maurer, 2016, theorem 3 yields the same dependence on the scalarized class.)

**Step C.4.1 Four-point decomposition with a free $k$-prototype.** Fix any $\mu \in \mathcal{P}_k$. For every $\theta$,

$$\mathbb{E}_{\hat{\mu}_s} g_a(\theta) - \mathbb{E}_{\hat{\mu}_\tau} g_a(\theta) = \underbrace{\left(\mathbb{E}_{\hat{\mu}_s} - \mathbb{E}_{\mu_s}\right) g_a(\theta)}_{\text{empirical vs. population on } S} + \underbrace{\left(\mathbb{E}_{\mu_s} - \mathbb{E}_\mu\right) g_a(\theta)}_{\text{bridge within } \mathcal{P}_k}$$
$$+ \underbrace{\left(\mathbb{E}_\mu - \mathbb{E}_{\mu_\tau}\right) g_a(\theta)}_{\text{alignment to population } T} + \underbrace{\left(\mathbb{E}_{\mu_\tau} - \mathbb{E}_{\hat{\mu}_\tau}\right) g_a(\theta)}_{\text{empirical vs. population on } T} .$$

Taking $\|\cdot\|$ and the supremum over $\theta \in \Gamma_a$, then applying the triangle inequality,

$$\sup_\theta \left\| \mathbb{E}_{\hat{\mu}_s} g_a(\theta) - \mathbb{E}_{\hat{\mu}_\tau} g_a(\theta) \right\| \leq \underbrace{\sup_\theta \left\| \mathbb{E}_{\hat{\mu}_s} g_a(\theta) - \mathbb{E}_{\mu_s} g_a(\theta) \right\|}_{=:\Delta_S} + \underbrace{\sup_\theta \left\| \mathbb{E}_{\mu_s} g_a(\theta) - \mathbb{E}_\mu g_a(\theta) \right\|}_{=:B(\mu)}$$
$$+ \underbrace{\sup_\theta \left\| \mathbb{E}_\mu g_a(\theta) - \mathbb{E}_{\mu_\tau} g_a(\theta) \right\|}_{=:A_{\text{pop}}(\mu)} + \underbrace{\sup_\theta \left\| \mathbb{E}_{\hat{\mu}_\tau} g_a(\theta) - \mathbb{E}_{\mu_\tau} g_a(\theta) \right\|}_{=:\Delta_T}. \tag{12}$$

Choose $\mu = \mu_s \in \mathcal{P}_k$ (the population behind the $k$-prototype), so that $B(\mu_s) = 0$. The corresponding preconditioned intrinsic floor is

$$\Delta_a^\star := \inf_{\mu \in \mathcal{P}_k} \Delta_a(\mu_\tau, \mu) = \inf_{\mu \in \mathcal{P}_k} \sup_{\theta \in \Gamma_a} \left\| P_a(\theta) \left( \mathbb{E}_{\mu_\tau} g_a(\theta) - \mathbb{E}_\mu g_a(\theta) \right) \right\|.$$

Since $\|P_a(\theta)\| \leq \kappa_a$, the empirical fluctuations around $\mu_s$ and $\mu_\tau$ enter the preconditioned discrepancy with an additional factor $\kappa_a$. Then from Eq. (12),

$$\Delta_a(\hat{\mu}_s, \hat{\mu}_\tau) \leq \Delta_a^\star + \kappa_a(\Delta_S + \Delta_T). \tag{13}$$

**Step C.4.2 Bound $\Delta_S$ by Rademacher complexity.** By duality and scalarization (the class is symmetric so "$\sup(\cdot)$" equals "$\sup |\cdot|$"),

$$\Delta_S = \sup_\theta \sup_{\|v\|_* \leq 1} \left| \mathbb{E}_{\hat{\mu}_s} \langle v, g_a(\theta; z) \rangle - \mathbb{E}_{\mu_s} \langle v, g_a(\theta; z) \rangle \right| = \sup_{f \in \mathcal{F}_a} \left| \mathbb{E}_{\hat{\mu}_s} f - \mathbb{E}_{\mu_s} f \right|.$$

Applying Eq. (11) with $m = k$ and failure probability $\varepsilon/2$ and using $|f| \leq B_g$,

$$\Delta_S \leq \widehat{\mathfrak{R}}_S(\mathcal{F}_a) + B_g \sqrt{\frac{\log(4/\varepsilon)}{2k}} \leq \mathfrak{R}_k(\mathcal{F}_a) + B_g \sqrt{\frac{\log(4/\varepsilon)}{2k}}. \tag{14}$$

**Step C.4.3 Bound $\Delta_T$ analogously.** A symmetric argument for $T$ with size $n$ yields, with probability $\geq 1 - \varepsilon/2$,

$$\Delta_T = \sup_{f \in \mathcal{F}_a} \left| \mathbb{E}_{\hat{\mu}_\tau} f - \mathbb{E}_{\mu_\tau} f \right| \leq \widehat{\mathfrak{R}}_T(\mathcal{F}_a) + B_g \sqrt{\frac{\log(4/\varepsilon)}{2n}} \leq \mathfrak{R}_n(\mathcal{F}_a) + B_g \sqrt{\frac{\log(4/\varepsilon)}{2n}}. \tag{15}$$

**Step C.4.4 Combining all and applying union bound.** Combining Eq. (13), Eq. (14), and Eq. (15), and taking a union bound over the two events (each failing with prob. $\leq \varepsilon/2$), we obtain that with probability at least $1 - \varepsilon$,

$$\Delta_a(\hat{\mu}_s, \hat{\mu}_\tau) \leq \Delta_a^\star + \kappa_a \left( \widehat{\mathfrak{R}}_S(\mathcal{F}_a) + \widehat{\mathfrak{R}}_T(\mathcal{F}_a) + B_g \sqrt{\frac{\log(4/\varepsilon)}{2}} \left( \frac{1}{\sqrt{k}} + \frac{1}{\sqrt{n}} \right) \right).$$

Replacing the empirical complexities by their expectations gives a sample-independent version with $\mathfrak{R}_k(\mathcal{F}_a) + \mathfrak{R}_n(\mathcal{F}_a)$. If one uses the "no-2" normalization for Rademacher complexity (Mohri et al., 2018), the bound incurs the standard extra factor 2 in front of $\mathfrak{R}_m^{\text{std}}(\mathcal{F}_a)$. $\square$

**Lemma C.5** (Test generalization bound). *Let $\mathcal{L}_a := \{z \mapsto \ell(\theta; z) : \theta \in \Gamma_a\}$ with $|\ell(\theta; z)| \le B_\ell$. For i.i.d. test sample $\hat{\nu}$ of size $m$ from $q$, with probability at least $1 - \varepsilon$,*

$$\sup_{\theta \in \Gamma_a} \left| R_\nu(\theta) - \hat{R}(\theta) \right| \le 2\mathfrak{R}_m(\mathcal{L}_a) + B_\ell \sqrt{\frac{2\log(4/\varepsilon)}{m}}.$$

*Proof.* Standard empirical process bound via symmetrization and concentration (Mohri et al. 2018, theorem 3.3; see also Bartlett & Mendelson 2002).

**Step C.5.1 Setup.** Define

$$\Psi(U) := \sup_{f \in \mathcal{L}_a} \left| \mathbb{E}_\nu f - \mathbb{E}_{\hat{U}} f \right|, \quad \mathbb{E}_{\hat{U}} f = \frac{1}{m} \sum_{i=1}^{m} f(Z_i).$$

By symmetry,

$$\Psi(U) \le \sup_f (\mathbb{E}_\nu f - \mathbb{E}_{\hat{U}} f) + \sup_f (\mathbb{E}_{\hat{U}} f - \mathbb{E}_\nu f),$$

so it suffices to control $\Phi(U) := \sup_{f \in \mathcal{L}_a} (\mathbb{E}_\nu f - \mathbb{E}_{\hat{U}} f)$.

**Step C.5.2 Symmetrization.** Introduce an independent "ghost sample" $U' = (Z_1', \ldots, Z_m') \sim q^m$. Since $\mathbb{E}_\nu f = \mathbb{E}_{U'} \mathbb{E}_{\hat{U}'} f$, by Jensen

$$\mathbb{E}_U[\Phi(U)] \le \mathbb{E}_{U,U'} \left[ \sup_{f \in \mathcal{L}_a} \frac{1}{m} \sum_{i=1}^{m} \left( f(Z_i') - f(Z_i) \right) \right].$$

Adding Rademacher variables $\sigma_i \in \{\pm 1\}$ and applying the triangle inequality yields

$$\mathbb{E}_U[\Phi(U)] \le 2\mathbb{E}_U \mathbb{E}_\sigma \left[ \frac{1}{m} \sup_{f \in \mathcal{L}_a} \sum_{i=1}^{m} \sigma_i f(Z_i) \right] = 2\mathfrak{R}_m(\mathcal{L}_a).$$

**Step C.5.3 Concentration.** Replacing a single sample point $Z_i$ by $\tilde{Z}_i$ changes $\Phi(U)$ by at most $2B_\ell/m$, since $|f(z)| \le B_\ell$. Hence $\Phi(U)$ satisfies the bounded-differences condition, and McDiarmid's inequality gives

$$\Pr\{\Phi(U) - \mathbb{E}\Phi(U) \ge t\} \le \exp\left( -\frac{mt^2}{2B_\ell^2} \right).$$

Choosing $t = B_\ell \sqrt{\frac{2\log(2/\delta)}{m}}$ yields, with probability $\ge 1 - \delta$,

$$\Phi(U) \le \mathbb{E}\Phi(U) + B_\ell \sqrt{\frac{2\log(2/\delta)}{m}}.$$

**Step C.5.4 Combining all.** Substituting $\mathbb{E}\Phi(U) \le 2\mathfrak{R}_m(\mathcal{L}_a)$ from Step 2, and repeating the argument for $\sup_f (\mathbb{E}_{\hat{U}} f - \mathbb{E}_\nu f)$, a union bound with $\delta = \varepsilon/2$ gives

$$\sup_{f \in \mathcal{L}_a} \left| \mathbb{E}_\nu f - \mathbb{E}_{\hat{U}} f \right| \le 2\mathfrak{R}_m(\mathcal{L}_a) + B_\ell \sqrt{\frac{2\log(4/\varepsilon)}{m}}$$

with probability at least $1 - \varepsilon$. Replacing $f$ by $\ell(\theta; \cdot)$ completes the proof. $\square$

**Lemma C.6** (Information-corrected two-sample deviation). *If the distilled dataset $\mathcal{D}_s$ depends on $\mathcal{D}_\tau$, then with probability at least $1 - \varepsilon$,*

$$\Delta_a(\hat{\mu}_s, \hat{\mu}_\tau) \le \Delta_a^\star + \kappa_a \left[ 2(\mathfrak{R}_k(\mathcal{G}_a) + \mathfrak{R}_n(\mathcal{G}_a)) + B_g \sqrt{2\log\tfrac{8}{\varepsilon}} \left( \tfrac{1}{\sqrt{k}} + \tfrac{1}{\sqrt{n}} \right) \right]$$

$$+ \kappa_a C_I \sqrt{\frac{I(\mathcal{D}_s; \mathcal{D}_\tau) + \log\tfrac{8}{\varepsilon}}{k}}.$$

*Proof.* **Mutual-information (MI) tail inequality.** The only difference from Lemma C.4 is that the distilled dataset $\mathcal{D}_s$ may depend on $\mathcal{D}_\tau$. Thus, we apply a high-probability MI generalization bound (Bu et al. 2020, theorem 7; cf. Xu & Raginsky 2017, theorem 1, Steinke & Zakynthinou 2020, theorem 1): for any $\varepsilon_{\mathrm{mi}} \in (0, 1)$, with probability at least $1 - \varepsilon_{\mathrm{mi}}$,

$$\phi(S, \mathcal{D}_\tau) \leq \mathbb{E}\big[\phi(S, \mathcal{D}_\tau) \mid \mathcal{D}_\tau\big] + \sqrt{2\sigma_k^2\Big(I(S; \mathcal{D}_\tau) + \log \tfrac{1}{\varepsilon_{\mathrm{mi}}}\Big)}. \tag{16}$$

Because $S$ is (possibly randomized) post-processing of $\mathcal{D}_s$, data processing yields $I(S; \mathcal{D}_\tau) \leq I(\mathcal{D}_s; \mathcal{D}_\tau)$. Combining Eq. (14) Eq. (16), and the assumption $k << n$ gives

$$\Delta_S \leq \mathfrak{R}_k(\mathcal{G}_a) + \sqrt{\frac{2 B_g^2}{k}\Big(I(\mathcal{D}_s; \mathcal{D}_\tau) + \log \tfrac{1}{\varepsilon_{\mathrm{mi}}}\Big)} \leq \mathfrak{R}_k(\mathcal{G}_a) + C_I\sqrt{\frac{I(\mathcal{D}_s; \mathcal{D}_\tau) + \log \tfrac{1}{\varepsilon_{\mathrm{mi}}}}{k}}, \tag{17}$$

where $C_I := \sqrt{2}\, B_g$ (or a slightly larger universal constant to absorb scalarization).

**Adding the empirical fluctuation term.** Independently, the usual (conditional) concentration around the conditional mean yields, for any $\varepsilon_S \in (0, 1)$, with probability at least $1 - \varepsilon_S$,

$$\Delta_S \leq \mathfrak{R}_k(\mathcal{G}_a) + B_g\sqrt{\frac{\log(2/\varepsilon_S)}{2k}}. \tag{18}$$

A union bound over Eq. (17) and Eq. (18) will then provide both terms simultaneously.

Choose

$$\varepsilon_T = \varepsilon_S = \varepsilon_{\mathrm{mi}} = \varepsilon/4,$$

and apply a union bound (note independence is not required for a union bound). We obtain, with probability at least $1 - \varepsilon$,

$$\Delta_a(\hat{\mu}_s, \hat{\mu}_\tau) \leq \Delta_a^\star + \kappa_a \underbrace{\big(\mathfrak{R}_k(\mathcal{G}_a) + \mathfrak{R}_n(\mathcal{G}_a)\big)}_{\text{rad. complexities}}$$

$$+ \kappa_a B_g \underbrace{\sqrt{\frac{\log(8/\varepsilon)}{2}}\Big(\frac{1}{\sqrt{k}} + \frac{1}{\sqrt{n}}\Big)}_{\text{empirical concentration}} + \kappa_a C_I \underbrace{\sqrt{\frac{I(\mathcal{D}_s; \mathcal{D}_\tau) + \log(4/\varepsilon)}{k}}}_{\text{MI correction}}.$$

Switching to the common Rademacher normalization with the factor 2 (as in the lemma statement) gives $2\big(\mathfrak{R}_k(\mathcal{G}_a) + \mathfrak{R}_n(\mathcal{G}_a)\big)$, and tightening constants in the Hoeffding terms leads to $B_g\sqrt{2\log(8/\varepsilon)}\big(\frac{1}{\sqrt{k}} + \frac{1}{\sqrt{n}}\big)$. Absorbing the remaining numerical constants into $C_I$ yields

$$\Delta_a(\hat{\mu}_s, \hat{\mu}_\tau) \leq \Delta_a^\star + \kappa_a\Big[2(\mathfrak{R}_k(\mathcal{G}_a) + \mathfrak{R}_n(\mathcal{G}_a)) + B_g\sqrt{2\log\tfrac{8}{\varepsilon}}\Big(\frac{1}{\sqrt{k}} + \frac{1}{\sqrt{n}}\Big)\Big]$$

$$+ \kappa_a C_I\sqrt{\frac{I(\mathcal{D}_s; \mathcal{D}_\tau) + \log\tfrac{8}{\varepsilon}}{k}}.$$

$\square$

## C.2. Proof of Theorem 4.3

*Proof of Theorem 4.3.*
**Step C.2.1 Parameter gap.** Apply Lemma C.2 with $c := \eta\, \Delta_a(\hat{\mu}_s, \hat{\mu}_\tau)$ and $\rho_t := \rho_{a,t} \in (0, 1)$, then Lemma C.3 gives

$$\|\delta_T\| \leq \Big(\prod_{t=0}^{T-1} \rho_{a,t}\Big)\|\delta_0\| + \eta\, C_{2,a}^{\mathrm{traj}}\, \Delta_a(\hat{\mu}_s, \hat{\mu}_\tau). \tag{19}$$

**Step C.2.2 Risk gap.** By Lemma C.1,

$$\big|\hat{R}(\theta_T^{(s)}) - \hat{R}(\theta_T^{(\tau)})\big| \leq L_R\, \|\delta_T\|. \tag{20}$$

Combine Eq. (19)–(20) to obtain

$$\left|\hat{R}(\theta_T^{(s)}) - \hat{R}(\theta_T^{(\tau)})\right| \leq L_R\left(\prod_{t=0}^{T-1} \rho_{a,t}\right)\|\delta_0\| + \eta L_R C_{2,a}^{\text{traj}} \Delta_a(\hat{\mu}_s, \hat{\mu}_\tau). \tag{21}$$

**Step C.2.3 Two-sample discrepancy bound.** Apply Lemma C.4 with probability $\geq 1 - \varepsilon/2$:

$$\Delta_a(\hat{\mu}_s, \hat{\mu}_\tau) \leq \Delta_a^\star + \kappa_a\left[2(\Re_k(\mathcal{G}_a) + \Re_n(\mathcal{G}_a)) + B_g\sqrt{2\log\tfrac{4}{\varepsilon}}\left(\tfrac{1}{\sqrt{k}} + \tfrac{1}{\sqrt{n}}\right)\right].$$

Plug this into Eq. (21) and regroup the terms as

$$\eta L_R C_{2,a}^{\text{traj}}\left[\Delta_a^\star + \kappa_a\left(2(\Re_k(\mathcal{G}_a) + \Re_n(\mathcal{G}_a)) + B_g\sqrt{2\log\tfrac{4}{\varepsilon}}\left(\tfrac{1}{\sqrt{k}} + \tfrac{1}{\sqrt{n}}\right)\right)\right].$$

Define the training-side fluctuation term

$$e_g(n, k, \varepsilon) := 2(\Re_k(\mathcal{G}_a) + \Re_n(\mathcal{G}_a)) + B_g\sqrt{2\log\tfrac{4}{\varepsilon}}\left(\tfrac{1}{\sqrt{k}} + \tfrac{1}{\sqrt{n}}\right),$$

Thus,

$$\left|\hat{R}(\theta_T^{(s)}) - \hat{R}(\theta_T^{(\tau)})\right| \leq L_R\left(\prod_{t=0}^{T-1} \rho_{a,t}\right)\|\delta_0\| + \eta L_R C_{2,a}^{\text{traj}}\left(\Delta_a^\star + \kappa_a e_g(n, k, \varepsilon)\right).$$

**Step C.2.4 Test generalization.** Using Lemma C.5,

$$\left|R_\nu(\theta) - \hat{R}(\theta)\right| \leq e_{\text{te}}(m, \varepsilon) := 2\Re_m(\mathcal{L}_a) + B_\ell\sqrt{\tfrac{2\log(4/\varepsilon)}{m}}.$$

A union bound over the training part and the test part yields the bound stated in Theorem 4.3 with probability at least $1 - \varepsilon$.

**Information-corrected variant.** Replace Lemma C.4 by Lemma C.6 and repeat the steps above; this introduces the additional $C_I\sqrt{\tfrac{I(\mathcal{D}_s;\mathcal{D}_\tau)+\log(8/\varepsilon)}{k}}$ term, as claimed. $\square$

### C.3. Proof of Corollary 4.5 (Complexity Consequences)

*Proof of Corollary 4.5.* Start from Theorem 4.3:

$$\left|\hat{R}(\theta_T^{(s)}) - \hat{R}(\theta_T^{(\tau)})\right| \leq L_R\left(\prod_{t=0}^{T-1} \rho_{a,t}\right)\|\delta_0\| + \eta L_R C_{2,a}^{\text{traj}}(\Delta_a^\star + \kappa_a e_g(n, k, \varepsilon)) + e_{\text{te}}(m, \varepsilon).$$

Fix a target $\epsilon_0 > 0$ and an error split $(\beta_0, \beta_1, \beta_{\text{te}})$ with $\sum \beta = 1$. It suffices that each term is $\leq$ its budget:

$$L_R\left(\prod_{t=0}^{T-1} \rho_{a,t}\right)\|\delta_0\| \leq \beta_0\epsilon_0, \qquad \eta L_R C_{2,a}^{\text{traj}}(\Delta_a^\star + \kappa_a e_g(n, k, \varepsilon)) \leq \beta_1\epsilon_0, \qquad e_{\text{te}} \leq \beta_{\text{te}}\epsilon_0.$$

**Iterations $T$.** Under Assumption 4.1(iii), the optimization term is controlled by $\prod_{t=0}^{T-1} \rho_{a,t}$. Assume this trajectory contraction product admits an envelope decay

$$\prod_{t=0}^{T-1} \rho_{a,t} \leq \epsilon_{\text{opt}}(T),$$

where $\epsilon_{\text{opt}}(T)$ is decreasing in $T$ (e.g., exponential if $\rho_{a,t} \leq \bar{\rho} < 1$ after a burn-in). To enforce $L_R \epsilon_{\text{opt}}(T) \|\delta_0\| \leq \beta_0\epsilon_0$, it suffices that

$$\epsilon_{\text{opt}}(T) \leq \frac{\beta_0\epsilon_0}{L_R\|\delta_0\|}.$$

**Distilled samples $k$.** Under standard rates for Rademacher complexity (Bartlett & Mendelson 2002; Xu et al. 2016; Mohri et al. 2018), assume there exist $C_g, C_{\text{te}}$ such that $\Re_k(\mathcal{G}_a) \leq C_g/\sqrt{k}$, $\Re_n(\mathcal{G}_a) \leq C_g/\sqrt{n}$. Then

$$e_g(n, k, \varepsilon) \leq 2\Big(\tfrac{C_g}{\sqrt{k}} + \tfrac{C_g}{\sqrt{n}}\Big) + B_g\sqrt{2\log\tfrac{4}{\varepsilon}}\Big(\tfrac{1}{\sqrt{k}} + \tfrac{1}{\sqrt{n}}\Big).$$

Absorb constants into $C_g$ and define $C'_g := C_g + B_g\sqrt{2\log(4/\varepsilon)}$ (monotone in $\varepsilon$). Then

$$\eta L_R\, C^{\text{traj}}_{2,a}\Big(\Delta^\star_a + \kappa_a e_g\Big) \leq \eta L_R\, C^{\text{traj}}_{2,a}\Big(\Delta^\star_a + \kappa_a C'_g\big(\tfrac{1}{\sqrt{k}} + \tfrac{1}{\sqrt{n}}\big)\Big) \leq \beta_1\epsilon_0,$$

which is implied by

$$\frac{\kappa_a C'_g}{\sqrt{k}} \leq \frac{\beta_1\epsilon_0}{\eta L_R\, C^{\text{traj}}_{2,a}} - \Delta^\star_a - \frac{\kappa_a C'_g}{\sqrt{n}}, \qquad \text{hence} \qquad k \geq \Big(\frac{\kappa_a C'_g}{\frac{\beta_1}{\eta L_R\, C^{\text{traj}}_{2,a}}\epsilon_0 - \Delta^\star_a - \kappa_a C'_g/\sqrt{n}}\Big)^2.$$

**Test size $m$.** Similarly, with $\Re_m(\mathcal{L}_a) \leq C_{\text{te}}/\sqrt{m}$, Lemma C.5 gives

$$e_{\text{te}}(m, \varepsilon) \leq \frac{2C_{\text{te}}}{\sqrt{m}} + B_\ell\sqrt{\frac{2\log(4/\varepsilon)}{m}} \leq \frac{C'_{\text{te}}}{\sqrt{m}} \leq \beta_{\text{te}}\epsilon_0,$$

where $C'_{\text{te}}$ absorbs constants. Rearranging yields $m \geq \big(\frac{C'_{\text{te}}}{\beta_{\text{te}}\epsilon_0}\big)^2$. Combining the three parts proves the corollary. If $\mathcal{D}_s$ depends on $\mathcal{D}_\tau$, replace the $k$-constraint by the one obtained using Lemma C.6, which adds the $I(\mathcal{D}_s; \mathcal{D}_\tau)$ penalty inside $e_g$. $\qquad\square$

### C.4. Proof of Key Insights

*Derivation of the Key Insights.* From Theorem 4.3,

$$\big|R_\nu(\theta^{(s)}_T) - R_\nu(\theta^{(\tau)}_T)\big| \leq L_R\Big(\prod_{t=0}^{T-1} \rho_{a,t}\Big)\|\delta_0\| + \eta L_R\, C^{\text{traj}}_{2,a}(\Delta^\star_a + \kappa_a e_g(n, k, \varepsilon)) + e_{\text{te}}(m, \varepsilon).$$

(i) Let $T, k, n, m$ be sufficiently large while keeping the configuration fixed and $\varepsilon$ fixed. By Lemma C.5, $e_{\text{te}} \to 0$. By Lemma C.4, $e_g \to 0$. By Assumption 4.1(iii), the optimization dynamics along the realized training trajectory are locally stable, and the associated contraction product

$$\prod_{t=0}^{T-1} \rho_{a,t}$$

vanishes as $T$ increases in the stable regime. Therefore the optimization-induced term disappears, and the limit inferior is $\eta L_R C^{\text{traj}}_{2,a}\Delta^\star_a$, which gives the irreducible error floor.

(ii) The remaining terms vanish at their canonical rates. The optimization term is controlled by

$$L_R\Big(\prod_{t=0}^{T-1} \rho_{a,t}\Big)\|\delta_0\|,$$

which admits an exponential envelope whenever $\rho_{a,t} \leq \bar\rho < 1$ after a burn-in period, or a slower but still vanishing envelope under weaker trajectory-local stability. The statistical terms satisfy $e_g = O(1/\sqrt{k} + 1/\sqrt{n})$ by Lemma C.4, and $e_{\text{te}} = O(1/\sqrt{m})$ by Lemma C.5.

(iii) If any of the three budgets in Eq. (22) is violated, the corresponding resource must diverge (e.g., $k \to \infty$ if $\frac{\beta_1\epsilon_0}{\eta L_R\, C^{\text{traj}}_{2,a}} \downarrow \Delta^\star_a + \kappa_a C'_g/\sqrt{n}$, making the target error unattainable under finite resources. This establishes the stated resource tradeoff. $\quad\square$

# D. Proofs for the Coverage–Aware Bounds

This section provides detailed proof of the configuration coverage theorem. First, we incorporate (i) an explicit transfer analysis from cover centers to arbitrary configurations, (ii) a union-of-classes Rademacher argument with exact constants, (iii) the appearance of both $\sqrt{\mathcal{H}_{\mathrm{cov}}(\mathcal{A}, r)/k}$ and $\mathcal{H}_{\mathrm{cov}}(\mathcal{A}, r)/k$ terms in the prior-averaged bound via Bernstein-type deviations, and (iv) mutually consistent mutual-information corrections. We keep all notation from Sections. 3–5 and Appendix C. Throughout, $\|\cdot\|$ is the Euclidean/operator norm, and we write $\mathcal{H}_{\mathrm{cov}}(r)$ as shorthand for $\mathcal{H}_{\mathrm{cov}}(\mathcal{A}, r)$ when $\mathcal{A}$ is fixed.

Then, we recall the assumptions used in this proof in addition to Assumption 4.1.

**Assumption D.1** (Total boundedness and measurability). The metric space $(\mathcal{A}, d_{\mathcal{A}})$ is totally bounded (hence admits finite $r$-covers for any $r > 0$). For each $a \in \mathcal{A}$, the feasible set $\Gamma_a \subset \mathbb{R}^p$ is closed and the optimization trajectories $\{\theta_t^{(s,a)}\}_{t \leq T}, \{\theta_t^{(\tau,a)}\}_{t \leq T}$ remain in a common compact $\Gamma \subset \bigcap_{a \in \mathcal{A}} \Gamma_a$. The vector-field class $\mathcal{G}_a = \{z \mapsto g_a(\theta; z) : \theta \in \Gamma\}$ is pointwise separable and uniformly bounded:

$$\sup_{a \in \mathcal{A}} \sup_{\theta \in \Gamma} \sup_z \|g_a(\theta; z)\| \leq B_g, \qquad \sup_{a \in \mathcal{A}} \sup_{\theta \in \Gamma} \|P_a(\theta)\| \leq \kappa_{\max}.$$

**Assumption D.2** (Uniform configuration-Lipschitz transfer in $\mu$ and $\theta$). There exist constants $L_{\mathrm{conf}}, L_\theta \geq 0$ such that for all $a, a' \in \mathcal{A}$, all $\mu \in \{\hat{\mu}_\tau, \hat{\mu}_s\}$, and all $\theta, \theta' \in \Gamma$,

$$\left\| P_a(\theta)\mathbb{E}_\mu g_a(\theta; z) - P_{a'}(\theta)\mathbb{E}_\mu g_{a'}(\theta; z) \right\| \leq L_{\mathrm{conf}} \, d_{\mathcal{A}}(a, a'), \tag{22}$$

$$\left\| P_a(\theta)\mathbb{E}_\mu g_a(\theta; z) - P_a(\theta')\mathbb{E}_\mu g_a(\theta'; z) \right\| \leq L_\theta \, \|\theta - \theta'\|. \tag{23}$$

*Remarks.* (i) Inequality Eq. (22) strengthens the definition of $d_{\mathcal{A}}$ (which is anchored at $\hat{\mu}_\tau$ and fixed $\theta$) to hold uniformly over $\mu \in \{\hat{\mu}_\tau, \hat{\mu}_s\}$ and all $\theta \in \Gamma$. (ii) Inequality Eq. (23) is standard if $P_a$ and $g_a$ are Lipschitz in $\theta$ on $\Gamma$.

**Covering the configuration family.** Fix a radius $r > 0$ and let $\{a_1, \ldots, a_N\}$ be a minimal $r$-cover of $\mathcal{A}$ under $d_{\mathcal{A}}$:

$$N = N(\mathcal{A}, d_{\mathcal{A}}, r) = \exp\left(\mathcal{H}_{\mathrm{cov}}(\mathcal{A}, r)\right).$$

For any $a \in \mathcal{A}$ there exists $i(a) \in \{1, \ldots, N\}$ with $d_{\mathcal{A}}(a, a_{i(a)}) \leq r$.

**Lemma D.3** (Cross-configuration recursion under contractive dynamics). *Fix $a, a_i \in \mathcal{A}$ with $d_{\mathcal{A}}(a, a_i) \leq r$ and let $\Delta_t^{(a, a_i)} := \theta_t^{(\mu, a)} - \theta_t^{(\mu, a_i)}$ denote the parameter difference under the same data distribution $\mu$. Suppose that each one-step update $\Phi_a^\mu(\theta) = \theta - \eta P_a(\theta)\mathbb{E}_\mu[g_a(\theta; z)]$ is contractive with rate $\bar{\rho}_a \in (0, 1)$, i.e.*

$$\|\Phi_a^\mu(x) - \Phi_a^\mu(y)\| \leq \bar{\rho}_a \|x - y\|, \qquad \forall x, y \in \Gamma,$$

*Then for any step size $\eta > 0$,*

$$\|\Delta_t^{(a, a_i)}\| \leq \bar{\rho}_a^t \|\Delta_0^{(a, a_i)}\| + \frac{\eta L_{\mathrm{conf}}}{1 - \bar{\rho}_a} \, d_{\mathcal{A}}(a, a_i).$$

*Proof.* We decompose the one-step difference as

$$\Delta_{t+1}^{(a, a_i)} = \Phi_a^\mu(\theta_t^{(\mu, a)}) - \Phi_{a_i}^\mu(\theta_t^{(\mu, a_i)})$$

$$= \underbrace{\left[\Phi_a^\mu(\theta_t^{(\mu, a)}) - \Phi_a^\mu(\theta_t^{(\mu, a_i)})\right]}_{T_1} + \underbrace{\left[\Phi_a^\mu(\theta_t^{(\mu, a_i)}) - \Phi_{a_i}^\mu(\theta_t^{(\mu, a_i)})\right]}_{T_2}.$$

For the first term $T_1$, the contractive dynamics assumption gives

$$\|T_1\| \leq \bar{\rho}_a \|\theta_t^{(\mu, a)} - \theta_t^{(\mu, a_i)}\| = \bar{\rho}_a \|\Delta_t^{(a, a_i)}\|.$$

For the second term $T_2$, we compute

$$\Phi_a^\mu(\theta) - \Phi_{a_i}^\mu(\theta) = -\eta\left(P_a(\theta)\mathbb{E}_\mu g_a(\theta; z) - P_{a_i}(\theta)\mathbb{E}_\mu g_{a_i}(\theta; z)\right),$$

hence by configuration-Lipschitz continuity,

$$\|T_2\| \;\leq\; \eta L_{\mathrm{conf}}\, d_{\mathcal{A}}(a, a_i).$$

Combining the two bounds yields the one-step recursion

$$\|\Delta_{t+1}^{(a,a_i)}\| \;\leq\; \bar{\rho}_a \,\|\Delta_t^{(a,a_i)}\| + \eta L_{\mathrm{conf}}\, d_{\mathcal{A}}(a, a_i).$$

Iterating this recursion and applying the discrete Grönwall inequality, we obtain

$$\|\Delta_t^{(a,a_i)}\| \;\leq\; \bar{\rho}_a^t \|\Delta_0^{(a,a_i)}\| + \eta L_{\mathrm{conf}}\, d_{\mathcal{A}}(a, a_i) \sum_{j=0}^{t-1} \bar{\rho}_a^j.$$

Since $\sum_{j=0}^{t-1} \bar{\rho}_a^j \leq (1 - \bar{\rho}_a)^{-1}$, we conclude

$$\|\Delta_t^{(a,a_i)}\| \;\leq\; \bar{\rho}_a^t \|\Delta_0^{(a,a_i)}\| + \frac{\eta L_{\mathrm{conf}}}{1 - \bar{\rho}_a}\, d_{\mathcal{A}}(a, a_i).$$

$\square$

**Lemma D.4** (Union-of-classes Rademacher and Bernstein deviations). *Let $\{\mathcal{F}_i\}_{i=1}^N$ be classes of functions uniformly bounded by $B$. For i.i.d. sample of size $k$, for any $\varepsilon \in (0,1)$, with probability at least $1 - \varepsilon$, simultaneously for all $i$,*

$$\sup_{f \in \mathcal{F}_i} \left( \mathbb{E}f - \mathbb{E}_{\hat{S}}f \right) \leq \mathfrak{R}_k(\mathcal{F}_i) + B\sqrt{\frac{\log(2N/\varepsilon)}{2k}}, \tag{24}$$

$$\sup_{f \in \mathcal{F}_i} \left( \mathbb{E}f - \mathbb{E}_{\hat{S}}f \right) \leq c_1\, \mathfrak{R}_k(\mathcal{F}_i) + c_2\sqrt{\frac{\log(2N/\varepsilon)}{k}} + c_3\, \frac{\log(2N/\varepsilon)}{k}, \tag{25}$$

*where $c_1, c_2, c_3 > 0$ are universal constants (depending only on the choice of empirical Bernstein inequality; see, e.g., [Boucheron et al., 2003](), theorem 2.10).*

*Proof of* (24). By symmetrization (e.g. [Mohri et al., 2018](), theorem 3.1),

$$\mathbb{E}_{\hat{S}}\!\left[ \sup_{f \in \mathcal{F}} (\mathbb{E}f - \mathbb{E}_{\hat{S}}f) \right] \;\leq\; \mathbb{E}_{\hat{S}, \hat{S}'}\!\left[ \sup_{f \in \mathcal{F}} \frac{1}{k} \sum_{j=1}^k \big( f(Z_j') - f(Z_j) \big) \right] \;\leq\; 2\,\mathfrak{R}_k(\mathcal{F}), \tag{26}$$

where $\hat{S}'$ is an independent ghost sample. To pass from expectation to a high-probability bound we note that the map $\hat{S} \mapsto \sup_{f \in \mathcal{F}} (\mathbb{E}f - \mathbb{E}_{\hat{S}}f)$ is $B/k$-Lipschitz in each coordinate (changing one $Z_j$ perturbs $\mathbb{E}_{\hat{S}}f$ by at most $B/k$). Hence McDiarmid's inequality yields that, for any $\delta \in (0,1)$, with probability at least $1 - \delta$,

$$\sup_{f \in \mathcal{F}} (\mathbb{E}f - \mathbb{E}_{\hat{S}}f) \;\leq\; \mathbb{E}_{\hat{S}}\!\left[ \sup_{f \in \mathcal{F}} (\mathbb{E}f - \mathbb{E}_{\hat{S}}f) \right] \;+\; B\sqrt{\frac{\log(1/\delta)}{2k}}.$$

Combining with (26) gives, for each fixed $i$,

$$\sup_{f \in \mathcal{F}_i} (\mathbb{E}f - \mathbb{E}_{\hat{S}}f) \;\leq\; 2\,\mathfrak{R}_k(\mathcal{F}_i) \;+\; B\sqrt{\frac{\log(1/\delta)}{2k}}.$$

Since $\mathfrak{R}_k(\mathcal{F}_i) \leq 2\,\mathfrak{R}_k(\mathcal{F}_i)$ and we can absorb the factor 2 into the definition (some texts define $\mathfrak{R}_k$ with a factor 2), we present the right-hand side as $\mathfrak{R}_k(\mathcal{F}_i) + B\sqrt{\log(1/\delta)/(2k)}$. Applying a union bound over $i = 1, \dots, N$ with $\delta = \varepsilon/(2N)$ yields (24). $\square$

*Proof of* (25). We refine the concentration step by replacing Hoeffding/McDiarmid with an empirical-Bernstein deviation for bounded variables. For a fixed $f$, Boucheron et al., 2003, theorem 2.10 implies that for any $\delta \in (0,1)$,

$$\P\left( \mathbb{E}f - \mathbb{E}_{\hat{S}}f \ \geq \ \sqrt{\frac{2\,Var(f(Z))\,\log(1/\delta)}{k}} + \frac{7B\,\log(1/\delta)}{3(k-1)} \right) \ \leq \ \delta. \tag{27}$$

To make (27) uniform over $f \in \mathcal{F}_i$, we proceed by localization via symmetrization: for any $r > 0$, define the localized class $\mathcal{F}_i(r) := \{f \in \mathcal{F}_i : Var(f(Z)) \leq r\}$. By the same symmetrization step as in (26), applied to the truncated excess loss $f - \mathbb{E}f$ and then peeling over dyadic radii $r_m = 2^{-m}B^2$, we obtain (see, e.g., Mohri et al., 2018, section 3.5) that with probability at least $1 - \delta$,

$$\sup_{f \in \mathcal{F}_i} \left( \mathbb{E}f - \mathbb{E}_{\hat{S}}f \right) \ \leq \ c_1\,\mathfrak{R}_k(\mathcal{F}_i) \ + \ c_2\,\sqrt{\frac{\log(1/\delta)}{k}} \ + \ c_3\,\frac{\log(1/\delta)}{k},$$

where $c_1, c_2, c_3 > 0$ are universal constants collecting the numerical factors from: (i) the symmetrization/localization step, (ii) the empirical-Bernstein tail in (27), and (iii) the geometric peeling (finite sum over $m$). Finally, a union bound over $i = 1, \ldots, N$ with $\delta = \varepsilon/(2N)$ gives (25). $\qquad\square$

**Notation for complexity constants.** We write, for $k$-sample complexity on the distilled side and $n$-sample complexity on the real side,

$$C_G^+ := \sup_{a \in \mathcal{A}} 2\kappa_a\,\mathfrak{R}_k(\mathcal{G}_a) \ \leq \ 2\kappa_{\max} \sup_{a \in \mathcal{A}} \mathfrak{R}_k(\mathcal{G}_a), \tag{28}$$

$$\widetilde{C}_G^+ := \sup_{a \in \mathcal{A}} \left( 2\kappa_a\,\mathfrak{R}_n(\mathcal{G}_a) + B_g\sqrt{2\log\frac{4}{\varepsilon}} \right) \ \leq \ 2\kappa_{\max} \sup_a \mathfrak{R}_n(\mathcal{G}_a) + B_g\sqrt{2\log\frac{4}{\varepsilon}}. \tag{29}$$

The $n$-side quantity $\widetilde{C}_G^+$ will be collected in the $k$-independent floor.

## D.1. Proof of the Uniform Bound over Configurations in Theorem 5.2

We first prove that, with probability at least $1 - \varepsilon$ over all randomness,

$$\textit{(Uniform over configurations)} \qquad \sup_{a \in \mathcal{A}} \left| R_\nu(\theta_T^{(s,a)}) - R_\nu(\theta_T^{(\tau,a)}) \right| \ \leq \ \epsilon_{\text{bound}} \ + \ \frac{C_{\text{cov}}(\mathcal{A})}{\sqrt{k}}. \tag{30}$$

**Step D.1.1 Single-configuration risk bound at cover centers.** Fix a center $a_i$ and abbreviate $\theta_T^{(s)} := \theta_T^{(\hat{\mu}_s, a_i)}$ and $\theta_T^{(\tau)} := \theta_T^{(\hat{\mu}_\tau, a_i)}$. We first bound the *empirical test risk gap* and then convert it to the *population risk gap*.

**D.1.1(a) Empirical test risk gap at $a_i$.** By the single-configuration analysis under trajectory-local stable dynamics, we have the parameter gap

$$\|\theta_T^{(s)} - \theta_T^{(\tau)}\| \ \leq \ \left( \prod_{t=0}^{T-1} \rho_{a_i,t} \right)\|\delta_0\| \ + \ \eta\,C_{2,a_i}^{\text{traj}} \cdot \Xi_i, \qquad \Xi_i := \Delta_{a_i}(\hat{\mu}_s, \hat{\mu}_\tau).$$

Since $R_\nu$ is $L_R$-Lipschitz in $\theta$ and $\hat{R}$ averages the same bounded loss, we immediately get for the empirical test risk

$$\left| \hat{R}(\theta_T^{(s)}) - \hat{R}(\theta_T^{(\tau)}) \right| \ \leq \ L_R\left( \prod_{t=0}^{T-1} \rho_{a_i,t} \right)\|\delta_0\| \ + \ \eta\,L_R\,C_{2,a_i}^{\text{traj}}\,\Xi_i. \tag{31}$$

**D.1.1(b) From empirical to population risk at $a_i$.**

$$\left| R_\nu(\theta_T^{(s)}) - R_\nu(\theta_T^{(\tau)}) \right| \ \leq \ \underbrace{\left| \hat{R}(\theta_T^{(s)}) - \hat{R}(\theta_T^{(\tau)}) \right|}_{\text{empirical gap}} + \underbrace{\left| R_\nu(\theta_T^{(s)}) - \hat{R}(\theta_T^{(s)}) \right|}_{\text{test dev. at } \theta_T^{(s)}} + \underbrace{\left| R_\nu(\theta_T^{(\tau)}) - \hat{R}(\theta_T^{(\tau)}) \right|}_{\text{test dev. at } \theta_T^{(\tau)}}.$$

Since $\ell \in [0, B_\ell]$, Hoeffding's inequality gives, for any fixed $\theta$, with prob. $\geq 1 - \delta$, $|R_\nu(\theta) - \hat{R}(\theta)| \leq B_\ell \sqrt{\frac{\log(2/\delta)}{2m}}$. We need a bound that holds *simultaneously* for the two random iterates $\theta_T^{(s)}$ and $\theta_T^{(\tau)}$ at each center $a_i$, and then uniformly over $i$. By a union bound over the $2N$ query points (two per center), with $\delta = \varepsilon/(2N)$, we get with probability $\geq 1 - \varepsilon/2$,

$$\max_{i \in [N]} \max \left\{ |R_\nu(\theta_T^{(s,a_i)}) - \hat{R}(\theta_T^{(s,a_i)})|, \; |R_\nu(\theta_T^{(\tau,a_i)}) - \hat{R}(\theta_T^{(\tau,a_i)})| \right\} \; \leq \; B_\ell \sqrt{\frac{2 \log(4N/\varepsilon)}{m}}. \tag{32}$$

Combining (31) and (32), we obtain, with probability $\geq 1 - \varepsilon/2$, simultaneously for all centers $i$,

$$\left| R_\nu(\theta_T^{(s,a_i)}) - R_\nu(\theta_T^{(\tau,a_i)}) \right| \; \leq \; L_R \left( \prod_{t=0}^{T-1} \rho_{a_i,t} \right) \|\delta_0\| \; + \; \eta \, L_R \, C_{2,a_i}^{\text{traj}} \, \Xi_i \; + \; 2 B_\ell \sqrt{\frac{2 \log(4N/\varepsilon)}{m}}. \tag{33}$$

**Step D.1.2 Uniform control of the training-side drift $\Xi_i$.**   Recall

$$\Xi_i = \Delta_{a_i}(\hat{\mu}_s, \hat{\mu}_\tau).$$

By the definition of $\Delta_{a_i}$ and the bound $\|P_{a_i}\| \leq \kappa_{a_i}$,

$$\Xi_i \leq \kappa_{a_i} \sup_{\theta \in \Gamma} \sup_{\|v\|_* \leq 1} \left\langle v, \; \mathbb{E}_{\hat{\mu}_s} g_{a_i}(\theta; Z) - \mathbb{E}_{\hat{\mu}_\tau} g_{a_i}(\theta; Z) \right\rangle.$$

Add and subtract the population expectations under $\mu_s := \mathbb{E}[\hat{\mu}_s]$ and $\mu_\tau := \mathbb{E}[\hat{\mu}_\tau]$ (the real sampling distributions), then apply the triangle inequality:

$$\Xi_i \leq \Delta_{a_i}^\star + \kappa_{a_i} \underbrace{\sup_{\theta \in \Gamma} \left\| \mathbb{E}_{\hat{\mu}_s} g_{a_i}(\theta; Z) - \mathbb{E}_{\mu_s} g_{a_i}(\theta; Z) \right\|}_{\text{distilled sampling dev.}}$$

$$+ \kappa_{a_i} \underbrace{\sup_{\theta \in \Gamma} \left\| \mathbb{E}_{\hat{\mu}_\tau} g_{a_i}(\theta; Z) - \mathbb{E}_{\mu_\tau} g_{a_i}(\theta; Z) \right\|}_{\text{real sampling dev.}}. \tag{34}$$

Each sampling deviation term is a supremum over the function class $\mathcal{F}_i := \{ z \mapsto \langle v, g_{a_i}(\theta; z) \rangle : \theta \in \Gamma, \|v\|_* \leq 1 \}$, which is uniformly bounded by $B_g$. Applying Lemma D.4 with a union bound across the $N$ centers, we obtain, with probability at least $1 - \varepsilon/2$, simultaneously for all $i$,

$$\Xi_i \; \leq \; \Delta_{a_i}^\star \; + \; \kappa_{a_i} \left[ 2 \big( \mathfrak{R}_k(\mathcal{G}_{a_i}) + \mathfrak{R}_n(\mathcal{G}_{a_i}) \big) \; + \; B_g \sqrt{2 \log \frac{4N}{\varepsilon}} \left( \frac{1}{\sqrt{k}} + \frac{1}{\sqrt{n}} \right) \right]. \tag{35}$$

Insert (35) into (33), and upper bound the configuration-dependent constants by the uniform ones: $\prod_{t=0}^{T-1} \rho_{a_i,t} \leq \rho_{\max}^T$, $\kappa_{a_i} \leq \kappa_{\max}$, and $C_{2,a_i}^{\text{traj}} \leq C_{2,\max}^{\text{traj}} := \sup_{a \in \mathcal{A}} C_{2,a}^{\text{traj}}$. Then (33) becomes, for all $i$,

$$\left| R_\nu(\theta_T^{(s,a_i)}) - R_\nu(\theta_T^{(\tau,a_i)}) \right| \leq \; L_R \rho_{\max}^T \|\delta_0\| \; + \; \eta \, L_R \, C_{2,\max}^{\text{traj}} \, \Delta_{a_i}^\star \; + \; 2\eta \, \kappa_{\max} \, L_R \, C_{2,\max}^{\text{traj}} \, \mathfrak{R}_n(\mathcal{G}_{a_i})$$

$$+ \; 2\eta \, \kappa_{\max} \, L_R \, C_{2,\max}^{\text{traj}} \, \mathfrak{R}_k(\mathcal{G}_{a_i}) \; + \; \eta \, \kappa_{\max} \, L_R \, C_{2,\max}^{\text{traj}} \, B_g \sqrt{2 \log \frac{4N}{\varepsilon}} \left( \frac{1}{\sqrt{k}} + \frac{1}{\sqrt{n}} \right)$$

$$+ \; 2 B_\ell \sqrt{\frac{2 \log(4N/\varepsilon)}{m}}. \tag{36}$$

Using the shorthands (28)–(29) and $\sqrt{\log(4N/\varepsilon)} \leq \sqrt{\log(4/\varepsilon)} + \sqrt{\mathcal{H}_{\text{cov}}(r)}$, we isolate all $k$-independent terms into a (population) floor

$$\epsilon_{\text{bound}} := L_R \, \rho_{\max}^T \|\delta_0\| + \eta \, L_R \, C_{2,\max}^{\text{traj}} \sup_{a \in \mathcal{A}} \Delta_a^\star$$

$$+ \eta \, L_R \, C_{2,\max}^{\text{traj}} \, \widetilde{C}_G^+ \frac{1}{\sqrt{n}} + 2 B_\ell \left( \sqrt{2 \log(4/\varepsilon)} + \sqrt{2 \mathcal{H}_{\text{cov}}(r)} \right) \cdot \frac{1}{\sqrt{m}}. \tag{37}$$

and the $k$-dependent remainder (center level)

$$\eta \, L_R \, C_{2,\max}^{\text{traj}} \left( C_G^+ + 2\kappa_{\max} B_g \sqrt{2 \, \mathcal{H}_{\text{cov}}(r)} \right) \frac{1}{\sqrt{k}}. \tag{38}$$

**Step D.1.3 Transfer from cover centers to arbitrary configurations in population risk.** Fix $a \in \mathcal{A}$ and pick $i = i(a)$ with $d_{\mathcal{A}}(a, a_i) \leq r$. Consider the parameter differences at time $T$ (same distribution $\mu \in \{\hat{\mu}_s, \hat{\mu}_\tau\}$):

$$\Delta_T^{(\mu)} := \theta_T^{(\mu,a)} - \theta_T^{(\mu,a_i)}.$$

By the cross-configuration one-step decomposition (same distribution, different configurations),

$$\Delta_{t+1}^{(\mu)} = \underbrace{\left[\Phi_a^\mu(\theta_t^{(\mu,a)}) - \Phi_a^\mu(\theta_t^{(\mu,a_i)})\right]}_{\text{contraction}} + \underbrace{\left[\Phi_a^\mu(\theta_t^{(\mu,a_i)}) - \Phi_{a_i}^\mu(\theta_t^{(\mu,a_i)})\right]}_{\text{eco mismatch}},$$

we have $\|\Phi_a^\mu(x) - \Phi_a^\mu(y)\| \leq \bar{\rho}_{\max}\|x - y\|$ by contractivity. Here $\bar{\rho}_{\max} \in (0, 1)$ denotes a configuration-uniform one-step Lipschitz constant of the update map in $\theta$, used only for the cover-transfer recursion.

$$\left\|\Phi_a^\mu(\theta) - \Phi_{a_i}^\mu(\theta)\right\| = \eta\left\|P_a(\theta)\mathbb{E}_\mu g_a(\theta; Z) - P_{a_i}(\theta)\mathbb{E}_\mu g_{a_i}(\theta; Z)\right\| \leq \eta\, L_{\text{conf}}\, d_{\mathcal{A}}(a, a_i) \leq \eta\, L_{\text{conf}}\, r.$$

Therefore,

$$\|\Delta_{t+1}^{(\mu)}\| \leq \bar{\rho}_{\max}\|\Delta_t^{(\mu)}\| + \eta L_{\text{conf}} r, \qquad \Rightarrow \qquad \|\Delta_T^{(\mu)}\| \leq \bar{\rho}_{\max}^T\|\Delta_0^{(\mu)}\| + \frac{\eta L_{\text{conf}}}{1 - \bar{\rho}_{\max}}\, r.$$

As the initialization is common ($\Delta_0^{(\mu)} = 0$),

$$\max\left\{\|\theta_T^{(\hat{\mu}_s,a)} - \theta_T^{(\hat{\mu}_s,a_i)}\|,\ \|\theta_T^{(\hat{\mu}_\tau,a)} - \theta_T^{(\hat{\mu}_\tau,a_i)}\|\right\} \leq \frac{\eta L_{\text{conf}}}{1 - \bar{\rho}_{\max}}\, r =: C_{\text{path}}\, r. \tag{39}$$

Using $L_R$-Lipschitz continuity of the empirical risk,

$$\left|\hat{R}(\theta_T^{(s,a)}) - \hat{R}(\theta_T^{(s,a_i)})\right| \leq L_R C_{\text{path}}\, r, \qquad \left|\hat{R}(\theta_T^{(\tau,a)}) - \hat{R}(\theta_T^{(\tau,a_i)})\right| \leq L_R C_{\text{path}}\, r. \tag{40}$$

By the triangle inequality,

$$\left|\hat{R}(\theta_T^{(s,a)}) - \hat{R}(\theta_T^{(\tau,a)})\right| \leq \left|\hat{R}(\theta_T^{(s,a_i)}) - \hat{R}(\theta_T^{(\tau,a_i)})\right| + 2L_R C_{\text{path}}\, r. \tag{41}$$

Combining (36)–(38) with (41) and absorbing $2L_R C_{\text{path}} r$ (for fixed $r$) into $\epsilon_{\text{bound}}$ in (37), we get, *uniformly over $a \in \mathcal{A}$,*

$$\sup_{a \in \mathcal{A}}\left|R_\nu(\theta_T^{(s,a)}) - R_\nu(\theta_T^{(\tau,a)})\right| \leq \epsilon_{\text{floor}}' + \eta\, L_R\, C_{2,\max}^{\text{traj}}\left(C_G^+ + 2\kappa_{\max}B_g\sqrt{2\mathcal{H}_{\text{cov}}(r)}\right)\frac{1}{\sqrt{k}}. \tag{42}$$

where

$$\epsilon_{\text{floor}}' := \epsilon_{\text{bound}} + 2L_R C_{\text{path}} r, \tag{43}$$
$$C_{2,\max}^{\text{traj}} := \sup_{a \in \mathcal{A}} C_{2,a}^{\text{traj}},$$

and

$$C_{\text{cov}}(\mathcal{A}) := \eta\, L_R\, C_{2,\max}^{\text{traj}}\left(C_G^+ + 2\kappa_{\max}B_g\sqrt{2\mathcal{H}_{\text{cov}}(r)}\right). \tag{44}$$

to match the right-hand side of (42) with (30).

**Step D.1.4 MI correction when $\mathcal{D}_s$ may depend on $\mathcal{D}_\tau$.** If the distilled set $\mathcal{D}_s$ is generated from (or depends on) $\mathcal{D}_\tau$, the Hoeffding-type bound used for the *distilled-side* sampling deviation in (35) should be replaced by a high-probability information-theoretic tail. By Bu et al. (2020, theorem 7) (see also Xu & Raginsky, 2017; Steinke & Zakynthinou, 2020), if the class is bounded by $B_g$ (thus sub-Gaussian with proxy $B_g$), there exists a universal constant $C_I' > 0$ such that, with probability at least $1 - \varepsilon/2$,

$$\sup_{i \in [N]}\sup_{\theta \in \Gamma}\left\|\mathbb{E}_{\hat{\mu}_s} g_{a_i}(\theta; Z) - \mathbb{E}_{\mu_s} g_{a_i}(\theta; Z)\right\| \leq \mathfrak{R}_k(\mathcal{G}_{a_i}) + \frac{C_I'}{\sqrt{k}}\sqrt{I(\mathcal{D}_s; \mathcal{D}_\tau) + \log\frac{4N}{\varepsilon}}.$$

Plugging this in place of the distilled-side Hoeffding term in (35) propagates through (36)–(42) and yields the MI-corrected uniform bound

$$\sup_{a \in \mathcal{A}}\left|R_\nu(\theta_T^{(s,a)}) - R_\nu(\theta_T^{(\tau,a)})\right| \leq \epsilon_{\text{bound}} + \frac{C_{\text{cov}}'(\mathcal{A})}{\sqrt{k}} + \frac{C_I'}{\sqrt{k}}\sqrt{I(\mathcal{D}_s; \mathcal{D}_\tau)}, \tag{45}$$

where

$$C_{\text{cov}}'(\mathcal{A}) := C_{\text{cov}}(\mathcal{A}) + C_I'\left(\sqrt{\log\frac{4}{\varepsilon}} + \sqrt{\mathcal{H}_{\text{cov}}(r)}\right)$$

This proves (30); the MI term (45) can be included when dependence is present.

## D.2. Proof of the Prior-Averaged Bound in Theorem 5.2

We prove the prior-averaged statement of averaged over configurations

$$\mathbb{E}_{a\sim\Pi}\left|R_\nu(\theta_T^{(s,a)}) - R_\nu(\theta_T^{(\tau,a)})\right| \leq \epsilon_{\text{bound}} + \left[A_1 \frac{\mathcal{H}_{\text{cov}}(\mathcal{A},r)}{k} + A_2 \sqrt{\frac{\mathcal{H}_{\text{cov}}(\mathcal{A},r)}{k}}\right], \tag{46}$$

for any prior $\Pi$ supported on $\mathcal{A}$.

**Step D.2.1 Center-wise population risk gap with Bernstein refinement.** Fix a cover center $a_i$. For the two training sources $\mu \in \{\hat{\mu}_s, \hat{\mu}_\tau\}$, define $\theta_T^{(\mu)} := \theta_T^{(\mu,a_i)}$. As in the single-configuration analysis (contractive recursion and stability),

$$\|\theta_T^{(\hat{\mu}_s)} - \theta_T^{(\hat{\mu}_\tau)}\| \leq \left(\prod_{t=0}^{T-1} \rho_{a_i,t}\right)\|\delta_0\| + \eta\, C_{2,a_i}^{\text{traj}} \cdot \Xi_i, \qquad \Xi_i := \Delta_{a_i}(\hat{\mu}_s, \hat{\mu}_\tau). \tag{47}$$

Since $R_\nu$ is $L_R$-Lipschitz in $\theta$ and $\hat{R}$ averages the same bounded loss, we immediately get for the empirical test risk

$$\left|\hat{R}(\theta_T^{(\hat{\mu}_s)}) - \hat{R}(\theta_T^{(\hat{\mu}_\tau)})\right| \leq L_R\left(\prod_{t=0}^{T-1} \rho_{a_i,t}\right)\|\delta_0\| + \eta\, L_R\, C_{2,a_i}^{\text{traj}}\, \Xi_i. \tag{48}$$

We now convert (48) to a *population* gap by adding and subtracting $\hat{R}$:

$$\left|R_\nu(\theta_T^{(\hat{\mu}_s)}) - R_\nu(\theta_T^{(\hat{\mu}_\tau)})\right| \tag{49}$$
$$\leq \left|\hat{R}(\theta_T^{(\hat{\mu}_s)}) - \hat{R}(\theta_T^{(\hat{\mu}_\tau)})\right| + \left|R_\nu(\theta_T^{(\hat{\mu}_s)}) - \hat{R}(\theta_T^{(\hat{\mu}_s)})\right| + \left|R_\nu(\theta_T^{(\hat{\mu}_\tau)}) - \hat{R}(\theta_T^{(\hat{\mu}_\tau)})\right|. \tag{50}$$

Since $|\ell| \leq B_\ell$, Hoeffding yields for any fixed $\theta$ that $|R_\nu(\theta) - \hat{R}(\theta)| \leq B_\ell\sqrt{\log(2/\varepsilon)/(2m)}$ with prob. $\geq 1 - \varepsilon$. Applying a union bound to the two random iterates at each $a_i$ (and then across $i$) gives, with prob. $\geq 1 - \varepsilon/2$,

$$\max_{i\in[N]} \max\left\{|R_\nu(\theta_T^{(\hat{\mu}_s,a_i)}) - \hat{R}(\theta_T^{(\hat{\mu}_s,a_i)})|,\ |R_\nu(\theta_T^{(\hat{\mu}_\tau,a_i)}) - \hat{R}(\theta_T^{(\hat{\mu}_\tau,a_i)})|\right\} \leq B_\ell\sqrt{\frac{2\log(4N/\varepsilon)}{m}}. \tag{51}$$

Combining (48), (50), and (51) yields, uniformly over centers,

$$\left|R_\nu(\theta_T^{(\hat{\mu}_s,a_i)}) - R_\nu(\theta_T^{(\hat{\mu}_\tau,a_i)})\right| \leq L_R\left(\prod_{t=0}^{T-1} \rho_{a_i,t}\right)\|\delta_0\| + \eta\, L_R\, C_{2,a_i}^{\text{traj}}\, \Xi_i + 2\,B_\ell\sqrt{\frac{2\log(4N/\varepsilon)}{m}}. \tag{52}$$

**Drift $\Xi_i$ with union-Bernstein.** Write

$$\Xi_i \leq \Delta_{a_i}^\star + \kappa_{a_i}\sup_{\theta\in\Gamma}\left\|\mathbb{E}_{\hat{\mu}_s} g_{a_i}(\theta;Z) - \mathbb{E}_{\mu_s} g_{a_i}(\theta;Z)\right\| \tag{53}$$

$$+ \kappa_{a_i}\sup_{\theta\in\Gamma}\left\|\mathbb{E}_{\hat{\mu}_\tau} g_{a_i}(\theta;Z) - \mathbb{E}_{\mu_\tau} g_{a_i}(\theta;Z)\right\|. \tag{54}$$

Each sampling deviation is a supremum over $\mathcal{F}_i = \{z \mapsto \langle v, g_{a_i}(\theta;z)\rangle : \theta \in \Gamma, \|v\|_* \leq 1\}$, bounded by $B_g$. Applying the union-of-classes empirical-Bernstein deviation (Lemma D.4) *across the $N$ centers* gives, with prob. $\geq 1 - \varepsilon/2$, simultaneously for all $i$,

$$\Xi_i \leq \Delta_{a_i}^\star + \kappa_{a_i} c_1\big(\mathfrak{R}_k(\mathcal{G}_{a_i}) + \mathfrak{R}_n(\mathcal{G}_{a_i})\big) + c_2\sqrt{\frac{\log(2N/\varepsilon)}{k}} + c_2\sqrt{\frac{\log(2N/\varepsilon)}{n}}$$
$$+ c_3\frac{\log(2N/\varepsilon)}{k} + c_3\frac{\log(2N/\varepsilon)}{n}, \tag{55}$$

where $c_1, c_2, c_3 > 0$ are numerical constants from the empirical-Bernstein inequality.

**Center-wise population gap with explicit $k$-terms.** Insert (55) into (52); upper bound configuration-dependent constants by $\prod_{t=0}^{T-1}\rho_{a_i,t}\leq\rho_{\max}^T$, $\kappa_{a_i}\leq\kappa_{\max}$, and $C_{2,a_i}^{\mathrm{traj}}\leq C_{2,\max}^{\mathrm{traj}}:=\sup_{a\in\mathcal{A}}C_{2,a}^{\mathrm{traj}}$.. Using the shorthands $C_G^+$ and $\widetilde{C}_G^+$ and the inequality $\log(2N/\varepsilon)\leq\log(2/\varepsilon)+\mathcal{H}_{\mathrm{cov}}(r)$, we separate the $k$–independent (floor) terms:

$$\epsilon_{\mathrm{bound}}:=L_R\,\rho_{\max}^T\|\delta_0\|+\eta\,L_R\,C_{2,\max}^{\mathrm{traj}}\,\sup_{a\in\mathcal{A}}\Delta_a^\star+\eta L_R C_{2,\max}^{\mathrm{traj}}\,\widetilde{C}_G^+\,\frac{1}{\sqrt{n}}+2\,B_\ell\sqrt{\frac{2\log(4N/\varepsilon)}{m}}, \tag{56}$$

and collect the distilled-side $k$–dependence as (for some absolute constants $\bar{c}_1,\bar{c}_2>0$)

$$\big|R_\nu(\theta_T^{(\hat{\mu}_s,a_i)})-R_\nu(\theta_T^{(\hat{\mu}_\tau,a_i)})\big|\leq\epsilon_{\mathrm{bound}}+\underbrace{\bar{c}_1\eta L_R\kappa_{\max}C_{2,\max}^{\mathrm{traj}}\,\frac{\mathcal{H}_{\mathrm{cov}}(r)}{k}}_{\text{Bernstein linear term}}$$
$$+\underbrace{\bar{c}_2\eta L_R C_{2,\max}^{\mathrm{traj}}\Big(C_G^++\kappa_{\max}B_g\sqrt{\mathcal{H}_{\mathrm{cov}}(r)}\Big)\frac{1}{\sqrt{k}}}_{\text{RC and sub-Gaussian term}}. \tag{57}$$

Here we used that $\mathfrak{R}_k(\mathcal{G}_{a_i})\leq\sup_a\mathfrak{R}_k(\mathcal{G}_a)$ and $\sqrt{\log(2N/\varepsilon)}\lesssim\sqrt{\log(2/\varepsilon)}+\sqrt{\mathcal{H}_{\mathrm{cov}}(r)}$, and absorbed numerical constants into $(\bar{c}_1,\bar{c}_2)$.

**Step D.2.2 Averaging centers against the prior $\Pi$.** Let $i(a)\in[N]$ be the index of the cover center assigned to $a$ (measurable selection with $d_\mathcal{A}(a,a_i)\leq r$). Define the cell masses $p_i:=\Pi\big(\{a\in\mathcal{A}:\ i(a)=i\}\big)$ so that $\sum_{i=1}^N p_i=1$ and $\mathbb{E}_{a\sim\Pi}[\cdot]=\sum_{i=1}^N p_i\,\mathbb{E}_{a\sim\Pi(\cdot|i(a)=i)}[\cdot]$.

Taking expectation over $a\sim\Pi$ and using (57) evaluated at $i(a)$ yields

$$\mathbb{E}_{a\sim\Pi}\big|R_\nu(\theta_T^{(\hat{\mu}_s,a_{i(a)})})-R_\nu(\theta_T^{(\hat{\mu}_\tau,a_{i(a)})})\big|=\sum_{i=1}^N p_i\,\big|R_\nu(\theta_T^{(\hat{\mu}_s,a_i)})-R_\nu(\theta_T^{(\hat{\mu}_\tau,a_i)})\big|$$
$$\leq\epsilon_{\mathrm{bound}}+\bar{c}_1\eta L_R\kappa_{\max}C_{2,\max}^{\mathrm{traj}}\,\frac{\mathcal{H}_{\mathrm{cov}}(r)}{k}+\bar{c}_2\eta L_R C_{2,\max}^{\mathrm{traj}}\Big(C_G^++\kappa_{\max}B_g\sqrt{\mathcal{H}_{\mathrm{cov}}(r)}\Big)\frac{1}{\sqrt{k}}, \tag{58}$$

because the right-hand side of (57) is independent of the particular cell beyond its index $i$ and $(p_i)$ sums to 1. We now transfer from the center $a_{i(a)}$ back to the original configuration $a$.

**Step D.2.3 Prior-averaged transfer from centers to arbitrary configurations (population risk).** For each $a$, consider the parameter deviations at time $T$ under the same training distribution $\mu$:

$$\Delta_T^{(\mu)}(a):=\theta_T^{(\mu,a)}-\theta_T^{(\mu,a_{i(a)})}.$$

By the cross-configuration one-step recursion (same $\mu$, different configurations) and configuration-Lipschitz mismatch,

$$\|\Delta_{t+1}^{(\mu)}(a)\|\leq\bar{\rho}_{\max}\|\Delta_t^{(\mu)}(a)\|+\eta L_{\mathrm{conf}}\,d_\mathcal{A}(a,a_{i(a)}),$$

and because $\Delta_0^{(\mu)}(a)=0$ (same initialization), we obtain

$$\|\Delta_T^{(\mu)}(a)\|\leq\frac{\eta L_{\mathrm{conf}}}{1-\bar{\rho}_{\max}}\,d_\mathcal{A}(a,a_{i(a)})\leq\frac{\eta L_{\mathrm{conf}}}{1-\bar{\rho}_{\max}}\,r:=C_{\mathrm{path}}\,r. \tag{59}$$

By $L_R$–Lipschitz continuity of the risk $\hat{R}$,

$$\big|\hat{R}(\theta_T^{(\hat{\mu}_s,a)})-\hat{R}(\theta_T^{(\hat{\mu}_s,a_{i(a)})})\big|\leq L_R\,C_{\mathrm{path}}\,r,\qquad\big|\hat{R}(\theta_T^{(\hat{\mu}_\tau,a)})-\hat{R}(\theta_T^{(\hat{\mu}_\tau,a_{i(a)})})\big|\leq L_R\,C_{\mathrm{path}}\,r. \tag{60}$$

Hence, by triangle inequality,

$$\big|\hat{R}(\theta_T^{(\hat{\mu}_s,a)})-\hat{R}(\theta_T^{(\hat{\mu}_\tau,a)})\big|\leq\big|\hat{R}(\theta_T^{(\hat{\mu}_s,a_{i(a)})})-\hat{R}(\theta_T^{(\hat{\mu}_\tau,a_{i(a)})})\big|+2L_R\,C_{\mathrm{path}}\,r. \tag{61}$$

Taking expectation over $a \sim \Pi$ and invoking (58),

$$\mathbb{E}_{a \sim \Pi} \left| R_\nu(\theta_T^{(\hat{\mu}_s, a)}) - R_\nu(\theta_T^{(\hat{\mu}_\tau, a)}) \right| \leq \mathbb{E}_{a \sim \Pi} \left| R_\nu(\theta_T^{(\hat{\mu}_s, a_{i(a)})}) - R_\nu(\theta_T^{(\hat{\mu}_\tau, a_{i(a)})}) \right| \; + \; 2 L_R \, C_{\text{path}} \, r$$

$$\leq \epsilon_{\text{bound}} \; + \; \bar{c}_1 \eta L_R \kappa_{\max} C_{2,\max}^{\text{traj}} \frac{\mathcal{H}_{\text{cov}}(r)}{k} \; + \; \bar{c}_2 \eta L_R C_{2,\max}^{\text{traj}} \left( C_G^+ + \kappa_{\max} B_g \sqrt{\mathcal{H}_{\text{cov}}(r)} \right) \frac{1}{\sqrt{k}} \; + \; 2 L_R \, C_{\text{path}} \, r. \quad (62)$$

Since $r$ is fixed in the covering argument, we absorb the additive constant $2 L_R C_{\text{path}} r$ into $\epsilon_{\text{bound}}$ (redefining it harmlessly). This proves (46) with

$$A_1 := \bar{c}_1 \, \eta L_R \kappa_{\max} C_{2,\max}^{\text{traj}}, \qquad A_2 := \bar{c}_2 \, \eta L_R C_{2,\max}^{\text{traj}} \left( C_G^+ / \sqrt{\mathcal{H}_{\text{cov}}(r)} + \kappa_{\max} B_g \right)$$

i.e. more transparently,

$$A_1 = \Theta\left( \eta L_R \kappa_{\max} C_{2,\max}^{\text{traj}} \right), \qquad A_2 = \Theta\left( \eta L_R C_{2,\max}^{\text{traj}} \right) \cdot \left( C_G^+ / \sqrt{\mathcal{H}_{\text{cov}}(r)} + \kappa_{\max} B_g \right).$$

**Mutual-information (MI) corrections: two consistent variants** (High-probability variant). If the distilled dataset $\mathcal{D}_s$ can depend on the real dataset $\mathcal{D}_\tau$, the distilled-side sampling deviation in (55) should be replaced by a high-probability information-theoretic tail (e.g., Bu et al., 2020, theorem 7; cf. Xu & Raginsky, 2017; Steinke & Zakynthinou, 2020). There exists a universal constant $C_I' > 0$ such that, with probability at least $1 - \varepsilon$,

$$\sup_{i \in [N]} \sup_{\theta \in \Gamma} \left\| \mathbb{E}_{\hat{\mu}_s} g_{a_i}(\theta; Z) - \mathbb{E}_{\mu_s} g_{a_i}(\theta; Z) \right\| \; \leq \; \mathfrak{R}_k(\mathcal{G}_{a_i}) \; + \; \frac{C_I'}{\sqrt{k}} \sqrt{I(\mathcal{D}_s; \mathcal{D}_\tau) + \log \frac{4N}{\varepsilon}}.$$

Propagating this replacement through (55)–(62) adds

$$+ \frac{\widetilde{C}_I}{\sqrt{k}} \sqrt{I(\mathcal{D}_s; \mathcal{D}_\tau)}$$

to the right-hand side of (46), for some $\widetilde{C}_I = \Theta(\eta \kappa_{\max} C_{2,\max}^{\text{traj}})$.

**(In-expectation variant).** If one states the result *in expectation* over $(\hat{\mu}_\tau, \hat{\mu}_s, \hat{\nu})$ (dropping the $1 - \varepsilon$ qualifier), expected MI generalization bounds (e.g., Xu & Raginsky, 2017; Russo & Zou, 2016) yield a linear penalty

$$+ \frac{C_I}{k} I(D_s; D_\tau),$$

with $C_I = \Theta(\eta \kappa_{\max} C_{2,\max}^{\text{traj}})$. The rate in $\mathcal{H}_{\text{cov}}(r)$ remains the same in both variants.

**Additional Interpretations to Theorem 5.2** Combining the uniform bound Eq. (30) and the prior-averaged bound Eq. (46) yields Theorem 5.2.

**Floor terms.** In the uniform case, the floor term $\epsilon_{\text{bound}}^{\text{uni}}$ is given in Eq. (37). It aggregates all $k$-independent contributions: the transient term $L_R \rho_{\max}^T \|\delta_0\|$, the worst-case intrinsic alignment $\sup_{a \in \mathcal{A}} \Delta_a^\star$, the $n$-side deviation $\widetilde{C}_G^+$, and the test-sample concentration term. In the averaged case, the corresponding floor $\epsilon_{\text{bound}}^{\text{avg}}$ in Eq. (56) is structurally the same but uses the prior-averaged intrinsic alignment $\Delta_\sharp^\star = \mathbb{E}_{a \sim \Pi} \Delta_a^\star$ instead of the supremum.

**Coverage-dependent terms.** In the uniform inequality the constant $C_{\text{cov}}(\mathcal{A}, r)$ Eq. (44) multiplies $1/\sqrt{k}$ and captures the dependence on the covering complexity $\mathcal{H}_{\text{cov}}(r)$. It grows with both the Rademacher complexity $C_G^+$ and the envelope term $B_g \sqrt{\mathcal{H}_{\text{cov}}(r)}$. In the averaged inequality the constants $(A_1, A_2)$ appear in Eq. (46), where $A_1 \mathcal{H}_{\text{cov}}(\mathcal{A}, r)/k$ comes from the linear (Bernstein) tail $\log(N/\varepsilon)/k$, while $A_2 \sqrt{\mathcal{H}_{\text{cov}}(\mathcal{A}, r)/k}$ arises from the Rademacher and sub-Gaussian deviations.

**Why only the distilled side scales with $\mathcal{H}_{\text{cov}}(r)$.** The dependence on the covering number comes solely from the distilled side, which requires a union bound across the $N = \exp(\mathcal{H}_{\text{cov}}(r))$ cover centers. On the real-data side, all configurations share the same empirical distribution $\hat{\mu}_\tau$, so no union is needed. Consequently, $n$-side deviations remain independent of $\mathcal{H}_{\text{cov}}(r)$ and are absorbed into the floor terms.

**Choice of intrinsic alignment.** The uniform bound requires the worst-case intrinsic alignment $\sup_{a \in \mathcal{A}} \Delta_a^\star$, while the averaged bound admits the weaker prior-averaged quantity $\Delta_\sharp^\star$. This separation avoids introducing the looser maximum $\max\{\Delta_\sharp^\star, \sup_a \Delta_a^\star\}$ and keeps each statement as tight as possible for its regime.

**Mutual-information correction.** When the distilled dataset $\mathcal{D}_s$ depends on the real dataset $\mathcal{D}_\tau$, the distilled-side deviation requires an additional correction. In the high-probability setting, one obtains an additive penalty of order $\frac{C_I'}{\sqrt{k}}\sqrt{I(\mathcal{D}_s; \mathcal{D}_\tau)}$ in both uniform and averaged inequalities. In the in-expectation setting, one instead obtains a linear penalty $\frac{C_I}{k}I(\mathcal{D}_s; \mathcal{D}_\tau)$. In either case the rates $\sqrt{\mathcal{H}_{\mathrm{cov}}/k}$ and $\mathcal{H}_{\mathrm{cov}}/k$ remain unaffected.

**On the covering radius.** The covering radius $r$ is fixed throughout, and $\mathcal{H}_{\mathrm{cov}}(r)$ always refers to the coverage complexity at that scale. Optimizing $r$ affects only the constants but not the asymptotic rates $\sqrt{\mathcal{H}_{\mathrm{cov}}/k}$ or $\mathcal{H}_{\mathrm{cov}}/k$.

### D.3. Proof of Corollary 5.4

We now derive the corollary in Section 5 directly from the uniform bound in Appendix D.

**Corollary D.5** (Coverage Law (required $k$ at a fixed error)). *For any $\epsilon_0 > \epsilon_{\mathrm{bound}}$,*

$$\sup_{a \in \mathcal{A}} \left| R_\nu(\theta_T^{(s,a)}) - R_\nu(\theta_T^{(\tau,a)}) \right| \leq \epsilon_0 \quad \Longleftarrow \quad k \geq K_{\min}(\epsilon_0, \mathcal{A}) = \left( \tfrac{C_{\mathrm{cov}}(\mathcal{A})}{\epsilon_0 - \epsilon_{\mathrm{bound}}} \right)^2 = \Theta(\mathcal{H}_{\mathrm{cov}}(\mathcal{A}, r)). \tag{63}$$

*Thus,* doubling configuration diversity doubles the required distilled size.

*Proof.* Fix any target $\epsilon_0 > \epsilon_{\mathrm{bound}}$. A sufficient condition for $\sup_{a \in \mathcal{A}} |\hat{R}(\theta_T^{(s,a)}) - \hat{R}(\theta_T^{(\tau,a)})| \leq \epsilon_0$ is that the $k$–dependent term in Eq. (46) is at most $\epsilon_0 - \epsilon_{\mathrm{bound}}$:

$$\frac{C_{\mathrm{cov}}(\mathcal{A})}{\sqrt{k}} \leq \epsilon_0 - \epsilon_{\mathrm{bound}}. \tag{64}$$

Since $C_{\mathrm{cov}}(\mathcal{A}) \geq 0$ and $\epsilon_0 - \epsilon_{\mathrm{bound}} > 0$, Eq. (64) is equivalent to

$$k \geq \left( \tfrac{C_{\mathrm{cov}}(\mathcal{A})}{\epsilon_0 - \epsilon_{\mathrm{bound}}} \right)^2 =: K_{\min}(\epsilon_0, \mathcal{A}), \tag{65}$$

which proves the first displayed formula.

It remains to show $K_{\min}(\epsilon_0, \mathcal{A}) = \Theta(\mathcal{H}_{\mathrm{cov}}(\mathcal{A}, r))$. For this algebraic step, write $\mathcal{H} := \mathcal{H}_{\mathrm{cov}}(\mathcal{A}, r)$. Using Eq. (44) and the elementary inequality $(x + y)^2 \leq 2x^2 + 2y^2$ for $x, y \geq 0$,

$$K_{\min}(\epsilon_0, \mathcal{A}) = \frac{1}{(\epsilon_0 - \epsilon_{\mathrm{bound}})^2} \left[ \eta\, L_R C_{2,\max}^{\mathrm{traj}} \left( C_G^+ + 2\kappa_{\max} B_g \sqrt{2\,\mathcal{H}} \right) \right]^2 \tag{66}$$

$$\leq \frac{1}{(\epsilon_0 - \epsilon_{\mathrm{bound}})^2} \left( \eta\, L_R C_{2,\max}^{\mathrm{traj}} \right)^2 \cdot 2 \left[ (C_G^+)^2 + \left( 2\kappa_{\max} B_g \sqrt{2\,\mathcal{H}} \right)^2 \right]$$

$$= \underbrace{\frac{2\,\eta^2\, L_R^2\, (C_G^+)^2 C_{2,\max}^{\mathrm{traj}\,2}}{(\epsilon_0 - \epsilon_{\mathrm{bound}})^2}}_{=:\, C_{\mathrm{up},0}} + \underbrace{\frac{16\,\eta^2\, L_R^2\, \kappa_{\max}^2\, B_g^2 C_{2,\max}^{\mathrm{traj}\,2}}{(\epsilon_0 - \epsilon_{\mathrm{bound}})^2}}_{=:\, C_{\mathrm{up},1}}\, \mathcal{H}.$$

Hence $K_{\min}(\epsilon_0, \mathcal{A}) \leq C_{\mathrm{up},0} + C_{\mathrm{up},1}\, \mathcal{H}$ for all $\mathcal{H} \geq 0$, i.e., $K_{\min} = O(\mathcal{H})$.

For a matching lower bound, since $x \mapsto x^2$ is monotone on $x \geq 0$ and $2\, C_G^+ \geq 0$,

$$\left( C_G^+ + 2\kappa_{\max} B_g \sqrt{2\,\mathcal{H}} \right)^2 \geq \left( 2\kappa_{\max} B_g \sqrt{2\,\mathcal{H}} \right)^2 = 8\, \kappa_{\max}^2 B_g^2\, \mathcal{H}.$$

Therefore,

$$K_{\min}(\epsilon_0, \mathcal{A}) \geq \frac{1}{(\epsilon_0 - \epsilon_{\mathrm{bound}})^2} \left( \eta\, L_R C_{2,\max}^{\mathrm{traj}} \right)^2 \cdot 8\, \kappa_{\max}^2 B_g^2\, \mathcal{H} =: C_{\mathrm{low}}\, \mathcal{H}. \tag{67}$$

Combining Eq. (66)–Eq. (67), we obtain $C_{\mathrm{low}}\, \mathcal{H} \leq K_{\min}(\epsilon_0, \mathcal{A}) \leq C_{\mathrm{up},0} + C_{\mathrm{up},1}\, \mathcal{H}$. In particular, for all $\mathcal{H} \geq 1$, $K_{\min}(\epsilon_0, \mathcal{A}) \leq (C_{\mathrm{up},0} + C_{\mathrm{up},1})\, \mathcal{H}$, so $K_{\min}(\epsilon_0, \mathcal{A}) = \Theta(\mathcal{H}_{\mathrm{cov}}(\mathcal{A}, r))$. $\square$

### D.4. Proof of the Coverage Lower Bound (Theorem 5.3)

**Standing assumptions.** We use Assumption 4.1, the identifiability condition in Theorem 5.3, a $\varrho$–packing $\{a_1, \ldots, a_M\} \subset (\mathcal{A}, d_\mathcal{A})$ with $M = \exp(\mathcal{H}_{\mathrm{cov}}(\mathcal{A}, \varrho))$ up to packing-covering constants, and the uniform envelopes in App. D. We also use the configuration-Lipschitz transfer (Assumption D.2, Eq. (22)) to pass alignment statements across configurations. For each $a \in \mathcal{A}$, $\|g_a(\theta; z)\| \le B_g$, $\|P_a(\theta)\| \le \kappa_a \le \kappa_{\max}$ on $\Gamma$, and the trajectory-local dynamics have the accumulation constant $C_{2,a}^{\mathrm{traj}}$ defined in Assumption 4.1. For the lower-bound direction we additionally need two standard testing regularity conditions. First, risk gaps are locally observable in the alignment metric: for some $c_{\mathrm{obs}} > 0$, if the excess risk over the $k$-independent floor is at most $\epsilon$, then $\Delta_a(\hat\mu_\tau, \hat\mu_s) \le c_{\mathrm{obs}}\epsilon/(\eta C_{2,a}^{\mathrm{traj}})$. Second, resolving a $\varrho$-packing from a $k$-prototype distilled representation with alignment tolerance $\tilde\epsilon$ has information capacity at most

$$I(U; \hat\mu_s \mid \hat\mu_\tau) \le C_{\mathrm{cap}} \, k \Big(\frac{\tilde\epsilon}{\lambda \varrho}\Big)^2,$$

which is the usual sub-Gaussian/Fano scaling for $k$ bounded observations separated by margin $\lambda \varrho$.

**Step D.4.1 Packing and testing prior.** Pick a $\varrho$–packing $\{a_i\}_{i=1}^M$; let the hidden index $U$ be uniform on $[M]$ and $a_U$ the evaluation configuration. The distillation algorithm Alg maps $\mathcal{D}_\tau \sim \mu_\tau^n$ to $\hat\mu_s \in \mathcal{P}_k$ and does not observe $U$.

**Step D.4.2 Small risk gap $\Rightarrow$ small alignment (risk-to-alignment).** The upper-bound direction proved earlier only shows that small alignment is sufficient for small risk. For a lower bound we require the converse as an observability condition; otherwise a risk metric could hide large update mismatches. Thus, by the stated risk-alignment observability condition, if the risk gap at configuration $a$ is within $\epsilon$ of the $k$-independent floor, then

$$\Delta_a(\hat\mu_\tau, \hat\mu_s) \le \frac{c_{\mathrm{obs}}\epsilon}{\eta \, C_{2,a}^{\mathrm{traj}}}. \tag{68}$$

**Step D.4.3 Identifiability + configuration-Lipschitz $\Rightarrow$ pairwise lower bound and decoder.** For any distinct $a_i, a_j$ and any $\theta \in \Gamma$,

$$\|P_{a_i}\mathbb{E}_{\hat\mu_\tau} g_{a_i} - P_{a_j}\mathbb{E}_{\hat\mu_\tau} g_{a_j}\| \le \underbrace{\|P_{a_i}(\mathbb{E}_{\hat\mu_\tau} g_{a_i} - \mathbb{E}_{\hat\mu_s} g_{a_i})\|}_{=\Delta_{a_i}} + \underbrace{\|P_{a_i}\mathbb{E}_{\hat\mu_s} g_{a_i} - P_{a_j}\mathbb{E}_{\hat\mu_s} g_{a_j}\|}_{\le L_{\mathrm{conf}}\, d_\mathcal{A}(a_i, a_j)}$$
$$+ \underbrace{\|P_{a_j}(\mathbb{E}_{\hat\mu_s} g_{a_j} - \mathbb{E}_{\hat\mu_\tau} g_{a_j})\|}_{=\Delta_{a_j}}.$$

By identifiability at $\mu_\tau$, $\|P_{a_i}\mathbb{E}_{\hat\mu_\tau} g_{a_i} - P_{a_j}\mathbb{E}_{\hat\mu_\tau} g_{a_j}\| \ge \lambda \, d_\mathcal{A}(a_i, a_j)$. Using $d_\mathcal{A}(a_i, a_j) \ge \varrho$ (packing) and maximizing over $\theta$,

$$\Delta_{a_i} + \Delta_{a_j} \ge (\lambda - L_{\mathrm{conf}}) \, d_\mathcal{A}(a_i, a_j) \ge (\lambda - L_{\mathrm{conf}})\varrho. \tag{69}$$

Choosing (or refining) the packing so that $L_{\mathrm{conf}} \le \lambda/2$ yields

$$\Delta_{a_i} + \Delta_{a_j} \ge \lambda \varrho/2.$$

Consequently, if for the true configuration $a_U$ we have $\Delta_{a_U} \le (\lambda \varrho)/8$, then for every $i \ne U$,

$$\Delta_{a_i} \ge \lambda \varrho/2 - \Delta_{a_U} \ge 3\lambda \varrho/8 > (\lambda \varrho)/8,$$

and the decoder

$$\widehat{U}(\hat\mu_s) \in \arg\min_{i \in [M]} \Delta_{a_i}(\hat\mu_\tau, \hat\mu_s) \tag{70}$$

is correct (ties broken deterministically). Combining (68) with $\Delta_{a_U} \le (\lambda \varrho)/8$ shows that the decoder succeeds whenever

$$\epsilon \le \frac{\eta C_{2,\min}^{\mathrm{traj}}}{8 c_{\mathrm{obs}}} \, \lambda \, \varrho, \tag{71}$$

where $C_{2,\min}^{\mathrm{traj}} := \inf_{a \in \mathcal{A}} C_{2,a}^{\mathrm{traj}}$ over the packed family.

**Step D.4.4 Fano information lower bound.** If an algorithm achieved alignment tolerance $\tilde{\epsilon}$ at the true packed configuration with probability at least $3/4$, then the decoder in (70) would recover $U$ with error probability at most $1/4$. Fano's inequality gives

$$I(U; \hat{\mu}_s \mid \hat{\mu}_\tau) \ \geq \ c_F \log M$$

for a universal constant $c_F > 0$ whenever $M \geq 2$. On the other hand, the capacity regularity condition gives

$$I(U; \hat{\mu}_s \mid \hat{\mu}_\tau) \ \leq \ C_{\mathrm{cap}} \, k \Big(\frac{\tilde{\epsilon}}{\lambda_\varrho}\Big)^2 .$$

Therefore any such tolerance must satisfy

$$\tilde{\epsilon} \ \geq \ c\, \lambda_\varrho \sqrt{\frac{\log M}{k}} \ = \ \Omega\Big(\lambda_\varrho \sqrt{\frac{\mathcal{H}_{\mathrm{cov}}(\mathcal{A}, \varrho)}{k}}\Big), \tag{72}$$

where $c > 0$ is universal and packing-covering constants are absorbed.

**Conclusion.** Assume the algorithm attains, with probability at least $3/4$, the small risk gap at every packed configuration:

$$\big|\hat{R}(\theta_T^{(s,a_i)}) - \hat{R}(\theta_T^{(\tau,a_i)})\big| \ \leq \ \epsilon_{\mathrm{bound}} + \epsilon, \qquad \forall i \in [M]. \tag{73}$$

Then by (68) we have simultaneously for all $i$ $\Delta_{a_i} \leq c_{\mathrm{obs}}\epsilon/(\eta C_{2,a}^{\mathrm{traj}})$. Comparing with the necessary condition (72), we obtain

$$\epsilon \ \geq \ c' \, \eta C_{2,\min}^{\mathrm{traj}} \, \lambda_\varrho \sqrt{\frac{\log M}{k}} \ = \ \Omega\Big(\lambda_\varrho \sqrt{\frac{\mathcal{H}_{\mathrm{cov}}(\mathcal{A}, \varrho)}{k}}\Big). \tag{74}$$

Averaging over the uniform prior on the packing and absorbing all $k$–independent contributions into $\epsilon_{\mathrm{bound}}$ therefore yields

$$\mathbb{E}_a \big|R_\nu(\theta_T^{(s,a)}) - R_\nu(\theta_T^{(\tau,a)})\big| \ \geq \ \epsilon_{\mathrm{bound}} + c_{\mathrm{lb}}\lambda_\varrho \sqrt{\frac{\mathcal{H}_{\mathrm{cov}}(\mathcal{A}, \varrho)}{k}},$$

for some numerical $c_{\mathrm{lb}} \in (0,1)$. Equivalently, to achieve any target $\epsilon_0 > \epsilon_{\mathrm{bound}}$,

$$k \ \geq \ \Omega\Big(\frac{\mathcal{H}_{\mathrm{cov}}(\mathcal{A}, \varrho)}{(\epsilon_0 - \epsilon_{\mathrm{bound}})^2}\Big).$$

# E. Proofs for Unifying Distribution, Gradient, and Trajectory Matching

## E.1. Setup and Recall Assumptions

Fix a training configuration $a$ with feasible set $\Gamma_a$. The inner update follows

$$\theta_{t+1} \ = \ \Phi_a(\theta_t; \mu) \ = \ \theta_t - \eta \, P_a(\theta_t) \, \mathbb{E}_\mu g_a(\theta_t; z), \qquad t = 0, 1, \ldots, \tag{75}$$

and the outer variable $\xi$ parameterizes the synthetic distribution $\mu(\xi)$.

We assume the same *single-configuration regularity assumptions* used in section 4:

(i) $\|g_a(\theta; z)\| \leq B_g$ and $\|P_a(\theta)\| \leq \kappa_a$ on $\Gamma_a$;

(ii) the loss is $L_R$-Lipschitz in $\theta$;

(iii) *trajectory-local stable dynamics*: along the inner training trajectories induced by $\hat{\mu}_\tau$ and $\hat{\mu}_s$, there exist $\{\rho_{a,t}\}_{t=0}^{T-1}$ with $\rho_{a,t} \in (0,1)$ such that

$$\|\theta_{t+1}^{(s)} - \theta_{t+1}^{(\tau)}\| \leq \rho_{a,t} \|\theta_t^{(s)} - \theta_t^{(\tau)}\| + \eta \, \Delta_a(\hat{\mu}_\tau, \hat{\mu}_s), \qquad t = 0, \ldots, T-1. \tag{76}$$

We define the trajectory effective constant

$$C_{2,a}^{\mathrm{traj}} := \sum_{t=0}^{T-1} \prod_{s=t+1}^{T-1} \rho_{a,s}, \tag{77}$$

which is finite whenever $\rho_{a,t} < 1$ along the realized trajectory.

The alignment discrepancy is

$$\Delta_a(\mu, \nu) := \sup_{\theta \in \Gamma_a} \left\| P_a(\theta)\big(\mathbb{E}_\mu g_a(\theta; z) - \mathbb{E}_\nu g_a(\theta; z)\big) \right\|.$$

We also use the empirical risks $\widehat{R}(\theta) = \mathbb{E}_{\widehat{\nu}}\,\ell(\theta; z)$ and the risk-Lipschitz lemma $|\widehat{R}(\theta) - \widehat{R}(\theta')| \leq L_R \|\theta - \theta'\|$. Let $\delta_t := \theta_t^{(s,a)} - \theta_t^{(\tau,a)}$ denote the in-configuration parameter gap when training on $\widehat{\mu}_s$ vs. $\widehat{\mu}_\tau$.

**Outer surrogate contraction**

Let $M_\phi : \Xi \to \mathbb{R}_{\geq 0}$ be the outer surrogate for branch $\phi \in \{\mathrm{DM}, \mathrm{GM}, \mathrm{TM}\}$ as listed in section 6 (DM: $W_1$ or $\mathrm{MMD}_k$; GM: anchor-averaged squared field gap; TM: weighted path discrepancy over an $L_b$-step unroll). The outer variable $\xi \in \Xi$ parameterizes the synthetic distribution $\mu(\xi)$, and the outer update is

$$\xi^{(j+1)} \;=\; \xi^{(j)} - \eta_j\,\widehat{\nabla} M_\phi(\xi^{(j)}),$$

where $\widehat{\nabla} M_\phi(\xi^{(j)})$ denotes the (possibly stochastic) estimator of the exact gradient $\nabla M_\phi(\xi^{(j)})$ produced by mini-batching critics (DM), anchor/path sampling (GM/TM), or finite unrolls.

We assume throughout: ($L_\phi$-**smoothness in** $\xi$) $M_\phi$ is $L_\phi$-smooth:

$$M_\phi(y) \leq M_\phi(x) + \langle \nabla M_\phi(x), y - x \rangle + \frac{L_\phi}{2}\|y - x\|^2, \quad \forall x, y \in \Xi.$$

We do *not* impose a global PL/strong convexity condition on $M_\phi$. Instead, our outer-loop analysis will yield a standard *stationarity* guarantee in terms of $\frac{1}{J}\sum_{j=0}^{J-1} \mathbb{E}\|\nabla M_\phi(\xi^{(j)})\|^2$ under the estimator model below, and we absorb the resulting outer optimization residual into $\epsilon_{\mathrm{bound}}$.

(**Estimator model**) Write the gradient estimator as

$$\widehat{\nabla} M_\phi(\xi) \;=\; \nabla M_\phi(\xi) \;+\; e(\xi),$$

where $e(\xi)$ captures the randomness due to critics, anchors, finite unrolls, etc.

We will consider two subcases:

(i) *Unbiased finite-variance:* $\mathbb{E}[e(\xi) \mid \xi] = 0$ and $\mathbb{E}[\|e(\xi)\|^2 \mid \xi] \leq \sigma_\phi^2$.

(ii) *Biased-but-controlled:* $\|\mathbb{E}[e(\xi) \mid \xi]\| \leq \beta_\phi$ and $\mathbb{E}[\|e(\xi) - \mathbb{E}e(\xi)\|^2 \mid \xi] \leq \sigma_\phi^2$.

These two settings cover mini-batch $W_1$/MMD critics (variance) and approximate/implicitbackprop through unrolls (small bias).

## E.2. Proof of the Outer-Loop Progress Bound (No PL)

We provide a detailed proof for the outer-loop progress guarantee without assuming PL/strong convexity. The goal is to show that under $L_\phi$-smoothness and the estimator model, the outer update yields (i) a one-step descent inequality and (ii) a nonconvex stationarity bound.

**Step E.2.1 Apply the descent lemma at the actual update.** At iteration $b$, the outer-loop update is given by

$$\xi^{(j+1)} = \xi^{(j)} - \eta_j\,\widehat{\nabla} M_\phi(\xi^{(j)}),$$

where $\eta_j$ is the step size and $\widehat{\nabla} M_\phi(\xi^{(j)})$ is the stochastic estimator of the true gradient. Since $M_\phi$ has $L_\phi$-Lipschitz gradients, the descent lemma ensures that for any point $y$,

$$M_\phi(y) \leq M_\phi(\xi^{(j)}) + \langle \nabla M_\phi(\xi^{(j)}), y - \xi^{(j)} \rangle + \tfrac{L_\phi}{2}\|y - \xi^{(j)}\|^2.$$

Substituting $y = \xi^{(j+1)}$ yields

$$M_\phi(\xi^{(j+1)}) \leq M_\phi(\xi^{(j)}) - \eta_j \langle \nabla M_\phi(\xi^{(j)}), \widehat{\nabla} M_\phi(\xi^{(j)}) \rangle + \tfrac{L_\phi}{2}\eta_j^2 \|\widehat{\nabla} M_\phi(\xi^{(j)})\|^2. \tag{78}$$

This inequality expresses how the function value decreases after one update, up to a quadratic correction controlled by $L_\phi$.

**Step E.2.2 Expand the estimator and regroup.** We next separate the stochastic gradient estimator into the true gradient plus an error term:

$$\widehat{\nabla} M_\phi(\xi^{(j)}) = \nabla M_\phi(\xi^{(j)}) + e(\xi^{(j)}).$$

Substituting into Eq. (78), we expand and regroup terms:

$$M_\phi(\xi^{(j+1)}) \le M_\phi(\xi^{(j)}) - \eta_j \|\nabla M_\phi(\xi^{(j)})\|^2 - \eta_j \langle \nabla M_\phi(\xi^{(j)}), e(\xi^{(j)}) \rangle$$
$$+ \tfrac{L_\phi}{2}\eta_j^2 \Big( \|\nabla M_\phi(\xi^{(j)})\|^2 + 2\langle \nabla M_\phi(\xi^{(j)}), e(\xi^{(j)}) \rangle + \|e(\xi^{(j)})\|^2 \Big).$$

Collecting like terms gives the compact form:

$$M_\phi(\xi^{(j+1)}) \le M_\phi(\xi^{(j)}) - \eta_j \Big( 1 - \tfrac{L_\phi}{2}\eta_j \Big) \|\nabla M_\phi(\xi^{(j)})\|^2$$
$$+ \big( -\eta_j + L_\phi \eta_j^2 \big) \langle \nabla M_\phi(\xi^{(j)}), e(\xi^{(j)}) \rangle + \tfrac{L_\phi}{2}\eta_j^2 \|e(\xi^{(j)})\|^2. \tag{79}$$

**This decomposition highlights three distinct effects: (i) a contraction term proportional to $\|\nabla M_\phi\|^2$, (ii) a cross-term coupling gradient and error, and (iii) a pure variance term $\|e\|^2$.**

**Step E.2.3 Take conditional expectation to remove the cross term.** We now take conditional expectation given $\xi^{(j)}$, analyzing two regimes of the estimator.

**Case (i): unbiased estimator.** Suppose $\mathbb{E}[e(\xi^{(j)}) \mid \xi^{(j)}] = 0$ and $\mathbb{E}[\|e(\xi^{(j)})\|^2 \mid \xi^{(j)}] \le \sigma_\phi^2$. The cross term vanishes in expectation, leaving

$$\mathbb{E}\big[ M_\phi(\xi^{(j+1)}) \mid \xi^{(j)} \big] \le M_\phi(\xi^{(j)}) - \eta_j \Big( 1 - \tfrac{L_\phi}{2}\eta_j \Big) \|\nabla M_\phi(\xi^{(j)})\|^2 + \tfrac{L_\phi}{2}\eta_j^2 \sigma_\phi^2. \tag{80}$$

**Case (ii): biased but controlled estimator.** We decompose the estimation error into a deterministic bias and a zero-mean noise conditioned on $\xi^{(j)}$:

$$e(\xi^{(j)}) = \bar{e}(\xi^{(j)}) + \tilde{e}(\xi^{(j)}), \qquad \bar{e}(\xi^{(j)}) := \mathbb{E}\Big[ e(\xi^{(j)}) \,\Big|\, \xi^{(j)} \Big], \quad \mathbb{E}\Big[ \tilde{e}(\xi^{(j)}) \,\Big|\, \xi^{(j)} \Big] = 0.$$

Assume the bias is bounded and the conditional noise has bounded second moment:

$$\|\bar{e}(\xi^{(j)})\| \le \beta_\phi, \qquad \mathbb{E}\Big[ \|\tilde{e}(\xi^{(j)})\|^2 \,\Big|\, \xi^{(j)} \Big] \le \sigma_\phi^2.$$

**Take conditional expectation and separate terms.** Conditioning on $\xi^{(j)}$ in Eq. (79), the cross term splits as

$$\mathbb{E}\Big[ \langle \nabla M_\phi(\xi^{(j)}), e(\xi^{(j)}) \rangle \,\Big|\, \xi^{(j)} \Big] = \langle \nabla M_\phi(\xi^{(j)}), \bar{e}(\xi^{(j)}) \rangle + \underbrace{\mathbb{E}\Big[ \langle \nabla M_\phi(\xi^{(j)}), \tilde{e}(\xi^{(j)}) \rangle \,\Big|\, \xi^{(j)} \Big]}_{=0},$$

where the underbraced term vanishes because $\mathbb{E}[\tilde{e}(\xi^{(j)}) \mid \xi^{(j)}] = 0$. For the quadratic error term we have

$$\mathbb{E}\Big[ \|e(\xi^{(j)})\|^2 \,\Big|\, \xi^{(j)} \Big] = \|\bar{e}(\xi^{(j)})\|^2 + \mathbb{E}\Big[ \|\tilde{e}(\xi^{(j)})\|^2 \,\Big|\, \xi^{(j)} \Big] \le \beta_\phi^2 + \sigma_\phi^2.$$

Therefore,

$$\mathbb{E}\Big[ M_\phi(\xi^{(j+1)}) \,\Big|\, \xi^{(j)} \Big] \le M_\phi(\xi^{(j)}) - \eta_j \Big( 1 - \tfrac{L_\phi}{2}\eta_j \Big) \|\nabla M_\phi(\xi^{(j)})\|^2$$
$$+ \big( -\eta_j + L_\phi \eta_j^2 \big) \langle \nabla M_\phi(\xi^{(j)}), \bar{e}(\xi^{(j)}) \rangle + \tfrac{L_\phi}{2}\eta_j^2 \big( \beta_\phi^2 + \sigma_\phi^2 \big). \tag{81}$$

**Bound the cross term by Young's inequality.** Using Cauchy–Schwarz and Young's inequality $ab \le \tfrac{\tau}{2}a^2 + \tfrac{1}{2\tau}b^2$ (valid for any $\tau > 0$), we get

$$\Big| \big( -\eta_j + L_\phi \eta_j^2 \big) \langle \nabla M_\phi(\xi^{(j)}), \bar{e}(\xi^{(j)}) \rangle \Big| \le \big| -\eta_j + L_\phi \eta_j^2 \big| \, \|\nabla M_\phi(\xi^{(j)})\| \, \|\bar{e}(\xi^{(j)})\|$$
$$\le \frac{\tau}{2} \|\nabla M_\phi(\xi^{(j)})\|^2 + \frac{1}{2\tau} \big( -\eta_j + L_\phi \eta_j^2 \big)^2 \|\bar{e}(\xi^{(j)})\|^2. \tag{82}$$

Substituting Eq. (82) into Eq. (81) yields, for any $\tau > 0$,

$$\mathbb{E}\Big[M_\phi(\xi^{(j+1)}) \,\Big|\, \xi^{(j)}\Big] \leq M_\phi(\xi^{(j)}) - \eta_j\Big(1 - \tfrac{L_\phi}{2}\eta_j\Big)\|\nabla M_\phi(\xi^{(j)})\|^2 + \tfrac{\tau}{2}\,\|\nabla M_\phi(\xi^{(j)})\|^2$$
$$+ \underbrace{\Big(\tfrac{1}{2\tau}\big(-\eta_j + L_\phi\eta_j^2\big)^2 + \tfrac{L_\phi}{2}\eta_j^2\Big)}_{\mathcal{C}_\beta(\eta_j,\tau)} \|\bar{e}(\xi^{(j)})\|^2 + \tfrac{L_\phi}{2}\eta_j^2\,\sigma_\phi^2. \tag{83}$$

**Choose $\tau$ and simplify the gradient coefficient.** Set $\tau = \eta_j/2$ (valid for any $\eta_j > 0$). Then $\tfrac{\tau}{2} = \tfrac{\eta_j}{4}$, so the two gradient terms combine as

$$-\eta_j\Big(1 - \tfrac{L_\phi}{2}\eta_j\Big)\|\nabla M_\phi\|^2 + \tfrac{\eta_j}{4}\|\nabla M_\phi\|^2 = -\Big(\eta_j - \tfrac{L_\phi}{2}\eta_j^2 - \tfrac{\eta_j}{4}\Big)\|\nabla M_\phi\|^2.$$

If we enforce the natural step-size condition $\eta_j \leq 1/L_\phi$, then

$$\eta_j - \tfrac{L_\phi}{2}\eta_j^2 \geq \tfrac{\eta_j}{2} \quad\Longrightarrow\quad \eta_j - \tfrac{L_\phi}{2}\eta_j^2 - \tfrac{\eta_j}{4} \geq \tfrac{\eta_j}{4},$$

and hence

$$-\eta_j\Big(1 - \tfrac{L_\phi}{2}\eta_j\Big)\|\nabla M_\phi\|^2 + \tfrac{\eta_j}{4}\|\nabla M_\phi\|^2 \leq -\tfrac{\eta_j}{4}\|\nabla M_\phi(\xi^{(j)})\|^2. \tag{84}$$

**Consolidate the bias-dependent coefficient $\mathcal{C}_\beta$.** With $\tau = \eta_j/2$, we have $\tfrac{1}{2\tau} = \tfrac{1}{\eta_j}$ and therefore

$$\mathcal{C}_\beta(\eta_j,\tau) = \tfrac{1}{\eta_j}\big(-\eta_j + L_\phi\eta_j^2\big)^2 + \tfrac{L_\phi}{2}\eta_j^2.$$

Note that $\big(-\eta_j + L_\phi\eta_j^2\big)^2 = \eta_j^2(1 - L_\phi\eta_j)^2 \leq \eta_j^2$. Thus,

$$\tfrac{1}{\eta_j}\big(-\eta_j + L_\phi\eta_j^2\big)^2 \leq \eta_j, \qquad \text{and} \qquad \tfrac{L_\phi}{2}\eta_j^2 \leq \tfrac{1}{2}\,\eta_j \quad \text{whenever } \eta_j \leq \tfrac{1}{L_\phi}.$$

A slightly sharper consolidation uses

$$\tfrac{1}{\eta_j}\big(-\eta_j + L_\phi\eta_j^2\big)^2 + \tfrac{L_\phi}{2}\eta_j^2 = \eta_j(1 - 2L_\phi\eta_j + L_\phi^2\eta_j^2) + \tfrac{L_\phi}{2}\eta_j^2 = \eta_j - \tfrac{3}{2}L_\phi\eta_j^2 + L_\phi^2\eta_j^3,$$

which satisfies

$$\eta_j - \tfrac{3}{2}L_\phi\eta_j^2 + L_\phi^2\eta_j^3 \leq \eta_j \qquad \text{for all } \eta_j \in [0, 1/L_\phi],$$

since the cubic correction is nonpositive over this interval: $-\tfrac{3}{2}L_\phi\eta_j^2 + L_\phi^2\eta_j^3 = L_\phi\eta_j^2\big(-\tfrac{3}{2} + L_\phi\eta_j\big) \leq 0$. Therefore

$$\mathcal{C}_\beta(\eta_j,\tau)\,\|\bar{e}(\xi^{(j)})\|^2 \leq \eta_j\,\|\bar{e}(\xi^{(j)})\|^2 \leq \eta_j\,\beta_\phi^2. \tag{85}$$

**Collect all pieces.** Combining (83), (84), and (85), and recalling $\mathbb{E}[\|\tilde{e}(\xi^{(j)})\|^2 \mid \xi^{(j)}] \leq \sigma_\phi^2$, we obtain

$$\mathbb{E}\Big[M_\phi(\xi^{(j+1)}) \,\Big|\, \xi^{(j)}\Big] \leq M_\phi(\xi^{(j)}) - \tfrac{\eta_j}{4}\|\nabla M_\phi(\xi^{(j)})\|^2 + \eta_j\,\beta_\phi^2 + \tfrac{L_\phi}{2}\eta_j^2\,\sigma_\phi^2. \tag{86}$$

This is precisely the claimed biased-case inequality: the gradient term contracts with rate $\eta_j/4$, while the estimator contributes an additive floor composed of a *bias term* $\eta_j\beta_\phi^2$ and a *variance term* $\tfrac{L_\phi}{2}\eta_j^2\sigma_\phi^2$.

**Step E.2.4 Obtain a stationarity guarantee for the outer loop (no PL).** We do not convert $\|\nabla M_\phi(\xi^{(j)})\|^2$ into a function gap. Instead, we derive a standard nonconvex stationarity bound. Assume $\eta_j \equiv \eta \leq 1/L_\phi$.

From the descent inequality obtained in Steps E.2.1–E.2.3, summing over $j = 0, \ldots, J-1$ and taking total expectation yields

$$\frac{1}{J}\sum_{j=0}^{J-1}\mathbb{E}\big\|\nabla M_\phi(\xi^{(j)})\big\|^2 \leq \frac{4\big(M_\phi(\xi^{(0)}) - M_\phi^{\inf}\big)}{\eta J} + 2L_\phi\eta\,\sigma_\phi^2 + 4\beta_\phi^2, \tag{87}$$

where $M_\phi^{\inf} := \inf_{\xi\in\Xi} M_\phi(\xi)$ and $(\sigma_\phi, \beta_\phi)$ are from the estimator model. Consequently,

$$\min_{0\leq j\leq J-1}\mathbb{E}\big\|\nabla M_\phi(\xi^{(j)})\big\|^2 \leq \frac{4\big(M_\phi(\xi^{(0)}) - M_\phi^{\inf}\big)}{\eta J} + 2L_\phi\eta\,\sigma_\phi^2 + 4\beta_\phi^2. \tag{88}$$

**Step E.2.5 Choose stepsize and define an outer residual term.** Choose $\eta \leq 1/L_\phi$. Eq. (87) implies an $O(1/J)$ optimization term plus an estimator-induced floor. We define the outer-loop residual

$$\epsilon_{\text{outer}}^{(\phi)}(J, \eta) := \frac{4\big(M_\phi(\xi^{(0)}) - M_\phi^{\text{inf}}\big)}{\eta J} + 2L_\phi \eta \, \sigma_\phi^2 + 4\beta_\phi^2, \tag{89}$$

and carry $\epsilon_{\text{outer}}^{(\phi)}$ into the final bound (absorbed into $\epsilon_{\text{bound}}$).

Taking full expectation in (80) (resp. (86)) and using $\eta_j \leq 1/L_\phi$ so that $1 - \frac{L_\phi}{2}\eta_j \geq \frac{1}{2}$, we obtain the unified one-step progress:

$$\mathbb{E}\big[M_\phi(\xi^{(j+1)})\big] \leq \mathbb{E}\big[M_\phi(\xi^{(j)})\big] - \frac{\eta_j}{4}\,\mathbb{E}\big\|\nabla M_\phi(\xi^{(j)})\big\|^2 + \eta_j\,\beta_\phi^2 + \frac{L_\phi}{2}\eta_j^2\sigma_\phi^2, \tag{90}$$

where in the unbiased case one can set $\beta_\phi = 0$.

**Step E.2.6 Telescope the descent to obtain a stationarity bound.** Assume a constant step size $\eta_j \equiv \eta \leq 1/L_\phi$ and sum (90) over $j = 0, \ldots, J-1$:

$$\frac{\eta}{4}\sum_{j=0}^{J-1}\mathbb{E}\big\|\nabla M_\phi(\xi^{(j)})\big\|^2 \leq \mathbb{E}\big[M_\phi(\xi^{(0)})\big] - \mathbb{E}\big[M_\phi(\xi^{(J)})\big] + J\eta\,\beta_\phi^2 + \frac{L_\phi}{2}J\eta^2\sigma_\phi^2.$$

Since $M_\phi(\cdot) \geq 0$ for our surrogates, we can drop $\mathbb{E}[M_\phi(\xi^{(J)})]$ and obtain

$$\frac{1}{J}\sum_{j=0}^{J-1}\mathbb{E}\big\|\nabla M_\phi(\xi^{(j)})\big\|^2 \leq \frac{4\,M_\phi(\xi^{(0)})}{\eta J} + 4\beta_\phi^2 + 2L_\phi\eta\,\sigma_\phi^2. \tag{91}$$

Consequently,

$$\min_{0 \leq j \leq J-1}\mathbb{E}\big\|\nabla M_\phi(\xi^{(j)})\big\|^2 \leq \frac{4\,M_\phi(\xi^{(0)})}{\eta J} + 4\beta_\phi^2 + 2L_\phi\eta\,\sigma_\phi^2. \tag{92}$$

We define the outer residual term

$$\epsilon_{\text{outer}}^{(\phi)}(J, \eta) := \frac{4\,M_\phi(\xi^{(0)})}{\eta J} + 4\beta_\phi^2 + 2L_\phi\eta\,\sigma_\phi^2, \tag{93}$$

and carry $\epsilon_{\text{outer}}^{(\phi)}(J, \eta)$ into the final utility boundary bound (absorbed into $\epsilon_{\text{bound}}$ or the method-dependent residual term).

**Interpretation of $\sigma_\phi^2$ and $\beta_\phi$ across branches.** **Distribution Matching (DM).** Mini-batched critics or feature networks induce stochastic variance $\sigma_\phi^2$ (and possibly a small bias $\beta_\phi$ under early stopping).

**Gradient Matching (GM).** A finite set of anchors or truncated paths yields a Monte Carlo estimator of the field-gap; anchor/path sampling and mini-batch noise contribute to $\sigma_\phi^2$.

**Trajectory Matching (TM).** Finite unrolls or implicit differentiation introduce both variance (mini-batches) and bias (truncation), fitting the biased estimator model with $(\sigma_\phi^2, \beta_\phi)$.

In summary, without assuming PL, the outer loop guarantees stationarity up to the residual $\epsilon_{\text{outer}}^{(\phi)}(J, \eta)$.

**E.3. Bridge Inequalities: Surrogate $\Rightarrow$ Alignment $\Delta_a$**

*Connection to matching.* Finally, each surrogate bounds the matching discrepancy:

$$\Delta_a(\hat{\mu}_\tau, \hat{\mu}_s) \leq \underbrace{\kappa_a L_{z,a}\, W_1 \text{ or } \kappa_a C_k\, \text{MMD}_k}_{\mathfrak{B}_{\text{DM}}},$$

$$\Delta_a(\hat{\mu}_\tau, \hat{\mu}_s) \leq \underbrace{\kappa_a |\Theta_j|\, \mathcal{M}_{\text{GM}}}_{\mathfrak{B}_{\text{GM}}}, \tag{94}$$

$$\Delta_a(\hat{\mu}_\tau, \hat{\mu}_s) \leq \underbrace{\frac{L_\theta + 2/\eta}{\omega_{\min}}\mathcal{M}_{\text{TM}} + 2L_\theta\varepsilon_{\text{path}}}_{\mathfrak{B}_{\text{TM}}},$$

where each $\mathfrak{B}$ represents the branch-specific upper bound on the same matching discrepancy, and other symbols are defined in Table 3, Appendix Section A. The displayed GM bound corresponds to the exact finite-anchor case $\Gamma_a = \Theta_j$; for an $\varepsilon$-net one adds the residual term $\kappa_a L_\theta \varepsilon$ as stated below. Putting together the contraction recursion in the above inequalities, we can say that DM, GM, and TM are not completely different heuristics, but three surrogates that consistently shrink $\Delta_a$ through the same bi-level dynamics. In addition, we can obtain:

**Proposition E.1** (DM bridge). *Assume for each $\theta \in \Gamma_a$ the vector map $z \mapsto g_a(\theta; z) \in \mathbb{R}^d$ is $L_{z,a}$–Lipschitz with respect to the data metric $d_Z$:*

$$\|g_a(\theta; z) - g_a(\theta; z')\|_2 \le L_{z,a} \, d_Z(z, z') \qquad (\forall z, z').$$

*Then, for empirical real and synthetic measures $\widehat{\mu}_\tau, \widehat{\mu}_s$,*

$$\Delta_a(\widehat{\mu}_\tau, \widehat{\mu}_s) \;\le\; \kappa_a L_{z,a} \, W_1(\widehat{\mu}_s, \widehat{\mu}_\tau).$$

*Moreover, if a bounded–kernel RKHS $(\mathcal{H}_k, \langle \cdot, \cdot \rangle_{\mathcal{H}_k})$ is used and, for each $\theta$, the coordinate functions $g_{a,j}(\theta; \cdot) \in \mathcal{H}_k$ satisfy*

$$\Big( \sum_{j=1}^d \|g_{a,j}(\theta; \cdot)\|_{\mathcal{H}_k}^2 \Big)^{1/2} \le C_k \qquad (\text{uniformly in } \theta),$$

*then*

$$\Delta_a(\widehat{\mu}_\tau, \widehat{\mu}_s) \;\le\; \kappa_a \, C_k \, \mathrm{MMD}_k(\widehat{\mu}_s, \widehat{\mu}_\tau).$$

*Proof.*

**Step E.1.0 Reduce to an un-preconditioned vector gap.** By definition and the operator-norm bound $\|P_a(\theta)\|_{\mathrm{op}} \le \kappa_a$,

$$\begin{aligned}
\Delta_a(\widehat{\mu}_\tau, \widehat{\mu}_s) &= \sup_{\theta \in \Gamma_a} \Big\| P_a(\theta) \Big( \mathbb{E}_{\widehat{\mu}_s} g_a(\theta; z) - \mathbb{E}_{\widehat{\mu}_\tau} g_a(\theta; z) \Big) \Big\|_2 \\
&\le \kappa_a \cdot \sup_{\theta \in \Gamma_a} \Big\| \mathbb{E}_{\widehat{\mu}_s} g_a(\theta; z) - \mathbb{E}_{\widehat{\mu}_\tau} g_a(\theta; z) \Big\|_2.
\end{aligned} \tag{95}$$

Hence it suffices to upper bound the $\ell_2$–norm of the vector expectation difference.

**Step E.1.1 Support-function identity for the Euclidean norm.** For any $v \in \mathbb{R}^d$, $\|v\|_2 = \sup_{\|u\|_2=1} \langle u, v \rangle$. Apply this with

$$v_\theta := \mathbb{E}_{\widehat{\mu}_s} g_a(\theta; z) - \mathbb{E}_{\widehat{\mu}_\tau} g_a(\theta; z).$$

Then

$$\|v_\theta\|_2 = \sup_{\|u\|_2=1} \Big\langle u, \, \mathbb{E}_{\widehat{\mu}_s} g_a(\theta; z) - \mathbb{E}_{\widehat{\mu}_\tau} g_a(\theta; z) \Big\rangle = \sup_{\|u\|_2=1} \Big( \mathbb{E}_{\widehat{\mu}_s} h_{\theta,u}(z) - \mathbb{E}_{\widehat{\mu}_\tau} h_{\theta,u}(z) \Big), \tag{96}$$

where we set the scalar function $h_{\theta,u}(z) := \langle u, g_a(\theta; z) \rangle$.

**Part E.1.A: $W_1$–bridge.**

**Step E.1.A.1 Scalar Lipschitz constant of $h_{\theta,u}$.** Given the $L_{z,a}$–Lipschitzness of $g_a(\theta; \cdot)$ and $\|u\|_2 = 1$,

$$|h_{\theta,u}(z) - h_{\theta,u}(z')| = |\langle u, g_a(\theta; z) - g_a(\theta; z') \rangle| \le \|u\|_2 \cdot \|g_a(\theta; z) - g_a(\theta; z')\|_2 \le L_{z,a} \, d_Z(z, z').$$

Thus $h_{\theta,u}$ is scalar $L_{z,a}$–Lipschitz on $(\mathcal{Z}, d_Z)$.

**Step E.1.A.2 Kantorovich–Rubinstein duality.** By the KR dual for $W_1$ (apply to the scalar $h_{\theta,u}$), for any probability measures $\mu, \nu$,

$$\big| \mathbb{E}_\mu h_{\theta,u} - \mathbb{E}_\nu h_{\theta,u} \big| \;\le\; \mathrm{Lip}(h_{\theta,u}) \cdot W_1(\mu, \nu) \;\le\; L_{z,a} \, W_1(\mu, \nu).$$

With $\mu = \widehat{\mu}_s, \nu = \widehat{\mu}_\tau$,

$$\big| \mathbb{E}_{\widehat{\mu}_s} h_{\theta,u} - \mathbb{E}_{\widehat{\mu}_\tau} h_{\theta,u} \big| \;\le\; L_{z,a} \, W_1(\widehat{\mu}_s, \widehat{\mu}_\tau). \tag{97}$$

**Step E.1.A.3 Take the $\sup_{\|u\|=1}$ and then $\sup_\theta$.** Combine Eq. (96)–Eq. (97):

$$\|v_\theta\|_2 = \sup_{\|u\|_2=1} \left( \mathbb{E}_{\widehat{\mu}_s} h_{\theta,u} - \mathbb{E}_{\widehat{\mu}_\tau} h_{\theta,u} \right) \le L_{z,a} \, W_1(\widehat{\mu}_s, \widehat{\mu}_\tau).$$

This bound is uniform in $\theta$, hence

$$\sup_{\theta \in \Gamma_a} \|v_\theta\|_2 \le L_{z,a} \, W_1(\widehat{\mu}_s, \widehat{\mu}_\tau).$$

Finally, plug into Eq. (95) to conclude the $W_1$–bridge:

$$\Delta_a(\widehat{\mu}_\tau, \widehat{\mu}_s) \le \kappa_a \, L_{z,a} \, W_1(\widehat{\mu}_s, \widehat{\mu}_\tau).$$

**Part E.1.B: MMD–bridge.**

**Step E.1.B.1 RKHS norm of the linear functional $h_{\theta,u}$.** Assume for each $\theta$ the coordinate functions $g_{a,j}(\theta; \cdot) \in \mathcal{H}_k$, and define the *aggregate* RKHS norm bound

$$C_k := \sup_{\theta \in \Gamma_a} \left( \sum_{j=1}^d \|g_{a,j}(\theta; \cdot)\|_{\mathcal{H}_k}^2 \right)^{1/2}.$$

For any $u \in \mathbb{R}^d$ with $\|u\|_2 = 1$, the scalar $h_{\theta,u}(z) = \sum_{j=1}^d u_j \, g_{a,j}(\theta; z)$ belongs to $\mathcal{H}_k$ by linearity, and the RKHS norm satisfies (by Cauchy–Schwarz in $\mathbb{R}^d$):

$$\|h_{\theta,u}\|_{\mathcal{H}_k} = \left\| \sum_{j=1}^d u_j \, g_{a,j}(\theta; \cdot) \right\|_{\mathcal{H}_k} \le \left( \sum_{j=1}^d u_j^2 \right)^{1/2} \left( \sum_{j=1}^d \|g_{a,j}(\theta; \cdot)\|_{\mathcal{H}_k}^2 \right)^{1/2} \le C_k. \tag{98}$$

**Step E.1.B.2 MMD dual formulation.** The RKHS (kernel $k$) dual inequality says that for any $f \in \mathcal{H}_k$,

$$\left| \mathbb{E}_\mu f - \mathbb{E}_\nu f \right| \le \|f\|_{\mathcal{H}_k} \, \mathrm{MMD}_k(\mu, \nu).$$

Apply this with $f = h_{\theta,u}$, together with Eq. (98):

$$\left| \mathbb{E}_{\widehat{\mu}_s} h_{\theta,u} - \mathbb{E}_{\widehat{\mu}_\tau} h_{\theta,u} \right| \le \|h_{\theta,u}\|_{\mathcal{H}_k} \, \mathrm{MMD}_k(\widehat{\mu}_s, \widehat{\mu}_\tau) \le C_k \, \mathrm{MMD}_k(\widehat{\mu}_s, \widehat{\mu}_\tau). \tag{99}$$

**Step E.1.B.3 Take the $\sup_{\|u\|=1}$ and then $\sup_\theta$.** As in Eq. (96),

$$\|v_\theta\|_2 = \sup_{\|u\|_2=1} \left( \mathbb{E}_{\widehat{\mu}_s} h_{\theta,u} - \mathbb{E}_{\widehat{\mu}_\tau} h_{\theta,u} \right) \le C_k \, \mathrm{MMD}_k(\widehat{\mu}_s, \widehat{\mu}_\tau),$$

uniformly in $\theta$. Taking $\sup_\theta$ and using Eq. (95) gives the MMD bridge:

$$\Delta_a(\widehat{\mu}_\tau, \widehat{\mu}_s) \le \kappa_a \, C_k \, \mathrm{MMD}_k(\widehat{\mu}_s, \widehat{\mu}_\tau).$$

$$\square$$

**Proposition E.2** (GM bridge ). *Define the anchor-averaged GM surrogate*

$$\mathcal{M}_{\mathrm{GM}}(\mu, \nu; \Theta_j) := \tfrac{1}{|\Theta_j|} \sum_{\theta \in \Theta_j} \left\| \mathbb{E}_\mu g_a(\theta; z) - \mathbb{E}_\nu g_a(\theta; z) \right\|_2,$$

*where $\Theta_j \subset \Gamma_a$ is a finite anchor set. Assume the unpreconditioned expected update gap $d(\theta) := \mathbb{E}_\mu g_a(\theta; z) - \mathbb{E}_\nu g_a(\theta; z)$ is $L_\theta$–Lipschitz on $(\Gamma_a, \|\cdot\|_2)$, and let $\Theta_j$ be an $\varepsilon$-net of $(\Gamma_a, \|\cdot\|_2)$, i.e., for every $\theta \in \Gamma_a$ there exists $\widehat{\theta} \in \Theta_j$ with $\|\theta - \widehat{\theta}\|_2 \le \varepsilon$. Then*

$$\Delta_a(\mu, \nu) \le \kappa_a \Big( |\Theta_j| \, \mathcal{M}_{\mathrm{GM}}(\mu, \nu; \Theta_j) + L_\theta \, \varepsilon \Big). \tag{100}$$

*In particular, if $\Gamma_a = \Theta_j$ (finite), then $\varepsilon = 0$ and $\Delta_a(\mu, \nu) \le \kappa_a \, |\Theta_j| \, \mathcal{M}_{\mathrm{GM}}(\mu, \nu; \Theta_j)$.*

*Proof.*

**Step E.2.1 Reduce to an un-preconditioned vector gap.** For any $\theta$ and any $v$, $\|P_a(\theta)v\|_2 \leq \|P_a(\theta)\|_{\mathrm{op}}\|v\|_2 \leq \kappa_a\|v\|_2$. Let $d(\theta) := \mathbb{E}_\mu g_a(\theta; z) - \mathbb{E}_\nu g_a(\theta; z) \in \mathbb{R}^d$. Then

$$\Delta_a(\mu, \nu) = \sup_{\theta \in \Gamma_a} \|P_a(\theta)\, d(\theta)\|_2 \ \leq \ \kappa_a \sup_{\theta \in \Gamma_a} \|d(\theta)\|_2. \tag{101}$$

**Step E.2.2 Parameter–Lipschitz transfer on $\Gamma_a$.** By the assumed Lipschitzness of the unpreconditioned gap $d(\theta)$, for all $\theta, \theta' \in \Gamma_a$,

$$\|d(\theta) - d(\theta')\|_2 \ \leq \ L_\theta \|\theta - \theta'\|_2. \tag{102}$$

**Step E.2.3 Covering argument with the $\varepsilon$-net $\Theta_j$.** For any $\theta \in \Gamma_a$, choose $\widehat{\theta} \in \Theta_j$ with $\|\theta - \widehat{\theta}\|_2 \leq \varepsilon$. Then

$$\|d(\theta)\|_2 \ \leq \ \|d(\widehat{\theta})\|_2 + \|d(\theta) - d(\widehat{\theta})\|_2 \ \leq \ \max_{\vartheta \in \Theta_j} \|d(\vartheta)\|_2 \ + \ L_\theta\, \varepsilon,$$

where the last inequality uses (102). Taking $\sup_{\theta \in \Gamma_a}$,

$$\sup_{\theta \in \Gamma_a} \|d(\theta)\|_2 \ \leq \ \max_{\vartheta \in \Theta_j} \|d(\vartheta)\|_2 \ + \ L_\theta\, \varepsilon. \tag{103}$$

**Step E.2.4 Relate max to the anchor average.** Since all terms are nonnegative,

$$\max_{\vartheta \in \Theta_j} \|d(\vartheta)\|_2 \ \leq \ \sum_{\vartheta \in \Theta_j} \|d(\vartheta)\|_2 \ = \ |\Theta_j|\, \mathcal{M}_{\mathrm{GM}}(\mu, \nu; \Theta_j).$$

**Step E.2.5 Combine all.** Plug (103) into (101), and use Step 4:

$$\Delta_a(\mu, \nu) \ \leq \ \kappa_a \Big( \max_{\vartheta \in \Theta_j} \|d(\vartheta)\|_2 + L_\theta\, \varepsilon \Big) \ \leq \ \kappa_a \Big( |\Theta_j|\, \mathcal{M}_{\mathrm{GM}}(\mu, \nu; \Theta_j) + L_\theta\, \varepsilon \Big),$$

which is (100). If $\Gamma_a = \Theta_j$, then $\varepsilon = 0$ and the last inequality reduces accordingly, which means:

$$\Delta_a(\mu, \nu) \ \leq \ \kappa_a |\Theta_j|\, \mathcal{M}_{\mathrm{GM}}(\mu, \nu; \Theta_j).$$

$\square$

**Proposition E.3** (TM bridge ). *Fix a configuration $a$. For $\mu \in \{\widehat{\mu}_s, \widehat{\mu}_\tau\}$ define*

$$F_\mu(\theta) \ := \ P_a(\theta)\, \mathbb{E}_\mu g_a(\theta; z) \in \mathbb{R}^d, \qquad \Delta_a(\widehat{\mu}_\tau, \widehat{\mu}_s) \ := \ \sup_{\theta \in \Gamma_a} \|F_{\widehat{\mu}_s}(\theta) - F_{\widehat{\mu}_\tau}(\theta)\|_2.$$

*Let the inner updates be*

$$\theta_{t+1}^{(\cdot, a)} \ = \ \Phi_a(\theta_t^{(\cdot, a)}; \mu) \ := \ \theta_t^{(\cdot, a)} - \eta\, F_\mu(\theta_t^{(\cdot, a)}), \qquad t = 0, 1, \ldots, L_b - 1,$$

*run from a shared initialization under the same configuration $a$ but with $\mu \in \{\widehat{\mu}_s, \widehat{\mu}_\tau\}$. Assume the* path–Lipschitz *condition holds along the unrolled path $\{\theta_t^{(s,a)}\}_{t=0}^{L_b} \cup \{\theta_t^{(\tau,a)}\}_{t=0}^{L_b}$:*

$$\|F_\mu(\theta) - F_\mu(\theta')\|_2 \ \leq \ L_\theta \|\theta - \theta'\|_2, \qquad \forall\, \theta, \theta' \text{ on the path, } \forall\, \mu \in \{\widehat{\mu}_s, \widehat{\mu}_\tau\}, \tag{104}$$

*where $L_\theta$ is the parameter–Lipschitz constant from Assumption 5.1 (restricted to the path). Define the TM surrogate*

$$\mathcal{M}_{\mathrm{TM}} \ := \ \sum_{t=0}^{L_b} \omega_t \big\|\theta_t^{(s,a)} - \theta_t^{(\tau,a)}\big\|_2, \qquad \omega_t > 0, \quad \omega_{\min} := \min_{0 \leq t \leq L_b} \omega_t > 0.$$

*Then the alignment discrepancy restricted to the unrolled path obeys*

$$\Delta_a^{\text{path}} := \sup_{\bar{\theta} \in \{\theta_t^{(s,a)}\} \cup \{\theta_t^{(\tau,a)}\}} \|F_{\widehat{\mu}_s}(\bar{\theta}) - F_{\widehat{\mu}_\tau}(\bar{\theta})\|_2 \leq \frac{L_\theta + 2/\eta}{\omega_{\min}} \, \mathcal{M}_{\text{TM}}. \tag{105}$$

*Moreover, if the unrolled path is an $\varepsilon_{\text{path}}$–cover of the maximizer set in $\Gamma_a$ (i.e., for every maximizer $\theta^\star$ in the definition of $\Delta_a$ there exists a path point $\bar{\theta}$ with $\|\theta^\star - \bar{\theta}\|_2 \leq \varepsilon_{\text{path}}$), then by Assumption 5.1*

$$\Delta_a(\widehat{\mu}_\tau, \widehat{\mu}_s) \leq \frac{L_\theta + 2/\eta}{\omega_{\min}} \, \mathcal{M}_{\text{TM}} + 2L_\theta \, \varepsilon_{\text{path}}. \tag{106}$$

*Here $F_\mu$ already includes the preconditioner $P_a$, so no additional $\kappa_a$ factor is introduced in the TM bridge.*

*Proof.*

**Step E.3.0.** From the two updates,

$$\Delta\theta_{t+1} = \theta_{t+1}^{(s,a)} - \theta_{t+1}^{(\tau,a)} = \Delta\theta_t - \eta\Big(F_{\widehat{\mu}_s}(\theta_t^{(s,a)}) - F_{\widehat{\mu}_\tau}(\theta_t^{(\tau,a)})\Big),$$

we obtain the algebraic identity

$$F_{\widehat{\mu}_s}(\theta_t^{(s,a)}) - F_{\widehat{\mu}_\tau}(\theta_t^{(\tau,a)}) = \frac{1}{\eta}\Big(\Delta\theta_t - \Delta\theta_{t+1}\Big), \qquad t = 0, \ldots, L_b - 1. \tag{107}$$

No smoothness or inequality is used here.

**Step E.3.1.A Point-wise control at a path point, first choice.** Fix any $t \in \{0, \ldots, L_b - 1\}$ and choose $\bar{\theta} = \theta_t^{(s,a)}$. By the triangle inequality,

$$\|F_{\widehat{\mu}_s}(\bar{\theta}) - F_{\widehat{\mu}_\tau}(\bar{\theta})\|_2 \leq \underbrace{\|F_{\widehat{\mu}_s}(\theta_t^{(s,a)}) - F_{\widehat{\mu}_\tau}(\theta_t^{(\tau,a)})\|_2}_{\text{link across the two runs at time } t} + \underbrace{\|F_{\widehat{\mu}_\tau}(\theta_t^{(\tau,a)}) - F_{\widehat{\mu}_\tau}(\theta_t^{(s,a)})\|_2}_{\text{same } \mu = \widehat{\mu}_\tau}. \tag{108}$$

The second term is controlled by the path–Lipschitz property:

$$\|F_{\widehat{\mu}_\tau}(\theta_t^{(\tau,a)}) - F_{\widehat{\mu}_\tau}(\theta_t^{(s,a)})\|_2 \leq L_\theta \|\theta_t^{(\tau,a)} - \theta_t^{(s,a)}\|_2 = L_\theta \|\Delta\theta_t\|_2.$$

For the first term in (108), invoke (107) and the triangle inequality:

$$\|F_{\widehat{\mu}_s}(\theta_t^{(s,a)}) - F_{\widehat{\mu}_\tau}(\theta_t^{(\tau,a)})\|_2 = \frac{1}{\eta}\|\Delta\theta_t - \Delta\theta_{t+1}\|_2 \leq \frac{1}{\eta}\big(\|\Delta\theta_t\|_2 + \|\Delta\theta_{t+1}\|_2\big).$$

Hence,

$$\|F_{\widehat{\mu}_s}(\theta_t^{(s,a)}) - F_{\widehat{\mu}_\tau}(\theta_t^{(s,a)})\|_2 \leq \Big(L_\theta + \frac{2}{\eta}\Big) \max\{\|\Delta\theta_t\|_2, \|\Delta\theta_{t+1}\|_2\}. \tag{109}$$

**Step E.3.1.B Point-wise control, second choice and edge $t = L_b$.** Choosing instead $\bar{\theta} = \theta_t^{(\tau,a)}$ and repeating the same argument (swap $s$ and $\tau$ in (108), use (107) again) yields the identical bound

$$\|F_{\widehat{\mu}_s}(\theta_t^{(\tau,a)}) - F_{\widehat{\mu}_\tau}(\theta_t^{(\tau,a)})\|_2 \leq \Big(L_\theta + \frac{2}{\eta}\Big) \max\{\|\Delta\theta_t\|_2, \|\Delta\theta_{t+1}\|_2\} \qquad (t = 0, \ldots, L_b - 1). \tag{110}$$

For the terminal points $t = L_b$, apply (109) with index $t - 1$:

$$\|F_{\widehat{\mu}_s}(\theta_{L_b}^{(s,a)}) - F_{\widehat{\mu}_\tau}(\theta_{L_b}^{(s,a)})\|_2 \leq \Big(L_\theta + \frac{2}{\eta}\Big) \max\{\|\Delta\theta_{L_b-1}\|_2, \|\Delta\theta_{L_b}\|_2\},$$

and similarly for $\theta_{L_b}^{(\tau,a)}$. Thus the same form holds at $t = L_b$ after relabeling the pair $(t, t+1)$ as $(L_b - 1, L_b)$.

**Step E.3.2 Supremum over the unrolled path.** Collecting (109)–(110) (including the terminal case), we obtain

$$\Delta_a^{\text{path}} := \sup_{\bar{\theta} \in \{\theta_t^{(s,a)}\} \cup \{\theta_t^{(\tau,a)}\}} \|F_{\widehat{\mu}_s}(\bar{\theta}) - F_{\widehat{\mu}_\tau}(\bar{\theta})\|_2 \leq \left(L_\theta + \frac{2}{\eta}\right) \max_{0 \leq t \leq L_b} \|\Delta\theta_t\|_2. \tag{111}$$

**Step E.3.3 From $\max$ to the weighted TM surrogate.** Using $\sum_{t=0}^{L_b} \omega_t \|\Delta\theta_t\|_2 \geq \omega_{\min} \max_t \|\Delta\theta_t\|_2$, we have

$$\max_{0 \leq t \leq L_b} \|\Delta\theta_t\|_2 \leq \frac{1}{\omega_{\min}} \left(\sum_{t=0}^{L_b} \omega_t \|\Delta\theta_t\|_2\right) = \frac{1}{\omega_{\min}} \mathcal{M}_{\text{TM}}. \tag{112}$$

Substituting (112) into (111) yields the *path–restricted* TM bridge

$$\Delta_a^{\text{path}} \leq \frac{L_\theta + 2/\eta}{\omega_{\min}} \mathcal{M}_{\text{TM}}. \tag{113}$$

**Step E.3.4 From the path supremum to the global discrepancy via path coverage.** Let $\theta^\star \in \Gamma_a$ be a maximizer (or $\epsilon$–maximizer) of the sup defining $\Delta_a$. By the $\varepsilon_{\text{path}}$–cover assumption, choose a path point $\bar{\theta}$ with $\|\theta^\star - \bar{\theta}\|_2 \leq \varepsilon_{\text{path}}$. Then, using the parameter–Lipschitz property for each $\mu$ and the triangle inequality,

$$\begin{aligned} \|F_{\widehat{\mu}_s}(\theta^\star) - F_{\widehat{\mu}_\tau}(\theta^\star)\|_2 &\leq \|F_{\widehat{\mu}_s}(\bar{\theta}) - F_{\widehat{\mu}_\tau}(\bar{\theta})\|_2 + \|F_{\widehat{\mu}_s}(\theta^\star) - F_{\widehat{\mu}_s}(\bar{\theta})\|_2 + \|F_{\widehat{\mu}_\tau}(\bar{\theta}) - F_{\widehat{\mu}_\tau}(\theta^\star)\|_2 \\ &\leq \|F_{\widehat{\mu}_s}(\bar{\theta}) - F_{\widehat{\mu}_\tau}(\bar{\theta})\|_2 + L_\theta \|\theta^\star - \bar{\theta}\|_2 + L_\theta \|\theta^\star - \bar{\theta}\|_2 \\ &\leq \Delta_a^{\text{path}} + 2L_\theta \varepsilon_{\text{path}}. \end{aligned}$$

Taking sup over maximizers (or letting $\epsilon \downarrow 0$) yields

$$\Delta_a(\widehat{\mu}_\tau, \widehat{\mu}_s) \leq \Delta_a^{\text{path}} + 2L_\theta \varepsilon_{\text{path}}. \tag{114}$$

Combining (105) and (114) gives the desired global TM bridge

$$\Delta_a(\widehat{\mu}_\tau, \widehat{\mu}_s) \leq \frac{L_\theta + 2/\eta}{\omega_{\min}} \mathcal{M}_{\text{TM}} + 2L_\theta \varepsilon_{\text{path}}.$$

$$\square$$

## E.4. Exchangeability of TM/GM/DM up to constants (Lemma E.4)

**Lemma E.4** (Exchangeability up to constants(informal).). *Under standard smoothness/Lipschitz and contractive inner-loop conditions in Assumption 4.1, the inequality in Eq. (94) yields bounds in the following relations:*

$$\mathfrak{B}_{\text{TM}} = O(\mathfrak{B}_{\text{GM}}) \quad \text{and} \quad \mathfrak{B}_{\text{GM}} = O(\mathfrak{B}_{\text{DM}}),$$

*and $\Delta_a \leq \min\{\mathfrak{B}_{\text{DM}}, \mathfrak{B}_{\text{GM}}, \mathfrak{B}_{\text{TM}}\}$ with each consistently shrinking the same matching discrepancy through the bi-level dynamics.*

Lemma E.4 shows that the bounds in Eq. (94) are quantitatively interchangeable: controlling the trajectory surrogate (TM) automatically controls the gradient surrogate (GM), which in turn is controlled by the distribution surrogate (DM). This hierarchy explains why switching between matching methods is an effective way to reduce matching discrepancy: once DM (or GM) saturates, further progress can be made by GM (or TM) without altering the fundamental dependence on $k$ or $n$.

**Lemma E.5** (Precise exchangeability of bridge bounds (Formal)). *Fix configuration $a$. Assume: (i) $z \mapsto g_a(\theta; z)$ is $L_{z,a}$–Lipschitz for all $\theta \in \Gamma_a$; (ii) along the trained path $\{\theta_t\}_{t=0}^{L_b}$, the map $\theta \mapsto \mathbb{E}_\nu g_a(\theta; z)$ is $L_\theta$–Lipschitz uniformly for $\nu \in \{\widehat{\mu}_s, \widehat{\mu}_\tau\}$; (iii) the preconditioner satisfies $\|P_a(\theta)\|_{\text{op}} \leq \kappa_a$; (iv) the inner update is uniformly $\bar{\rho}$–contractive in expectation along the path, for some $\bar{\rho} < 1$, and trajectory weights satisfy $0 < \omega_{\min} \leq \omega_t \leq \omega_{\max} < \infty$. Let $\varepsilon_{\text{path}}$ denote the discretization error from sampling $\Theta_j$.*

*Define the bridge bounds as in Eq. (5):*

$$\mathfrak{B}_{\text{DM}} \in \left\{ \kappa_a L_{z,a} W_1(\widehat{\mu}_s, \widehat{\mu}_\tau), \ \kappa_a C_k \, \text{MMD}_k(\widehat{\mu}_s, \widehat{\mu}_\tau) \right\},$$

$$\mathfrak{B}_{\mathrm{GM}} := \kappa_a |\Theta_j| \, \mathcal{M}_{\mathrm{GM}},$$

$$\mathfrak{B}_{\mathrm{TM}} := \frac{L_\theta + 2/\eta}{\omega_{\min}} \, \mathcal{M}_{\mathrm{TM}} + 2L_\theta \, \varepsilon_{\mathrm{path}}.$$

*Then there exist finite constants $C_1, C_2 > 0$, depending only on $(\bar{\rho}, L_\theta, \eta_{\min}, \eta_{\max}, \omega_{\min}, \omega_{\max})$, such that*

$$\mathfrak{B}_{\mathrm{TM}} \leq C_1 \, \mathfrak{B}_{\mathrm{GM}} + 2L_\theta \varepsilon_{\mathrm{path}}, \qquad \mathfrak{B}_{\mathrm{GM}} \leq C_2 \, \mathfrak{B}_{\mathrm{DM}}. \tag{115}$$

*Proof.*

**Step E.4.1 From the TM bridge to a bound in terms of $\sum_t \|\delta_t\|$.** By the TM bridge (Proposition E.3),

$$\mathfrak{B}_{\mathrm{TM}} = \frac{L_\theta + 2/\eta}{\omega_{\min}} \, \mathcal{M}_{\mathrm{TM}} + 2L_\theta \, \varepsilon_{\mathrm{path}}, \qquad \mathcal{M}_{\mathrm{TM}} := \sum_{t=0}^{L_b} \omega_t \|\Delta_t\|.$$

One step of the two inner updates and the contractivity assumption give the exact split

$$\Delta_{t+1} = \Phi_{\widehat{\mu}_\tau}(\theta_t^{(s)}; \eta_t) - \Phi_{\widehat{\mu}_\tau}(\theta_t^{(\tau)}; \eta_t) - \eta_t \, \delta_t \quad \Rightarrow \quad \|\Delta_{t+1}\| \leq \bar{\rho} \|\Delta_t\| + \eta_t \|\delta_t\|.$$

Assuming the shared initialization $\Delta_0 = 0$, the recursion implies $\sum_t \|\Delta_t\| \leq \frac{\eta_{\max}}{1-\bar{\rho}} \sum_t \|\delta_t\|$. Using $\omega_t \leq \omega_{\max}$,

$$\mathcal{M}_{\mathrm{TM}} = \sum_t \omega_t \|\Delta_t\| \leq \omega_{\max} \sum_t \|\Delta_t\| \leq \frac{\omega_{\max} \eta_{\max}}{1-\bar{\rho}} \sum_t \|\delta_t\|.$$

Plugging back into the TM bridge yields

$$\mathfrak{B}_{\mathrm{TM}} \leq \underbrace{\frac{(L_\theta + 2/\eta) \, \omega_{\max} \eta_{\max}}{(1-\bar{\rho}) \, \omega_{\min}}}_{=: C_1} \sum_{t=0}^{L_b - 1} \|\delta_t\| + 2L_\theta \, \varepsilon_{\mathrm{path}}. \tag{116}$$

**Step E.4.2 Aligning $\sum_t \|\delta_t\|$ with the GM bridge.** By definition, $\delta_t = P_a(\theta_t^{(s)}) \, d(\theta_t^{(s)})$, hence

$$\sum_t \|\delta_t\| = \sum_t \|P_a(\theta_t^{(s)}) \, d(\theta_t^{(s)})\| \leq \sum_t \kappa_a \|d(\theta_t^{(s)})\| = \kappa_a \, |\Theta_j| \, \mathcal{M}_{\mathrm{GM}}.$$

Since $\mathfrak{B}_{\mathrm{GM}} = \kappa_a |\Theta_j| \, \mathcal{M}_{\mathrm{GM}}$, we have the clean comparison

$$\sum_t \|\delta_t\| \leq \mathfrak{B}_{\mathrm{GM}}. \tag{117}$$

Combining (116) and (117) we obtain

$$\mathfrak{B}_{\mathrm{TM}} \leq C_1 \, \mathfrak{B}_{\mathrm{GM}} + 2L_\theta \, \varepsilon_{\mathrm{path}}, \qquad C_1 := \frac{(L_\theta + 2/\eta) \, \omega_{\max} \eta_{\max}}{(1-\bar{\rho}) \, \omega_{\min}}. \tag{118}$$

**Step E.4.3 Comparing the GM and DM bridges (detailed).** Fix an anchor $\theta \in \Theta_j$. The GM residual is $d(\theta) = \mathbb{E}_{\widehat{\mu}_s} g_a(\theta; z) - \mathbb{E}_{\widehat{\mu}_\tau} g_a(\theta; z)$.

*(Wasserstein–1 case).* Reduce the vector norm to scalar test functions via the support function:

$$\|d(\theta)\|_2 = \sup_{\|v\|_2 \leq 1} \langle v, \, \mathbb{E}_{\widehat{\mu}_s} g_a(\theta; z) - \mathbb{E}_{\widehat{\mu}_\tau} g_a(\theta; z) \rangle$$

$$= \sup_{\|v\|_2 \leq 1} \left( \mathbb{E}_{\widehat{\mu}_s} \phi_{v,\theta} - \mathbb{E}_{\widehat{\mu}_\tau} \phi_{v,\theta} \right), \quad \phi_{v,\theta}(z) := \langle v, g_a(\theta; z) \rangle.$$

If $z \mapsto g_a(\theta; z)$ is $L_{z,a}$–Lipschitz uniformly in $\theta$, then $\phi_{v,\theta}$ is also $L_{z,a}$–Lipschitz for every $\|v\| \leq 1$. By Kantorovich–Rubinstein duality,

$$\|d(\theta)\|_2 \leq L_{z,a} W_1(\widehat{\mu}_s, \widehat{\mu}_\tau).$$

*(MMD case).* Assume either (i) a *vector-valued RKHS* model $g_a(\theta; z) \in \mathcal{H}_k^d$ with $\|g_a(\theta; z)\|_{\mathcal{H}_k^d} \leq C_k$ uniformly. Let $\mu \mapsto m_k(\mu) := \mathbb{E}_\mu[\varphi_k(z)]$ be the kernel mean embedding in $\mathcal{H}_k$. By the reproducing property (or its vector-valued analogue),

$$\|d(\theta)\|_2 = \|\mathbb{E}_{\widehat{\mu}_s} h_\theta - \mathbb{E}_{\widehat{\mu}_\tau} h_\theta\|_2 \leq \|g_a(\theta; z)\|_{\mathcal{H}_k^d} \|m_k(\widehat{\mu}_s) - m_k(\widehat{\mu}_\tau)\|_{\mathcal{H}_k} \leq C_k \,\mathrm{MMD}_k(\widehat{\mu}_s, \widehat{\mu}_\tau).$$

*(Averaging over anchors).* The right-hand sides of E.4–E.4 do not depend on $\theta$, so averaging preserves the bound:

$$\mathcal{M}_{\mathrm{GM}} = \frac{1}{|\Theta_j|} \sum_{\theta \in \Theta_j} \|d(\theta)\| \leq \begin{cases} L_{z,a} W_1(\widehat{\mu}_s, \widehat{\mu}_\tau), \\ C_k \,\mathrm{MMD}_k(\widehat{\mu}_s, \widehat{\mu}_\tau). \end{cases}$$

Multiplying by $\kappa_a |\Theta_j|$ and recalling the GM and DM bridge definitions,

$$\mathfrak{B}_{\mathrm{GM}} = \kappa_a |\Theta_j| \mathcal{M}_{\mathrm{GM}} \leq |\Theta_j| \cdot \underbrace{\kappa_a \begin{cases} L_{z,a} W_1(\widehat{\mu}_s, \widehat{\mu}_\tau), \\ C_k \,\mathrm{MMD}_k(\widehat{\mu}_s, \widehat{\mu}_\tau), \end{cases}}_{= \,\mathfrak{B}_{\mathrm{DM}}} =: C_2 \,\mathfrak{B}_{\mathrm{DM}}, \quad C_2 = |\Theta_j|. \tag{119}$$

**Step E.4.4 Conclusion.** Equations (118) and (119) give exactly

$$\mathfrak{B}_{\mathrm{TM}} \leq C_1 \,\mathfrak{B}_{\mathrm{GM}} + 2L_\theta \,\varepsilon_{\mathrm{path}}, \qquad \mathfrak{B}_{\mathrm{GM}} \leq C_2 \,\mathfrak{B}_{\mathrm{DM}},$$

with $C_1 := \frac{(L_\theta + 2/\eta)\,\omega_{\max} \eta_{\max}}{(1 - \bar{\rho})\,\omega_{\min}}$ and $C_2 := |\Theta_j|$. $\qquad \square$

## E.5. Unified Single-Configuration Risk Bound (Theorem 6.2)

**Theorem E.6** (Dynamic single-configuration risk bound for matching-based distillation (Formal))**.** *Fix a configuration $a$ and run $J$ outer iterations using a surrogate $\mathcal{M}_\phi$ with $\phi \in \{\mathrm{DM}(W_1), \mathrm{DM}(\mathrm{MMD}), \mathrm{GM}, \mathrm{TM}\}$. Suppose Assumption 4.1 holds (Lipschitz risk with constant $L_R$, path-Lipschitz field with constant $L_\theta$, preconditioner bound $\|P_a(\theta)\|_{op} \leq \kappa_a$, trajectory-local stable dynamics with factors $\{\rho_{a,t}\}_{t=0}^{T-1} \subset (0, 1)$ and stepsizes $\eta_t \in [\eta_{\min}, \eta_{\max}]$), and suppose the TM weights satisfy $\omega_t \geq \omega_{\min} > 0$. Then, for any $T \geq 0$, with probability at least $1 - \varepsilon$,*

$$\begin{aligned} \left| \widehat{R}(\theta_T^{(s,a)}) - \widehat{R}(\theta_T^{(\tau,a)}) \right| \leq\ & L_R \Big( \prod_{t=0}^{T-1} \rho_{a,t} \Big) \|\delta_0\| + e_{\mathrm{te}}(m, \varepsilon) \\ & + \eta_{\max} L_R C_{2,a}^{\mathrm{traj}} \Big[ C_{\phi,a} \Big( \mathcal{M}_\phi(\xi^{(J)}) + \epsilon_{\mathrm{outer}}^{(\phi)}(J, \eta) \Big) + e_{\mathrm{tr}}^{(\phi)}(k, n, \varepsilon) \Big] \\ & + \mathbf{1}\{\phi = \mathrm{TM}\} \cdot 2\eta_{\max} L_R L_\theta C_{2,a}^{\mathrm{traj}} \varepsilon_{\mathrm{path}}, \end{aligned} \tag{120}$$

*where the branch constants are*

$$C_{\phi,a} := \begin{cases} \kappa_a L_{z,a}, & \phi = \mathrm{DM}(W_1), \\ \kappa_a C_k, & \phi = \mathrm{DM}(\mathrm{MMD}), \\ \kappa_a |\Theta_j|, & \phi = \mathrm{GM}, \\ \dfrac{L_\theta + 2/\eta}{\omega_{\min}}, & \phi = \mathrm{TM}, \end{cases}$$

*and the training-side concentration terms satisfy*

$$\begin{aligned} e_{\mathrm{tr}}^{(\mathrm{GM})}, \ e_{\mathrm{tr}}^{(\mathrm{MMD})} &= \tilde{O}\Big( \tfrac{1}{\sqrt{k}} + \tfrac{1}{\sqrt{n}} \Big), \\ e_{\mathrm{tr}}^{(W_1)} &= \tilde{O}\big( k^{-1/d} + n^{-1/d} \big) \quad \textit{for data metric space of (effective) dimension } d, \\ e_{\mathrm{tr}}^{(\mathrm{TM})} &= \tilde{O}\Big( \frac{(L_\theta + 2/\eta)}{\omega_{\min}} \cdot \frac{L_b \eta}{1 - \bar{\rho}_a} \cdot \Big( \tfrac{1}{\sqrt{k}} + \tfrac{1}{\sqrt{n}} \Big) \Big). \end{aligned}$$

*Here $\bar{\rho}_a := \max_{0 \leq t < T} \rho_{a,t}$ is used only to simplify this rate expression. The test-side concentration is $e_{\mathrm{te}}(m, \varepsilon) = O\big(\sqrt{\log(1/\varepsilon)/m}\big)$.*

*Proof of Theorem 6.2.*

**Step E.5.1 Risk gap $\Rightarrow$ parameter gap (Lipschitz risk).** By risk Lipschitzness,

$$|R_\nu(\theta) - R_\nu(\theta')| = \big|\mathbb{E}_\nu[\ell(\theta; z) - \ell(\theta'; z)]\big| \leq \mathbb{E}_\nu L_R \|\theta - \theta'\| \leq L_R \|\theta - \theta'\|.$$

Therefore,

$$\big|R_\nu(\theta_T^{(s,a)}) - R_\nu(\theta_T^{(\tau,a)})\big| \leq L_R \|\delta_T\|. \tag{121}$$

**Step E.5.2 One-step recursion for $\delta_{t+1}$ (contractivity + two-sample drift).** Add and subtract the same-measure map:

$$\delta_{t+1} = \Phi_a(\theta_t^{(s,a)}; \hat{\mu}_s) - \Phi_a(\theta_t^{(\tau,a)}; \hat{\mu}_\tau)$$
$$= \underbrace{\Phi_a(\theta_t^{(s,a)}; \hat{\mu}_\tau) - \Phi_a(\theta_t^{(\tau,a)}; \hat{\mu}_\tau)}_{\text{same measure}} - \eta_t\Big(F_{\hat{\mu}_s}(\theta_t^{(s,a)}) - F_{\hat{\mu}_\tau}(\theta_t^{(s,a)})\Big).$$

By Assumption 4.1(iii), along the realized trajectories, there exist $\{\rho_{a,t}\}_{t=0}^{T-1}$ with $\rho_{a,t} \in (0,1)$ such that

$$\|\delta_T\| \leq \Big(\prod_{t=0}^{T-1} \rho_{a,t}\Big)\|\delta_0\| + \Delta_a(\hat{\mu}_\tau, \hat{\mu}_s) \sum_{t=0}^{T-1} \eta_t \prod_{s=t+1}^{T-1} \rho_{a,s} \leq \Big(\prod_{t=0}^{T-1} \rho_{a,t}\Big)\|\delta_0\| + \eta_{\max} C_{2,a}^{\mathrm{traj}} \Delta_a(\hat{\mu}_\tau, \hat{\mu}_s), \tag{122}$$

where

$$C_{2,a}^{\mathrm{traj}} := \sum_{t=0}^{T-1} \prod_{s=t+1}^{T-1} \rho_{a,s}.$$

**Step E.5.3 Replace the drift by $\Delta_a$.** By definition of $\Delta_a$,

$$\sup_\theta \big\|F_{\hat{\mu}_s}(\theta) - F_{\hat{\mu}_\tau}(\theta)\big\| = \sup_\theta \big\|P_a(\theta)(\mathbb{E}_{\hat{\mu}_s} g - \mathbb{E}_{\hat{\mu}_\tau} g)\big\| \leq \Delta_a(\hat{\mu}_\tau, \hat{\mu}_s).$$

Combining with (122) and then with (121),

$$|R_\nu(\theta_T^{(s,a)}) - R_\nu(\theta_T^{(\tau,a)})| \leq L_R \Big(\prod_{t=0}^{T-1} \rho_{a,t}\Big)\|\delta_0\| + L_R \eta_{\max} C_{2,a}^{\mathrm{traj}} \Delta_a(\hat{\mu}_\tau, \hat{\mu}_s). \tag{123}$$

**Step E.5.4 Bridge $\Delta_a$ to the surrogate $\mathcal{M}_\phi$ (branch choice).** We now invoke the three bridge propositions given earlier:

*(DM bridge)*: If $z \mapsto g_a(\theta; z)$ is $L_{z,a}$–Lipschitz, for $W_1$, for any $\theta$, $\big\|\mathbb{E}_{\hat{\mu}_s} g_a(\theta) - \mathbb{E}_{\hat{\mu}_\tau} g_a(\theta)\big\| \leq L_{z,a} W_1(\hat{\mu}_s, \hat{\mu}_\tau)$; hence

$$\Delta_a(\hat{\mu}_\tau, \hat{\mu}_s) \leq \kappa_a L_{z,a} W_1(\hat{\mu}_s, \hat{\mu}_\tau),$$

and

$$\Delta_a(\hat{\mu}_\tau, \hat{\mu}_s) \leq \kappa_a C_k \mathrm{MMD}_k(\hat{\mu}_s, \hat{\mu}_\tau).$$

*(GM bridge)*: For anchors $\Theta_j \subset \Gamma_a$ forming an $\varepsilon$–net and the anchor-averaged surrogate $\mathcal{M}_{\mathrm{GM}}$,

$$\Delta_a(\mu, \nu) \leq \kappa_a\big(|\Theta_j| \mathcal{M}_{\mathrm{GM}}(\mu, \nu; \Theta_j) + L_\theta \varepsilon\big).$$

In particular, if $\Gamma_a = \Theta_j$ (finite), $\varepsilon = 0$ and $\Delta_a(\mu, \nu) \leq \kappa_a |\Theta_j| \mathcal{M}_{\mathrm{GM}}$.

*(TM bridge)*: With weights $\omega_t \geq \omega_{\min} > 0$,

$$\Delta_a(\hat{\mu}_\tau, \hat{\mu}_s) \leq \frac{L_\theta + 2/\eta}{\omega_{\min}} \mathcal{M}_{\mathrm{TM}} + 2L_\theta \varepsilon_{\mathrm{path}}.$$

Summarizing, there exist branch-specific constants $C_{\phi,a}$ such that

$$\Delta_a(\hat{\mu}_\tau, \hat{\mu}_s) \;\le\; C_{\phi,a}\, \mathcal{M}_\phi \;+\; \mathbf{1}\{\phi = \mathrm{TM}\}\, 2L_\theta\, \varepsilon_{\mathrm{path}},$$

$$C_{\phi,a} = \begin{cases} \kappa_a L_{z,a} \text{ or } \kappa_a C_k, & \phi = W_1,\ \mathrm{MMD}, \\ \kappa_a |\Theta_j|, & \phi = \mathrm{GM}, \\ (L_\theta + 2/\eta)/\omega_{\min}, & \phi = \mathrm{TM}, \end{cases} \tag{124}$$

The factor $\eta_{\max} L_R C_{2,a}^{\mathrm{traj}}$ is applied only after this bridge is inserted into the risk bound.

**Step E.5.5 Outer-loop progress (no PL): keep $\mathcal{M}_\phi(\xi^{(J)})$ and add a residual term.** Running $J$ outer iterations with step size $\eta \le 1/L_\phi$ yields an $\epsilon_{\mathrm{outer}}^{(\phi)}(J, \eta)$-stationary iterate in the sense that

$$\min_{0 \le j \le J-1} \mathbb{E}\|\nabla \mathcal{M}_\phi(\xi^{(j)})\|^2 \le \epsilon_{\mathrm{outer}}^{(\phi)}(J, \eta), \qquad \epsilon_{\mathrm{outer}}^{(\phi)}(J, \eta) = \frac{4\, \mathcal{M}_\phi(\xi^{(0)})}{\eta J} + 4\beta_\phi^2 + 2L_\phi \eta \sigma_\phi^2.$$

In the risk bound we keep the attained surrogate value $\mathcal{M}_\phi(\xi^{(J)})$ explicitly and absorb the outer optimization imperfection into $\epsilon_{\mathrm{outer}}^{(\phi)}(J, \eta)$.

**Step E.5.6 Finite-sample penalties for each branch.** We now upper bound the *training-side* error $e_{\mathrm{tr}}^{(\phi)}(k, n, \varepsilon)$ that enters when replacing population quantities by empirical ones.

*(DM: $W_1$).* For empirical $\hat{\mu}_n$ of size $n$ from a distribution on $\mathbb{R}^d$ (with mild moment conditions), the Wasserstein-1 convergence rate is

$$\mathbb{E}W_1(\mu, \hat{\mu}_n) \;=\; \begin{cases} O(n^{-1/2}), & d = 1, \\ O(n^{-1/2} \log n), & d = 2, \\ O(n^{-1/d}), & d \ge 3, \end{cases}$$

with high-probability analogues. Hence, by the triangle inequality, $W_1(\hat{\mu}_s, \hat{\mu}_\tau) \le W_1(\hat{\mu}_s, \mu) + W_1(\mu, \hat{\mu}_\tau)$ yields

$$e_{\mathrm{tr}}^{(W_1)}(k, n, \varepsilon) = \tilde{O}\big(k^{-1/d} + n^{-1/d}\big)$$

(or faster under low-dimensional/covering assumptions).

*(DM: MMD).* For bounded kernels, the empirical MMD concentrates at a CLT rate: $\big|\mathrm{MMD}_k(\hat{\mu}, \hat{\nu}) - \mathrm{MMD}_k(\mu, \nu)\big| = O_\P(1/\sqrt{m_\mu} + 1/\sqrt{m_\nu})$. Thus $e_{\mathrm{tr}}^{(\mathrm{MMD})} = \tilde{O}(1/\sqrt{k} + 1/\sqrt{n})$.

*(GM).* Define the scalar class $\mathcal{F} := \{\, z \mapsto \langle v, g_a(\theta; z)\rangle \,:\, \theta \in \Gamma_a,\ \|v\|_2 \le 1 \,\}$. Then $\sup_\theta \|\mathbb{E}g_a(\theta) - \mathbb{E}_{\hat{\mu}} g_a(\theta)\| = \sup_{f \in \mathcal{F}} (\mathbb{E}f - \mathbb{E}_{\hat{\mu}} f)$. By symmetrization and Rademacher complexity plus Ledoux–Talagrand contraction, under boundedness/Lipschitz envelopes we get

$$\sup_\theta \big\|\mathbb{E}g_a(\theta) - \mathbb{E}_{\hat{\mu}} g_a(\theta)\big\| = \tilde{O}\big(1/\sqrt{m}\big).$$

Applying this to both synthetic and real samples and averaging over anchors gives $e_{\mathrm{tr}}^{(\mathrm{GM})} = \tilde{O}(1/\sqrt{k} + 1/\sqrt{n})$.

*(TM).* The TM bridge (Proposition "TM bridge") plus the one-step recursion yields

$$\mathcal{M}_{\mathrm{TM}} \;\le\; \frac{\kappa_a}{(1 - \bar{\rho}_a)\, \omega_{\min}} \sum_{t=0}^{L_b - 1} \eta_t \big\|\mathbb{E}_{\hat{\mu}_s} g_a(\theta_t^{(s)}) - \mathbb{E}_{\hat{\mu}_\tau} g_a(\theta_t^{(s)})\big\| + \frac{L_\theta}{1 - \bar{\rho}_a}\, \varepsilon_{\mathrm{path}},$$

where $\bar{\rho}_a := \max_t \rho_{a,t}$. The same Rademacher argument as for GM applied at each $\theta_t^{(s)}$ gives a CLT rate scaled by the schedule factor $S_a := \frac{(L_\theta + 2/\bar{\eta})}{\omega_{\min}} \cdot \frac{\sum_t \eta_t}{1 - \bar{\rho}_a}$:

$$e_{\mathrm{tr}}^{(\mathrm{TM})}(k, n, \varepsilon) = \tilde{O}\Big(S_a\Big(\tfrac{1}{\sqrt{k}} + \tfrac{1}{\sqrt{n}}\Big)\Big).$$

**Step E.5.7 Test-side concentration.** For the empirical test risk over $m$ samples, $e_{\mathrm{te}}(m, \varepsilon) = O\big(\sqrt{\log(1/\varepsilon)/m}\big)$.

**Step E.5.8: Assemble all pieces.** Insert (124) into (123), and add the training-side and test-side penalties from Steps E.5.6–E.5.7:

$$\left| R_\nu(\theta_T^{(s,a)}) - R_\nu(\theta_T^{(\tau,a)}) \right| \le L_R \Big( \prod_{t=0}^{T-1} \rho_{a,t} \Big) \|\delta_0\| + e_{\mathrm{te}}(m, \varepsilon)$$

$$+ L_R \, \eta_{\max} \, C_{2,a}^{\mathrm{traj}} \Big[ C_{\phi,a} \Big( \mathcal{M}_\phi(\xi^{(J)}) + \epsilon_{\mathrm{outer}}^{(\phi)}(J, \eta) \Big) + e_{\mathrm{tr}}^{(\phi)}(k, n, \varepsilon) \Big] \tag{125}$$

$$+ \mathbf{1}\{\phi = \mathrm{TM}\} \, 2 L_R \, \eta_{\max} \, C_{2,a}^{\mathrm{traj}} \cdot (L_\theta \, \varepsilon_{\mathrm{path}}). \tag{126}$$

This matches the theorem statement, with $C_{\phi,a}$ carrying only the branch-specific bridge constant and $\eta_{\max} L_R C_{2,a}^{\mathrm{traj}}$ applied after the bridge is inserted into the risk recursion. $\qquad \square$

## E.6. Coverage-Aware Bound with Dynamic Outer Progress (Theorem 6.3)

**Theorem E.7** (Coverage-aware bound with dynamic outer progress (formal)). *Let $(\mathcal{A}, d_\mathcal{A})$ be the target configuration family. Fix a radius $r > 0$ and let $\mathcal{N}_r = \{a_1, \ldots, a_N\} \subset \mathcal{A}$ be an $r$-net (i.e., an $r$-packing that also $r$-covers $\mathcal{A}$; if only an $r$-packing is given, replace $r$ by $2r$ via the standard packing→covering conversion). For each center $a_j \in \mathcal{N}_r$, run $J$ outer steps with branch $\phi \in \{\mathrm{DM}(W_1), \mathrm{DM}(\mathrm{MMD}), \mathrm{GM}, \mathrm{TM}\}$ and surrogate $\mathcal{M}_\phi$, under Assumption 4.1 uniformly over $a \in \mathcal{A}$ and the cross-configuration Lipschitz transfer Assumption 5.1:*

$$\|F_{\mu,a}(\theta) - F_{\mu,a'}(\theta)\| \le C_{\mathrm{trans}} \, d_\mathcal{A}(a, a') \qquad (\forall \, \theta, \, \mu \in \{\widehat{\mu}_s, \widehat{\mu}_\tau\}).$$

*Define the uniform constants*

$$C_{2,\max}^{\mathrm{traj}} := \sup_{a \in \mathcal{A}} C_{2,a}^{\mathrm{traj}}, \quad C_{\phi,\max} := \sup_{a \in \mathcal{A}} C_{\phi,a}, \quad \epsilon_{\mathrm{outer},\max}^{(\phi)}(J, \eta) := \sup_{a \in \mathcal{A}} \epsilon_{\mathrm{outer}}^{(\phi)}(J, \eta; a),$$

*where the bridge constants are*

$$C_{\phi,a} = \begin{cases} \kappa_a L_{z,a} \text{ or } \kappa_a C_k, & \phi = \mathrm{DM}(W_1), \, \mathrm{DM}(\mathrm{MMD}), \\ \kappa_a |\Theta_j|, & \phi = \mathrm{GM}, \\ (L_{\theta,a} + 2/\eta)/\omega_{\min}, & \phi = \mathrm{TM}. \end{cases}$$

*Then, with probability at least $1 - \varepsilon$,*

$$\sup_{a \in \mathcal{A}} \left| R_\nu(\theta_T^{(s,a)}) - R_\nu(\theta_T^{(\tau,a)}) \right| \le \underbrace{\sup_{a \in \mathcal{A}} L_R(a) \Big( \prod_{t=0}^{T-1} \rho_{a,t} \Big) \|\delta_0\|}_{\text{inner residual}} + \underbrace{e_{\mathrm{te}}(m, \varepsilon)}_{\text{test}}$$

$$+ \underbrace{2 \eta_{\max} L_{R,\max} C_{\mathrm{trans}} \, r \, C_{2,\max}^{\mathrm{traj}}}_{\text{coverage transfer}} + \mathbf{1}\{\phi = \mathrm{TM}\} \underbrace{2 \eta_{\max} L_{R,\max} L_{\theta,\max} C_{2,\max}^{\mathrm{traj}} \, \varepsilon_{\mathrm{path}}}_{\text{TM discretization}}$$

$$+ \eta_{\max} L_{R,\max} C_{2,\max}^{\mathrm{traj}} \Big[ C_{\phi,\max} \Big( \mathcal{M}_{\phi,\max}^{(J)} + \epsilon_{\mathrm{outer},\max}^{(\phi)}(J, \eta) \Big) + \widetilde{e}_{\mathrm{tr}}^{(\phi)}(k, n, \varepsilon, N) \Big], \tag{127}$$

*where*

$$\mathcal{M}_{\phi,\max}^{(J)} := \max_{1 \le j \le N} \mathcal{M}_\phi(\xi^{(J)}; a_j), \qquad L_{\theta,\max} := \sup_a L_{\theta,a}, \quad L_{R,\max} := \sup_a L_R(a),$$

*and*

$$\widetilde{e}_{\mathrm{tr}}^{(\phi)}(k, n, \varepsilon, N) = \begin{cases} \tilde{O}(k^{-1/d} + n^{-1/d}), & \phi = \mathrm{DM}(W_1) \\ \hat{O}\Big( \big( \frac{1}{\sqrt{k}} + \frac{1}{\sqrt{n}} \big) \sqrt{\log N + \log(1/\varepsilon)} \Big), & \phi = \mathrm{DM}(\mathrm{MMD}), \, \mathrm{GM}, \, \mathrm{TM}, \end{cases}$$

*and $N := |\mathcal{N}_r|$ is the covering number at scale $r$. The test-side term satisfies $e_{\mathrm{te}}(m, \varepsilon) = O\big( \sqrt{\log(1/\varepsilon)/m} \big)$.*

*Proof.*

**Step E.6.1 Alignment transfer from $a$ to its center $a_j$.** Fix $a \in \mathcal{A}$ and take $a_j \in \mathcal{N}_r$ with $d_{\mathcal{A}}(a, a_j) \leq r$. By the triangle inequality, for any $\theta$,

$$\|F_{\hat{\mu}_s,a}(\theta) - F_{\hat{\mu}_\tau,a}(\theta)\| \leq \|F_{\hat{\mu}_s,a}(\theta) - F_{\hat{\mu}_s,a_j}(\theta)\| + \|F_{\hat{\mu}_s,a_j}(\theta) - F_{\hat{\mu}_\tau,a_j}(\theta)\| + \|F_{\hat{\mu}_\tau,a_j}(\theta) - F_{\hat{\mu}_\tau,a}(\theta)\|$$
$$\leq 2C_{\text{trans}} d_{\mathcal{A}}(a, a_j) + \|F_{\hat{\mu}_s,a_j}(\theta) - F_{\hat{\mu}_\tau,a_j}(\theta)\|. \tag{128}$$

Taking the supremum over $\theta \in \Gamma_a$ yields

$$\Delta_a(\hat{\mu}_\tau, \hat{\mu}_s) \leq \Delta_{a_j}(\hat{\mu}_\tau, \hat{\mu}_s) + 2C_{\text{trans}} r. \tag{129}$$

**Step E.6.2 Risk–alignment reduction at $a$.** For any fixed configuration $a$, the single-configuration reduction gives

$$\left|R_\nu(\theta_T^{(s,a)}) - R_\nu(\theta_T^{(\tau,a)})\right| \leq L_R(a)\Big(\prod_{t=0}^{T-1} \rho_{a,t}\Big)\|\delta_0\| + \eta_{\max} L_R(a) C_{2,a}^{\text{traj}} \Delta_a(\hat{\mu}_\tau, \hat{\mu}_s) + e_{\text{te}}(m, \varepsilon). \tag{130}$$

**Step E.6.3 Plug the transfer bound into the risk inequality.** Combine (129) and (130):

$$\left|R_\nu(\theta_T^{(s,a)}) - R_\nu(\theta_T^{(\tau,a)})\right| \leq L_R(a)\Big(\prod_{t=0}^{T-1} \rho_{a,t}\Big)\|\delta_0\| + \eta_{\max} L_R(a) C_{2,a}^{\text{traj}}\Big[\Delta_{a_j}(\hat{\mu}_\tau, \hat{\mu}_s) + 2C_{\text{trans}}r\Big] + e_{\text{te}}(m, \varepsilon). \tag{131}$$

**Step E.6.4 Bridge and outer progress at the trained centers.** By construction, we *train* only at the centers $a_j$. At each $a_j$, the branch-specific bridge (DM/GM/TM) yields

$$\Delta_{a_j}(\hat{\mu}_\tau, \hat{\mu}_s) \leq C_{\phi,a_j} \mathcal{M}_\phi(\xi^{(J)}; a_j) + \mathbf{1}\{\phi = \text{TM}\} 2L_{\theta,a_j} \varepsilon_{\text{path}}. \tag{132}$$

Recall $C_{\phi,a_j}$ equals $\kappa_{a_j} L_{z,a_j}$ or $\kappa_{a_j} C_k$ (DM), $\kappa_{a_j}|\Theta_j|$ (GM), and $(L_{\theta,a_j} + 2/\eta)/\omega_{\min}$ (TM).

**Step E.6.5 Uniformize constants and take sup over $a \in \mathcal{A}$.** Define the worst-case constants over $\mathcal{A}$:

$$C_{2,\max}^{\text{traj}} := \sup_{a \in \mathcal{A}} C_{2,a}^{\text{traj}}, \quad C_{\phi,\max} := \sup_{a \in \mathcal{A}} C_{\phi,a}, \quad \kappa_{\max} := \sup_a \kappa_a, \quad L_{\theta,\max} := \sup_a L_{\theta,a}, \quad L_{R,\max} := \sup_a L_R(a).$$

Let

$$\mathcal{M}_{\phi,\max}^{(J)} := \max_{1 \leq j \leq N} \mathcal{M}_\phi(\xi^{(J)}; a_j), \qquad N := |\mathcal{N}_r|.$$

Using (132) in (131) gives

$$\left|R_\nu(\theta_T^{(s,a)}) - R_\nu(\theta_T^{(\tau,a)})\right| \leq \underbrace{L_R(a)\Big(\prod_{t=0}^{T-1} \rho_{a,t}\Big)\|\delta_0\|}_{\text{inner residual}} + e_{\text{te}}(m, \varepsilon) + 2\eta_{\max} L_{R,\max} C_{\text{trans}} r C_{2,\max}^{\text{traj}}$$
$$+ \eta_{\max} L_{R,\max} C_{2,\max}^{\text{traj}}\Big[C_{\phi,\max} \mathcal{M}_{\phi,\max}^{(J)}\Big]$$
$$+ \mathbf{1}\{\phi = \text{TM}\} 2\eta_{\max} L_{R,\max} L_{\theta,\max} C_{2,\max}^{\text{traj}} \varepsilon_{\text{path}}. \tag{133}$$

Taking the supremum in $a \in \mathcal{A}$ replaces $L_R(a)\prod_t \rho_{a,t}$ by $\sup_{a \in \mathcal{A}} L_R(a)\prod_t \rho_{a,t}$ on the first term, leaving the rest unchanged.

**Step E.6.6 Coverage-aware training-side concentration over $N$ centers.** We now upgrade the training-side surrogate estimation to hold *uniformly* over the $N$ trained centers. This produces the coverage-aware term $\tilde{e}_{\text{tr}}^{(\phi)}(k, n, \varepsilon, N)$ stated in the theorem.

*DM($W_1$).* For empirical measures in $\mathbb{R}^d$, nonasymptotic bounds give $W_1(\mu, \hat{\mu}_m) = O_{\P}(m^{-1/d})$ for $d \geq 3$, $O_{\P}(m^{-1/2} \log m)$ for $d = 2$, and $O_{\P}(m^{-1/2})$ for $d = 1$. A union bound over $N$ centers multiplies failure probability by $N$; in the $\tilde{O}(\cdot)$ notation (suppressing polylog factors), we retain the geometric-rate term:

$$\tilde{e}_{\text{tr}}^{(W_1)}(k, n, \varepsilon, N) = \tilde{O}(k^{-1/d} + n^{-1/d}).$$

---

**Algorithm 1** Single-configuration evaluation

---

**Input:** dataset $\mathcal{D}$; distilled sets $\{\mathcal{S}_k\}$ for budgets $k$; source configuration $a_0$.
**Output:** points $\{(k, \Delta_{a_0}(k))\}$ and a linear fit of $\Delta$ vs. $1/\sqrt{k}$.

**for each** budget $k$ **do**
    Load distilled set $\mathcal{S}_k$.
    **for** repeat $r = 1, \ldots, R$ **do**
        Initialize student $\theta \sim a_0$.
        Train $\theta$ on $\mathcal{S}_k$ using the student protocol of $a_0$.
        Evaluate $\mathrm{Acc}_{\mathrm{syn}}(k, r)$ on the test split of $\mathcal{D}$.
    **end for**
    Train a real-data baseline once under $a_0$ to obtain $\mathrm{Acc}_{\mathrm{real}}$.
    Set $\Delta_{a_0}(k) = \mathrm{Acc}_{\mathrm{real}} - \mathrm{mean}_r \, \mathrm{Acc}_{\mathrm{syn}}(k, r)$.
**end for**
Fit a line $y = ax + b$ with $x = 1/\sqrt{k}$ and $y = \Delta_{a_0}(k)$; report slope/intercept/$R^2$.

---

*DM(MMD) & GM.* For bounded kernels, $\mathrm{MMD}_k$ concentrates at CLT rate; for GM, let $\mathcal{F} := \{z \mapsto \langle v, g_a(\theta; z) \rangle : \|v\| \leq 1, \theta \in \Gamma\}$ and apply symmetrization + Rademacher complexity with Ledoux–Talagrand's contraction to obtain $O_{\P}(1/\sqrt{m})$. A union bound over $N$ centers contributes a $\sqrt{\log N + \log(1/\varepsilon)}$ factor:

$$\widetilde{e}_{\mathrm{tr}}^{(\mathrm{MMD})}, \, \widetilde{e}_{\mathrm{tr}}^{(\mathrm{GM})} = \tilde{O}\Big(\big(\tfrac{1}{\sqrt{k}} + \tfrac{1}{\sqrt{n}}\big)\sqrt{\log N + \log(1/\varepsilon)}\Big).$$

*TM.* The TM bridge plus the one-step recursion shows that the TM surrogate aggregates $L_b$ CLT-scale deviations, scaled by the schedule factor

$$S_a = \frac{(L_{\theta,a} + 2/\bar{\eta}_a)}{\omega_{\min,a}} \cdot \frac{\sum_t \eta_{t,a}}{1 - \bar{\rho}_a}, \qquad \bar{\rho}_a := \max_t \rho_{a,t}.$$

Uniformizing over $a \in \mathcal{A}$ (and thus over centers) and applying the same union bound gives

$$\widetilde{e}_{\mathrm{tr}}^{(\mathrm{TM})} = \tilde{O}\Big(S_{\max}\big(\tfrac{1}{\sqrt{k}} + \tfrac{1}{\sqrt{n}}\big)\sqrt{\log N + \log(1/\varepsilon)}\Big), \qquad S_{\max} := \sup_{a \in \mathcal{A}} S_a.$$

**Step E.6.7 Assemble and rename constants.** Collect the inner residual into $\sup_{a \in \mathcal{A}} L_R(a)(\prod_{t=0}^{T-1} \rho_{a,t})\|\delta_0\|$, keep $e_{\mathrm{te}}(m, \varepsilon)$ unchanged, Plugging the center-wise bridge (132) and the coverage-aware training terms into (133) yields the stated bound, where the method-dependent term uses $\mathcal{M}_{\phi,\max}^{(J)}$ and the outer residual is captured by $\epsilon_{\mathrm{outer},\max}^{(\phi)}(J, \eta)$ in the theorem statement. $\qquad\square$

## F. Experiments

### F.1. Detailed Experimental Setup

**Datasets and preprocessing.** We conduct experiments on **MNIST** (LeCun et al., 2002), **CIFAR-10/100** (Krizhevsky et al., 2009), and **ImageNette** (Deng et al., 2009), and additionally include **Tiny-ImageNet** (Le & Yang, 2015) and **ImageNet-1K** (Deng et al., 2009) to test robustness at larger scales. For MNIST/CIFAR/ImageNette, we use the standard official train/test splits. For each dataset, we strictly follow the preprocessing pipeline prescribed by the corresponding baseline implementation (e.g., dataset-specific normalization; ZCA whitening when enabled by the baseline) to ensure strict comparability and to avoid introducing confounding factors.

**Baselines and distilled data generation.** We evaluate four canonical distillation paradigms: *Gradient Matching* (DC (Zhao et al., 2020), DSA (Zhao & Bilen, 2021)), *Distribution Matching* (DM (Zhao & Bilen, 2023)), *Trajectory Matching* (MTT (Cazenavette et al., 2022)), and a recent *diffusion-based* method (MGD$^3$ (Chan-Santiago et al., 2025)). For DC/DSA/DM/MTT and MGD$^3$, we generate distilled datasets using the authors' official open-source repositories and default training schedules/hyperparameters; we do not modify method-specific losses, augmentations, or stopping criteria. The

---

**Algorithm 2** Configuration coverage: from per-configuration curves to the coverage law (with estimated $\mathcal{H}_{\text{cov}}$)

---

**Input:** dataset $\mathcal{D}$; distilled sets $\{\mathcal{S}_k\}$; candidate configurations $A \subset \mathcal{C}$ (sampled by a balanced design).

**Output:** coverage points $\{(X, Y)\}$ with $X = \sqrt{\widehat{\mathcal{H}_{\text{cov}}}(A_m, r)}/\sqrt{k}$ and $Y = \Delta(k, m)$; a global fit.

**for each** budget $k$ **do**
    **for each** configuration $a \in A$ **do**
        Train a student $\theta_a$ on $\mathcal{S}_k$ under $a$ and record $\text{Acc}_{\text{syn}}(k, a)$.
        Obtain once-per-configuration real baseline $\text{Acc}_{\text{real}}(a)$ (precomputed).
        Set $\Delta(k, a) = \text{Acc}_{\text{real}}(a) - \text{Acc}_{\text{syn}}(k, a)$.
    **end for**
**end for**
**for** $m = 1, \ldots, |A|$ **do**
    Choose a size-$m$ subset of configurations and denote it $A_m$ (prefix or random).
    Compute $\widehat{\mathcal{H}_{\text{cov}}}(A_m, r)$ via ESTIMATEHCOV$(A_m)$ (below).
    **for each** $k$ **do**
        Set $Y = \Delta(k, m) = \frac{1}{m} \sum_{a \in A_m} \Delta(k, a)$.
        Set $X = \sqrt{\widehat{\mathcal{H}_{\text{cov}}}(A_m, r)}/\sqrt{k}$.
        Append $(X, Y)$ to the coverage set.
    **end for**
**end for**
Fit a single line $Y = \alpha X + \beta$ over all coverage points; report slope/intercept/$R^2$.

**Procedure** ESTIMATEHCOV$(A_m)$:
    Fix $S$ random initializations $\{\theta_0^{(s)}\}$ and $B$ mini-batches $\{\mathcal{B}_b\}$; $\varepsilon = 10^{-8}$.
    **for each** $a \in A_m$ **do**
        Compute normalized one-step updates $u(a; s, b) = \frac{P_a g_a}{\|P_a g_a\|_2 + \varepsilon}$ for all $(s, b)$.
        Compute noise scale $\sigma_a = \text{median}_{s,b} \|u(a; s, b) - \bar{u}(a)\|_2$.
    **end for**
    Set radius $r = c \cdot \text{median}_{a \in A_m} \sigma_a$ with fixed $c \in [1, 2]$.
    Define $d_{\mathcal{A}}(a, a') = \frac{1}{SB} \sum_{s,b} \|u(a; s, b) - u(a'; s, b)\|_2$.
    Run farthest-first greedy $r$–cover under $d_{\mathcal{A}}$ to obtain $\widehat{N}_r$; return $\widehat{\mathcal{H}_{\text{cov}}}(A_m, r) = \log \widehat{N}_r$.

---

only controlled variable across these methods is the distillation budget $k$ (specified via IPC). For additional strong baselines (TESLA (Cui et al., 2023), NCFM (Wang et al., 2025), DATM (Guo et al., 2023), SRe2L (Jiang et al., 2025)), as well as for large-scale settings on Tiny-ImageNet and ImageNet-1K, we directly extract the reported results from the original papers when their IPC settings are available, since the corresponding pipelines are not re-run in our framework.

**Source configuration (distillation).** For all methods that we re-run (DC/DSA/DM/MTT and MGD³), distillation is performed in a fixed *source configuration* using `ConvNet+SGD` (following the default choices in the respective repositories), with DSA enabled whenever applicable by the baseline implementation. Each distillation run is executed to completion under the default schedules provided by the authors.

**Target configurations and training protocol.** A target configuration $a$ is defined as a triplet *(architecture × optimizer × augmentation)*. Target architectures are sampled from a ConvNet-family design space that factorizes architecture into: (i) depth $D \in \{1, 2, 3, 4\}$, (ii) width $W \in \{32, 64, 128, 256\}$, (iii) activation $\in \{\texttt{sigmoid}, \texttt{relu}, \texttt{leakyrelu}, \texttt{swish}\}$, and (iv) normalization $\in \{\texttt{layernorm}, \texttt{instancenorm}, \texttt{groupnorm}\}$, together with the optimizer choice (`SGD` or `Adam`) and an augmentation switch (DSA on/off). For every target configuration, we train for **300 epochs** with **batch size 256** and **initial learning rate 0.01**, and we do **not** use early stopping. Unless overridden by a baseline repository (for methods we re-run), we use a standard SGD setting with momentum 0.9 and weight decay $5 \times 10^{-4}$, and Adam with default betas. All reported accuracies are evaluated on the standard test set, and we compute the configuration-wise generalization discrepancy

$$\Delta_a(\hat{\mu}_\tau, \hat{\mu}_s) = \left| \hat{R}(\theta_T^{(s,a)}) - \hat{R}(\theta_T^{(\tau,a)}) \right|.$$

**Distillation budget (IPC).** Whenever the baseline repository supports sweeping IPC, we vary IPC $\in$ $\{2, 4, 6, 8, 12, 18, 28, 51, 100, 200\}$, yielding a total distilled size $k = \text{IPC} \times C$ where $C$ is the number of classes. For baselines extracted from papers (e.g., TESLA/NCFM/DATM/SRe2L and large-scale Tiny-ImageNet/ImageNet-1K settings), we report the IPC values available in the original publications, which are typically sparse.

**Randomness and repetitions.** To account for randomness in both distillation and target training, we repeat each experiment **three times** with random seeds $\{0, 1, 2\}$, and report the mean performance across runs (with variability shown in plots/tables where applicable).

**Coverage complexity estimation.** We estimate the coverage complexity $\mathcal{H}_{\text{cov}}(\mathcal{A}, r)$ using the update-induced metric and the greedy $r$-cover procedure defined in Eq. (8); pseudo-code is provided in Alg. 2 in Appendix F.

**Compute.** All experiments are run on a machine equipped with $4\times$ **RTX 4090 GPUs**. We do not emphasize wall-clock comparisons, as our focus is on generalization behavior and configuration-wise coverage rather than efficiency.

**Evaluation metric.** We report the generalization error

$$\Delta \;=\; \big|\hat{R}(\theta_T^{(\hat{\mu}_s, a)}) - \hat{R}(\theta_T^{(\hat{\mu}_\tau, a)})\big|,$$

the accuracy gap between training on distilled and real data within the same configuration $a$. Each result is averaged over 5 independent repeats; for single-configuration runs we regress $\Delta$ against $1/\sqrt{k}$ and report slope, intercept, and $R^2$.

**Target configuration training protocol.** Students are trained from scratch with batch size 256, strictly following the DC/DSA evaluation protocol. When DSA is enabled, students are trained for 1000 epochs; otherwise, for 300 epochs. SGD uses an initial learning rate of 0.01 with momentum 0.9 and weight decay $5 \times 10^{-4}$, decayed $\times 0.1$ midway through training. Adam uses its default settings. Architectures (`ConvNet`, `LeNet`, `ResNet-18`, `AlexNet`) all follow this protocol.

**Coverage-law construction.** To test the predicted scaling with coverage complexity, we aggregate multiple configurations. For each subset of size $m$, and each IPC $k$, we compute the averaged gap

$$Y \;=\; \Delta(k, m), \quad X \;=\; \frac{\sqrt{\log m}}{\sqrt{k}}.$$

We then regress $Y$ against $X$ to test the coverage law $Y \propto X$. Two subset strategies are used: `prefix` (deterministic) and `random` (averaged over $T{=}5$ trials). For each configuration included, the *real-data baseline* is trained once on the full dataset with the identical optimizer, epochs, and augmentation as in the distilled run; this baseline is reused across $k$.

**Hardware and software.** Experiments are conducted on servers with AMD EPYC 7642 CPUs (96 vCPUs), CUDA 12.4, and up to $4\times$NVIDIA RTX 4090 GPUs.

### F.2. Algorithms

For clarity, we briefly summarize the two evaluation protocols. In the *single-configuration evaluation* (Algorithm 1), we fix a source configuration and train students on distilled datasets of varying budget $k$. Each student is evaluated against its real-data counterpart in the same configuration, and the resulting accuracy gaps $\Delta(k)$ are regressed against $1/\sqrt{k}$ to reveal the single-configuration scaling law.

**Coverage-law evaluation with estimated coverage entropy.** Algorithm 2 turns per-configuration learning curves into a single *coverage law* that isolates how configurational diversity penalizes performance at a fixed distillation budget. Let $A \subset \mathcal{C}$ be a candidate configuration set obtained via a balanced sampling design. For each budget $k$ and configuration $a \in A$, we train a student on the distilled set $\mathcal{S}_k$ under $a$ and record $Acc_{syn}(k, a)$, while $Acc(a)$ is a once-per-configuration baseline computed on the real dataset. We define the *per-configuration gap*

$$\Delta(k, a) \;=\; Acc_{real}(a) - Acc_{syn}(k, a). \tag{134}$$

To aggregate across multiple targets, for each $m = 1, \ldots, |A|$ we select a size-$m$ subset $A_m \subseteq A$ (either as a prefix of a fixed ordering or by uniform random sampling) and define the *average gap*

$$\Delta(k, m) = \frac{1}{m} \sum_{a \in A_m} \Delta(k, a). \tag{135}$$

**Estimating configurational diversity via coverage entropy.**  Rather than using $m$ as a proxy for diversity, we estimate an intrinsic complexity of $A_m$ through an $r$–*covering* in the space of (normalized) one-step update directions. Fix $S$ random initializations $\{\theta_0^{(s)}\}_{s=1}^S$ and $B$ mini-batches $\{\mathcal{B}_b\}_{b=1}^B$, and let $\varepsilon > 0$ be a small constant. For each configuration $a \in A_m$, we compute normalized one-step updates

$$u(a; s, b) = \frac{P_a g_a}{\|P_a g_a\|_2 + \varepsilon}, \tag{136}$$

where $g_a$ denotes the gradient evaluated at $(\theta_0^{(s)}, \mathcal{B}_b)$ under configuration $a$, and $P_a$ is the configuration-induced projection/operator defining the update geometry. We estimate a configuration-specific noise scale by

$$\sigma_a = \mathrm{median}_{s,b} \left\| u(a; s, b) - \bar{u}(a) \right\|_2, \tag{137}$$

where $\bar{u}(a)$ is the average of $u(a; s, b)$ over $(s, b)$. We set the covering radius as

$$r = c \cdot \mathrm{median}_{a \in A_m} \sigma_a, \qquad c \in [1, 2] \text{ fixed}, \tag{138}$$

and define the induced distance between configurations

$$d_{\mathcal{A}}(a, a') = \frac{1}{SB} \sum_{s,b} \left\| u(a; s, b) - u(a'; s, b) \right\|_2. \tag{139}$$

Running a farthest-first greedy $r$–cover under $d_{\mathcal{A}}$ yields an estimate $\widehat{N}_r$ of the number of balls of radius $r$ needed to cover $A_m$. We return the *estimated coverage entropy*

$$\widehat{\mathcal{H}_{\mathrm{cov}}}(A_m, r) = \log \widehat{N}_r. \tag{140}$$

**From coverage entropy to the coverage law.**  For each pair $(k, m)$ we create one coverage point $(X, Y)$ with

$$Y = \Delta(k, m), \qquad X = \frac{\sqrt{\widehat{\mathcal{H}_{\mathrm{cov}}}(A_m, r)}}{\sqrt{k}}. \tag{141}$$

Pooling all points over budgets and subset sizes, we fit a single linear model

$$Y = \alpha X + \beta \tag{142}$$

and report $(\alpha, \beta)$ and $R^2$. Under the proposed coverage law, $\alpha$ quantifies the marginal penalty induced by configurational diversity (as measured by $\widehat{\mathcal{H}_{\mathrm{cov}}}$), while the $1/\sqrt{k}$ dependence captures the budget-controlled reduction in the average performance gap.

### F.3. Single-Configuration Regime

We report three complementary statistics to assess the empirical $1/\sqrt{k}$ scaling: (i) *single-line OLS* (slope with 95% CI and $R^2$) to quantify approximate linearity; (ii) *rank-based monotonicity* via Spearman $\rho$ and $p$-values to confirm that the error consistently decreases with $1/\sqrt{k}$ even when deviations from strict linearity exist; and (iii) *broken-line (piecewise) fits* with an estimated breakpoint and left/right slopes, selected by information criteria (AIC/BIC), to detect two-phase behavior (e.g., early rapid gains and later saturation). **Identifiability note:** piecewise regression requires a sufficiently dense IPC grid. For several baselines (TESLA, NCFM, DATM, SRe2L) we strictly follow the original papers that typically evaluate only three IPC values; in this regime the breakpoint model is statistically underdetermined and AIC/BIC comparisons can become ill-defined. Hence, for these methods the primary evidence is Spearman monotonicity and single-line OLS, while piecewise conclusions are emphasized only for methods with enough IPC points (DC/DSA/DM/MTT and MGD[3] on ImageNette).

*Table 4.* CIFAR-10: regression diagnostics for the $1/\sqrt{k}$ scaling.

| Method | OLS slope ($\pm$95% CI) | $R^2$ | Spearman $\rho$ ($p$) | Break $x$ | Left slope | Right slope | $\text{AIC}_{\text{single}} \to \text{AIC}_{\text{piece}}$ | Piecewise (BIC) |
|---|---|---|---|---|---|---|---|---|
| DC | $-0.28\,[-0.34, -0.22]$ | 0.94 | $-0.93\,(p = 2.4 \times 10^{-4})$ | 0.26 | $-0.06$ | $-0.24$ | $-74.1 \to -79.0$ | True |
| DSA | $-0.41\,[-0.49, -0.33]$ | 0.95 | $-1.00\,(p \approx 0)$ | 0.32 | $-0.63$ | $-0.27$ | $-76.2 \to -109.9$ | True |
| DM | $-0.49\,[-0.65, -0.32]$ | 0.85 | $-1.00\,(p \approx 0)$ | 0.32 | $-0.95$ | $-0.19$ | $-61.4 \to -92.9$ | True |
| MTT | $-0.24\,[-0.35, -0.12]$ | 0.82 | $-1.00\,(p \approx 0)$ | 0.38 | $-0.32$ | $-0.03$ | $-57.9 \to -64.9$ | True |
| TESLA | $-0.28\,[-0.47, -0.08]$ | 1.00 | $-1.00\,(p = 0.00)$ | – | – | – | – | – |
| NCFM | $-0.33\,[-0.34, -0.31]$ | 1.00 | $-1.00\,(p = 0.00)$ | – | – | – | – | – |
| DATM | $-0.38\,[-0.53, -0.24]$ | 0.96 | $-1.00\,(p = 1.40 \times 10^{-2})$ | – | – | – | – | – |

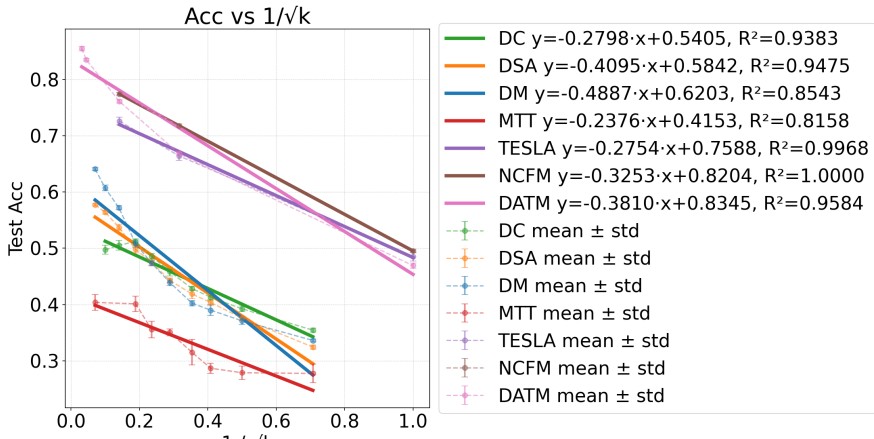

*Figure 6.* Single-configuration scaling law on CIFAR10.

**CIFAR-10.** As results shown in Tab. 4 and Fig. 6, all methods exhibit strong monotonic scaling, with Spearman $\rho \leq -0.93$ and most achieving $\rho = -1$, indicating a consistent decrease of $\Delta_a$ as $1/\sqrt{k}$ decreases. For **DC**, the single-line fit is stable ($R^2 = 0.94$) with a moderate negative slope ($-0.28$ with a reasonably tight CI), while piecewise selection (BIC) suggests a mild two-phase pattern: the left segment is nearly flat ($-0.06$) and the right segment is steeper ($-0.24$), consistent with a short warm-up region followed by a clearer scaling regime. **DSA** shows both high linearity ($R^2 = 0.95$) and the strongest evidence for two-phase behavior (AIC improves substantially), with a steep early slope ($-0.63$) transitioning to a gentler slope ($-0.27$) after the breakpoint, indicating pronounced diminishing returns as IPC increases. **DM** has the steepest overall OLS slope among the matching-based baselines ($-0.49$) but a lower $R^2$ (0.85), which is explained by an even sharper early descent (left slope $-0.95$) followed by strong saturation (right slope $-0.19$). Thus, DM is sample-efficient at small IPC but exhibits faster diminishing returns. **MTT** yields the weakest single-line $R^2$ (0.82) despite perfect monotonicity ($\rho = -1$), and the piecewise fit reveals an explicit plateau: the right slope is close to zero ($-0.03$), suggesting that additional distilled samples bring little benefit beyond the breakpoint, aligning with instability/trajectory-mismatch effects. For **TESLA/NCFM/DATM**, the single-line fits are already near-perfect ($R^2 \approx 0.96$–$1.00$ and $\rho \approx -1$), but piecewise inference is not supported (or not identifiable under sparse IPC points). Overall, CIFAR-10 confirms universal monotonic $1/\sqrt{k}$ scaling, while different methods differ markedly in *efficiency* (slope magnitude) and *saturation behavior* (piecewise right-slope).

**CIFAR-100.** As results shown in Tab. 5 and Fig. 7, on this harder dataset, monotonicity remains extremely strong ($\rho \approx -1$ for all methods), but deviations from a single global line become more pronounced for some matching-based approaches. **DC** remains highly linear ($R^2 = 0.97$) with a relatively mild slope ($-0.20$), and the piecewise fit suggests a short initial slow region (left slope $-0.04$) followed by a clearer scaling phase (right slope $-0.22$). This indicates robust and stable improvement as IPC increases. **DSA** maintains a strong fit ($R^2 = 0.92$) with a steeper slope ($-0.32$) and clear diminishing returns (left slope $-0.52$ to right slope $-0.22$), implying higher sample-efficiency than DC at small IPC but more saturation later. **DM** remains the most sample-efficient in slope magnitude ($-0.43$) yet shows reduced single-line linearity ($R^2 = 0.84$) explained by a dramatic early drop (left slope $-1.41$) and a much milder late slope ($-0.27$). This reflects a strong "early-win" regime and a harder long-tail regime on CIFAR-100. **MTT** exhibits the largest deviation from single-line linearity

*Table 5.* CIFAR-100: regression diagnostics for the $1/\sqrt{k}$ scaling.

| Method | OLS slope (±95% CI) | $R^2$ | Spearman $\rho$ ($p$) | Break $x$ | Left slope | Right slope | AIC$_{\text{single}}$ →AIC$_{\text{piece}}$ | Piecewise (BIC) |
|---|---|---|---|---|---|---|---|---|
| DC | $-0.20\,[-0.24, -0.17]$ | 0.97 | $-0.98\,(p = 1.94 \times 10^{-6})$ | 0.26 | $-0.04$ | $-0.22$ | $-85.78 \to -94.01$ | True |
| DSA | $-0.32\,[-0.41, -0.24]$ | 0.92 | $-0.98\,(p = 1.94 \times 10^{-6})$ | 0.26 | $-0.52$ | $-0.22$ | $-69.11 \to -76.93$ | True |
| DM | $-0.43\,[-0.60, -0.26]$ | 0.84 | $-1.00\,(p = 0.00)$ | 0.21 | $-1.41$ | $-0.27$ | $-56.56 \to -82.93$ | True |
| MTT | $-0.17\,[-0.26, -0.08]$ | 0.73 | $-0.98\,(p = 1.94 \times 10^{-6})$ | 0.21 | $-0.77$ | $-0.09$ | $-67.88 \to -91.72$ | True |
| TESLA | $-0.27\,[-0.33, -0.21]$ | 1.00 | $-1.00\,(p = 0.00)$ | – | – | – | – | – |
| NCFM | $-0.23\,[-0.51, 0.06]$ | 0.99 | $-1.00\,(p = 0.00)$ | – | – | – | – | – |
| DATM | $-0.32\,[-0.42, -0.21]$ | 0.99 | $-1.00\,(p = 0.00)$ | – | – | – | – | – |
| SRe2L | $-1.51\,[-1.80, -1.22]$ | 1.00 | $-1.00\,(p = 0.00)$ | – | – | – | – | – |

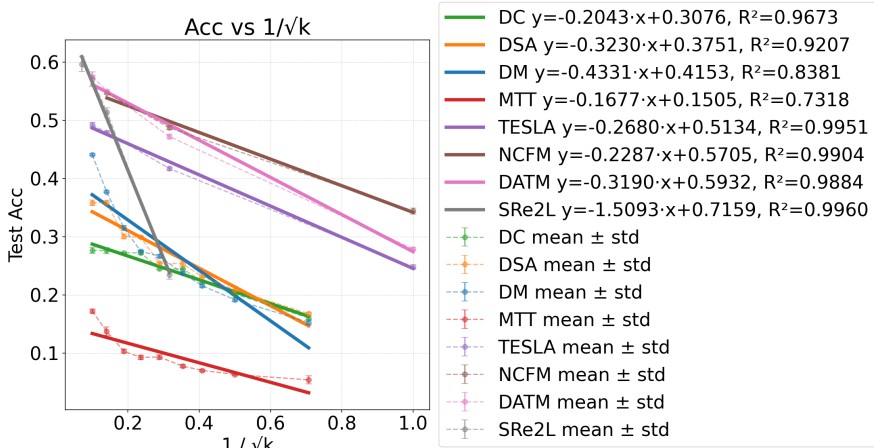

*Figure 7.* Single-configuration scaling law on CIFAR100.

($R^2 = 0.73$) while preserving monotonicity; the piecewise fit shows near-saturation after the breakpoint (right slope $-0.09$), consistent with amplified trajectory mismatch under increased task complexity. For **TESLA/NCFM/DATM/SRe2L**, the reported points follow a clean monotone trend with very high $R^2$ (often $\approx 1$), but piecewise analysis is not emphasized due to sparse IPC grids. In summary, CIFAR-100 strengthens the conclusion that the scaling trend is robust, while the *degree of saturation* becomes method-dependent and more visible in trajectory-based matching.

**MNIST.** As results shown in Tab. 6 and Fig. 8, MNIST exhibits the cleanest scaling behavior overall. **DSA** achieves an almost perfect single-line fit ($R^2 = 1$) with a narrow CI around slope $-0.13$. Although a broken-line fit can marginally improve AIC, BIC does not favor the extra complexity, indicating that MNIST+DSA is effectively a single linear regime. **DM** is also highly linear ($R^2 = 0.98$) with a small but consistent diminishing-returns effect (left slope $-0.13$ to right slope $-0.09$), suggesting steady but slightly saturating gains as IPC grows. **MTT** shows a steeper slope ($-0.33$) with high $R^2$ (0.93) and a moderate two-phase pattern (right slope $-0.19$), implying that trajectory-based matching can be sample-efficient on simple data without collapsing into a plateau. **DC** remains strongly linear ($R^2 = 0.92$) but has a noticeably weaker rank correlation ($\rho = -0.89$). The piecewise fit attributes this to a small early instability region where the left slope becomes slightly positive, followed by a clear decreasing regime (right slope $-0.15$). This clarifies that deviations are localized rather than contradicting the overall scaling trend.

**ImageNette (cross-architecture validation).** As results shown in Tab. 7 and Fig. 9, across architectures, Spearman $\rho \approx -0.99$ with highly significant $p$-values, and single-line $R^2$ values remain high (0.87–0.92), confirming that the scaling extends beyond a single backbone. Importantly, BIC consistently supports piecewise fits, indicating a robust two-phase structure. For **ConvNet**, the left slope is extremely steep ($-120.30$) and the right slope is much milder ($-39.04$), indicating large early gains and clear saturation. **ResNet18** shows a similar pattern (left $-103.34$ to right $-26.34$) with a later breakpoint, suggesting that stronger architectures delay saturation. **ResNet18_ap** yields the best single-line $R^2$ (0.92) and a comparatively less extreme drop (left $-87.35$ to right $-32.07$), implying more sustained improvement in the late regime. Overall, ImageNette provides an architecture-level robustness check and strong evidence that two-phase behavior is not an

*Table 6.* MNIST: regression diagnostics for the $1/\sqrt{k}$ scaling.

| Method | OLS slope ($\pm$95% CI) | $R^2$ | Spearman $\rho$ ($p$) | Break $x$ | Left slope | Right slope | AIC$_{\text{single}}$ $\to$AIC$_{\text{piece}}$ | Piecewise (BIC) |
|---|---|---|---|---|---|---|---|---|
| DC | $-0.12\,[-0.15, -0.09]$ | 0.92 | $-0.89\,(p = 5.42 \times 10^{-4})$ | 0.21 | 0.04 | $-0.15$ | $-96.46 \to -119.43$ | True |
| DSA | $-0.13\,[-0.13, -0.12]$ | 1.00 | $-1.00\,(p = 6.65 \times 10^{-64})$ | 0.26 | $-0.14$ | $-0.13$ | $-137.91 \to -138.05$ | AIC-yes / BIC-no |
| DM | $-0.10\,[-0.11, -0.09]$ | 0.98 | $-1.00\,(p = 6.65 \times 10^{-64})$ | 0.32 | $-0.13$ | $-0.09$ | $-114.47 \to -122.48$ | True |
| MTT | $-0.33\,[-0.41, -0.25]$ | 0.93 | $-1.00\,(p = 0.00)$ | 0.32 | $-0.29$ | $-0.19$ | $-70.08 \to -84.65$ | True |

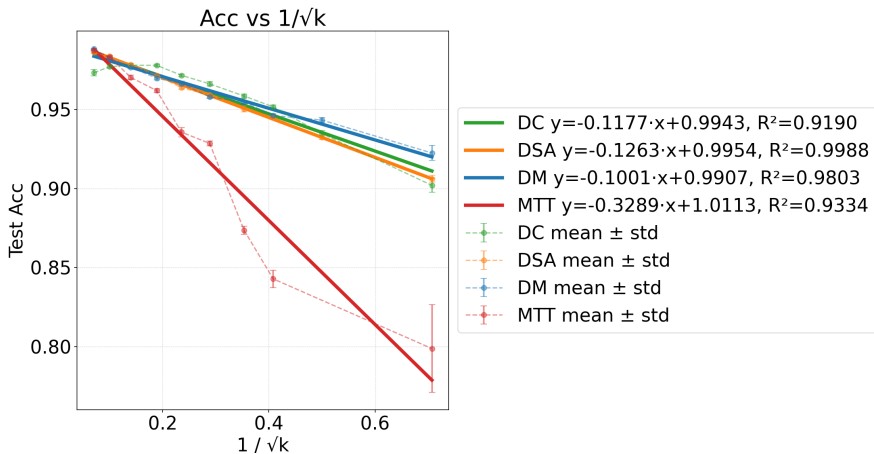

*Figure 8.* Single-configuration scaling law on MNIST.

artifact of a specific model.

**ImageNet-1K.** As results shown in Tab. 8 and Fig. 10, all reported methods show perfect monotonicity ($\rho = -1$) and high $R^2$ (0.89–1.00) under the available IPC settings. For **TESLA**, the OLS slope CI includes zero ($[-0.45, 0.01]$), indicating that with the limited IPC points the estimated linear rate is less statistically precise; nevertheless, the rank-based test confirms consistent monotone improvement. For **SRe2L** variants (R18/R50/R101), the fits are near-perfect ($R^2 \geq 0.98$) with strictly negative slopes, supporting a clean scaling trend. We do not draw piecewise conclusions here because the IPC grids inherited from prior work are too sparse to identify a breakpoint reliably.

**Tiny-ImageNet.** As results shown in Tab. 9 and Fig. 11, both methods show perfect monotonicity ($\rho = -1$) with high $R^2$ (0.96–1.00). **NCFM** exhibits a tight negative slope CI and a near-perfect fit, indicating stable scaling under the tested IPC values. **DATM** also shows strong $R^2$ but a wide slope CI that crosses zero, suggesting higher variance and insufficient precision to pin down the scaling rate from the sparse points, although the monotone trend remains unequivocal. As above, piecewise inference is not emphasized due to limited IPC grids.

**Takeaway across datasets and methods.** Overall, the empirical evidence strongly supports the universal *monotone* $1/\sqrt{k}$ scaling across datasets (Spearman $\rho$ is consistently strongly negative), while the *global single-line linearity* and the presence of *two-phase saturation* are method- and dataset-dependent. Matching-based methods (DC/DSA/DM/MTT) frequently exhibit identifiable two-phase behavior when evaluated on sufficiently dense IPC grids, whereas for baselines inheriting sparse IPC choices from prior work we rely on monotonicity and single-line OLS as the primary diagnostics.

### F.4. Cross-Configuration Coverage

**Cross-configuration coverage (detailed).** We report three complementary statistics to characterize empirical coverage scaling across configurations: (i) a *single-line OLS fit* (slope with 95% CI and $R^2$) to quantify approximate linearity; (ii) *rank-based monotonicity* via Spearman $\rho$ and associated $p$-values to verify that coverage error increases monotonically with the coverage complexity proxy; and (iii) *broken-line (piecewise) regression* with an estimated breakpoint and left/right slopes, selected by AIC/BIC, to identify potential multi-regime behavior (e.g., early instability vs. stable scaling). Throughout, a

*Table 7.* ImageNette: regression diagnostics for the $1/\sqrt{k}$ scaling.

| Architecture | OLS slope (±95% CI) | $R^2$ | Spearman $\rho$ ($p$) | Break $x$ | Left slope | Right slope | AIC$_{single}$ →AIC$_{piece}$ | Piecewise (BIC) |
|---|---|---|---|---|---|---|---|---|
| ConvNet | $-67.42$ $[-84.64, -50.19]$ | 0.90 | $-0.99$ ($p = 3.76 \times 10^{-9}$) | 0.38 | $-120.30$ | $-39.04$ | $43.82 \to 18.14$ | True |
| ResNet18 | $-57.04$ $[-72.87, -41.21]$ | 0.87 | $-0.99$ ($p = 1.30 \times 10^{-10}$) | 0.45 | $-103.34$ | $-26.34$ | $46.24 \to 24.70$ | True |
| ResNet18_ap | $-55.61$ $[-67.00, -44.22]$ | 0.92 | $-0.99$ ($p = 1.30 \times 10^{-10}$) | 0.45 | $-87.35$ | $-32.07$ | $38.34 \to 18.52$ | True |

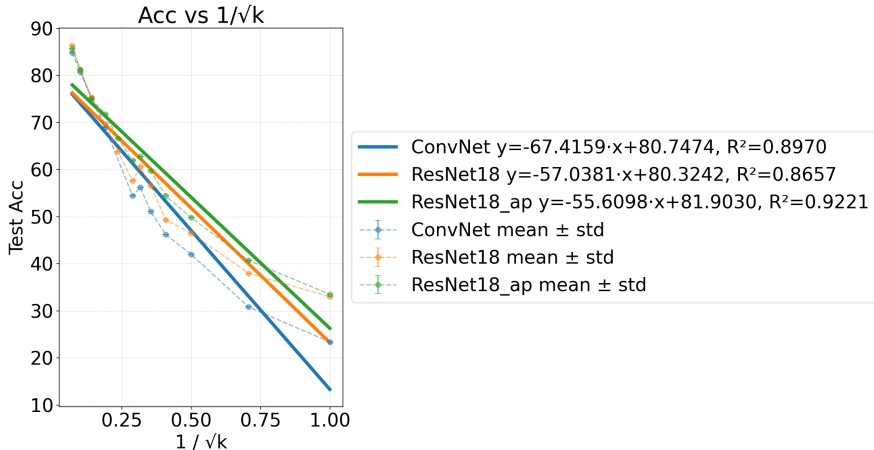

*Figure 9.* Single-configuration scaling law on ImageNette.

positive slope indicates higher sensitivity of the distilled dataset to increasing cross-configuration coverage complexity.

**MNIST.** As results shown in Tab. 10 and Fig. 12, MNIST exhibits the cleanest and most stable cross-configuration scaling among all datasets. All methods show extremely strong monotonicity, with Spearman $\rho \geq 0.88$ and $p$-values far below $10^{-50}$, confirming that coverage error consistently increases with the coverage complexity proxy. **DSA** achieves an almost perfect single-line fit ($R^2 = 0.99$) with a tightly concentrated slope (384,291 with 95% CI [378,328, 390,254]), indicating a highly regular scaling regime. The piecewise model is favored by both AIC and BIC, but the left and right slopes (394,817 vs. 306,358) are comparable, suggesting only mild diminishing returns rather than a qualitative regime change. **DC** also shows strong linearity ($R^2 = 0.98$) and high monotonicity ($\rho = 0.967$, $p = 1.51 \times 10^{-105}$). The piecewise fit reveals a short initial instability region (negative left slope $-131,651$) followed by a dominant linear regime with a steep right slope (474,732), indicating that once the configuration coverage becomes sufficiently rich, error growth follows a stable law. **DM** remains highly monotone ($\rho = 0.9996$, $p = 5.33 \times 10^{-233}$) with a strong single-line fit ($R^2 = 0.96$). The piecewise model indicates modest saturation, transitioning from a steeper early slope (266,903) to a milder late slope (149,993), consistent with a gradual reduction in sensitivity as IPC increases. **MTT** shows the weakest linear fit on MNIST ($R^2 = 0.72$) but still maintains strong monotonicity ($\rho = 0.875$, $p = 2.76 \times 10^{-53}$). The piecewise regression reveals a pronounced two-phase behavior: a very steep early slope (881,260) followed by a near-plateau (73,893), indicating that trajectory-matching objectives are highly sensitive to configuration diversity at small IPC but saturate quickly on simple datasets.

**CIFAR-10.** As results shown in Tab. 11 and Fig. 13, on CIFAR-10, monotonicity remains universal across methods (all $\rho \geq 0.73$, $p < 10^{-29}$), but deviations from a single global line become more pronounced, reflecting higher configuration diversity. **DSA** achieves the strongest overall fit ($R^2 = 0.90$) with an intermediate slope (310,534 with 95% CI [294,355, 326,713]). The piecewise model is strongly favored and reveals a clear diminishing-returns pattern, transitioning from a steep early slope (408,026) to a gentler late slope (143,668). **DC** shows moderate linearity ($R^2 = 0.74$) and a smaller slope (218,683 with 95% CI [198,431, 238,934]). The piecewise fit attributes the reduced $R^2$ to a short early region with negative slope ($-145,239$), followed by a stable positive-slope regime (223,219), suggesting localized instability rather than a violation of the scaling law. **DM** exhibits a relatively steep single-line slope (382,698 with 95% CI [352,600, 412,796]) but a lower $R^2$ (0.79). This is explained by a pronounced two-phase behavior: a very sharp early increase (639,800) followed by a negative late slope ($-68,151$), indicating over-sensitivity to configuration diversity at small IPC and strong saturation thereafter.

*Table 8.* ImageNet-1K: regression diagnostics for the $1/\sqrt{k}$ scaling.

| Architecture | OLS slope ($\pm$95% CI) | $R^2$ | Spearman $\rho$ $(p)$ | Break $x$ | Left slope | Right slope | AIC$_{\text{single}}$ $\rightarrow$AIC$_{\text{piece}}$ | Piecewise (BIC) |
|---|---|---|---|---|---|---|---|---|
| TESLA | $-0.22\,[-0.45, 0.01]$ | 0.89 | $-1.00\,(p = 0.00)$ | – | – | – | – | – |
| SRe2L_R18 | $-1.46\,[-1.47, -1.44]$ | 1.00 | $-1.00\,(p = 0.00)$ | – | – | – | – | – |
| SRe2L_R50 | $-1.49\,[-1.67, -1.31]$ | 1.00 | $-1.00\,(p = 0.00)$ | – | – | – | – | – |
| SRe2L_R101 | $-1.46\,[-2.11, -0.82]$ | 0.98 | $-1.00\,(p = 0.00)$ | – | – | – | – | – |

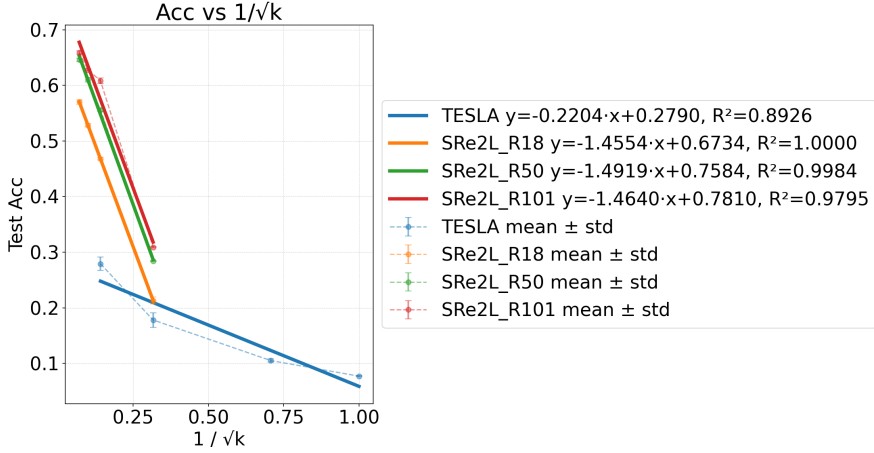

*Figure 10.* Single-configuration scaling law on ImageNet-1K.

**MTT** yields the weakest fit ($R^2 = 0.56$) despite strong monotonicity ($\rho = 0.879$, $p = 3.06 \times 10^{-54}$). The piecewise regression shows an explicit plateau with a negative right slope ($-82{,}911$), implying that additional coverage complexity does not translate into higher error beyond the breakpoint.

**CIFAR-100.**  As results shown in Tab. 12 and Fig. 14, CIFAR-100 presents the most challenging regime. While monotonicity remains statistically significant for all methods, single-line linearity degrades for several baselines, consistent with higher class count and increased variability. **DM** achieves the largest slope magnitude (259,424 with 95% CI [234,561, 284,287]) with a reasonable $R^2$ (0.72), indicating high sensitivity to configuration diversity. The piecewise fit reveals a dramatic early-growth regime (724,198) followed by a substantially milder late slope (127,013), reflecting strong early vulnerability and partial saturation. **DSA** maintains a strong fit ($R^2 = 0.82$) with a moderate slope (149,092 with 95% CI [138,118, 160,066]). The two-phase structure is evident, with a sharp early increase (242,359) transitioning to a much flatter late regime (18,906), suggesting effective stabilization as IPC grows. **DC** shows weaker linearity ($R^2 = 0.36$) and a smaller slope (88,174 with 95% CI [70,698, 105,649]). However, the piecewise model clarifies this behavior: an extremely steep early region (667,336) is followed by a nearly flat late regime (77,072), indicating that most coverage sensitivity is concentrated at small IPC. **MTT** exhibits the lowest $R^2$ (0.27) but preserves monotonicity ($\rho = 0.703$, $p = 6.27 \times 10^{-26}$). The piecewise regression again reveals strong early sensitivity (395,640) and a negative late slope ($-12{,}576$), consistent with amplified trajectory mismatch effects under high task complexity.

**Summary.**  Across datasets, the empirical coverage law is consistently supported by monotonicity and piecewise analysis. The OLS slope acts as a method-dependent measure of sensitivity to configuration diversity: trajectory-based methods (MTT) exhibit the strongest early sensitivity and fastest saturation, matching-based methods (DM) show high sample efficiency but pronounced two-phase behavior, while gradient-matching approaches (DC/DSA) achieve more stable and predictable scaling across configurations.

### F.5. Robustness to Alternative Configuration Complexity Proxies

In the main text, we instantiate configurational complexity using a coverage-based proxy $H_{\text{cov}}$ derived from local training-update distances. To assess whether the observed coverage law is an artifact of this particular estimator, we conduct an

*Table 9.* Tiny-ImageNet: regression diagnostics for the $1/\sqrt{k}$ scaling.

| Method | OLS slope ($\pm$95% CI) | $R^2$ | Spearman $\rho$ $(p)$ | Break $x$ | Left slope | Right slope | AIC$_{\text{single}}$ $\rightarrow$AIC$_{\text{piece}}$ | Piecewise (BIC) |
|--------|-------------------------|-------|-----------------------|-----------|------------|-------------|--------------------------------------------------------|-----------------|
| DATM | $-0.25\,[-0.86, 0.36]$ | 0.96 | $-1.00\,(p=0.00)$ | – | – | – | $-20.16 \rightarrow \infty$ | False |
| NCFM | $-0.13\,[-0.20, -0.06]$ | 1.00 | $-1.00\,(p=0.00)$ | – | – | – | $-32.89 \rightarrow \infty$ | False |

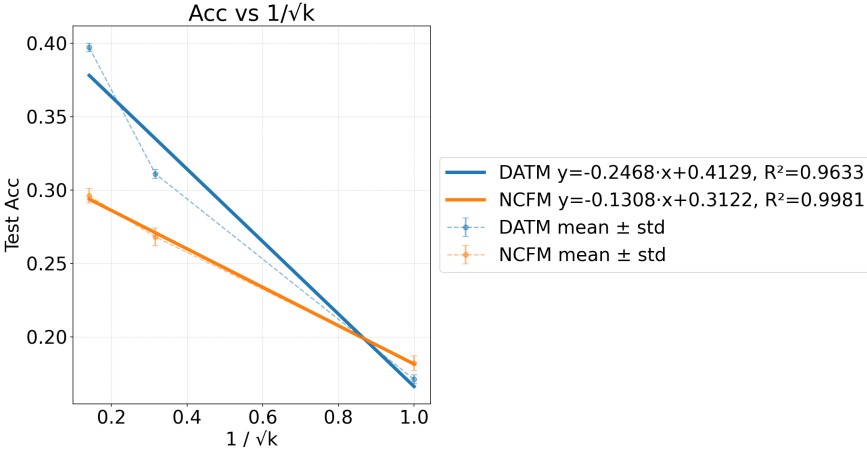

*Figure 11.* Single-configuration scaling law on Tiny-ImageNet.

additional robustness analysis by replacing $H_{\text{cov}}$ with a much coarser alternative, $\log M$, where $M$ denotes the number of target training configurations in the configuration.

Figure 15 shows the resulting relationship between the generalization error $\Delta_a(\hat{\mu}_\tau, \hat{\mu}_s, \nu)$ and $\sqrt{\log M/k}$ across MNIST, CIFAR-10, and CIFAR-100, for multiple dataset distillation methods (DC, DSA, DM, and MTT). Despite ignoring any geometric or dynamical similarity between configurations, $\log M$ exhibits a consistent and approximately linear scaling trend across datasets and methods, with goodness-of-fit statistics comparable to those obtained using $H_{\text{cov}}$.

This observation indicates that the proposed coverage law reflects a genuine dependence on configurational complexity rather than a fragile consequence of a specific complexity estimator. At the same time, $\log M$ constitutes a coarse-grained proxy that only captures the cardinality of the target configuration, and cannot distinguish between configurations with identical $M$ but different degrees of dispersion or redundancy among configurations. In contrast, $H_{\text{cov}}$ explicitly accounts for similarity structure through an $r$-cover construction in the space of training dynamics, enabling finer discrimination between such cases and supporting tighter quantitative predictions.

Taken together, these results demonstrate that the coverage law is robust to the choice of complexity proxy, while also motivating the use of structured estimators such as $H_{\text{cov}}$ when finer configurational distinctions are required.

### F.6. Predictive Validation of the Scaling and Coverage Laws

To empirically validate the predictive power of our theory, we design two complementary *out-of-sample prediction* experiments that probe generalization along two orthogonal axes: (i) distilled *dataset size* (budget $k$) and (ii) *training configuration* shift.

**Evaluation protocol and metrics.** In both experiments, we evaluate prediction quality only on the *held-out* (unseen) conditions. Let $a$ denote the actual test accuracy and $\hat{a}$ the predicted accuracy. We report absolute error statistics:

$$e = |\hat{a} - a|, \quad \text{MAE} = \mathbb{E}[e], \quad \text{RMSE} = \sqrt{\mathbb{E}[e^2]}, \quad \text{MaxAE} = \max(e), \quad \text{MAPE} = \mathbb{E}\left[\frac{e}{a + \epsilon}\right], \quad (143)$$

where $\epsilon$ is a small constant to avoid numerical instability for very small $a$. For readability, we present results at two granularities: (i) *global* parity plots aggregating all unseen points, and (ii) *per-dataset & per-method* error tables that reveal

*Table 10.* MNIST: regression diagnostics for the $1/\sqrt{k}$ scaling.

| Method | OLS slope ($\pm95\%$ CI) | $R^2$ | Spearman $\rho$ ($p$) | Break $x$ | Left slope | Right slope | AIC$_{\text{single}}$ →AIC$_{\text{piece}}$ | Piecewise (BIC) |
|---|---|---|---|---|---|---|---|---|
| DC | 437232.84 [427209.18, 447256.50] | 0.9771 | 0.967255 ($p = 1.510 \times 10^{-105}$) | 0 | −131650.69 | 474732.33 | −1408.50 → −1650.29 | True |
| DSA | 384291.08 [378328.38, 390253.77] | 0.9900 | 0.999717 ($p = 1.283 \times 10^{-266}$) | 0 | 394816.70 | 306357.60 | −1502.56 → −1737.94 | True |
| DM | 223703.97 [216202.84, 231205.10] | 0.9591 | 0.999623 ($p = 5.326 \times 10^{-233}$) | 0 | 266902.81 | 149993.33 | −1323.76 → −1474.68 | True |
| MTT | 720274.53 [651261.58, 789287.47] | 0.7227 | 0.875225 ($p = 2.758 \times 10^{-53}$) | 0 | 881259.83 | 73893.12 | −694.46 → −772.98 | True |

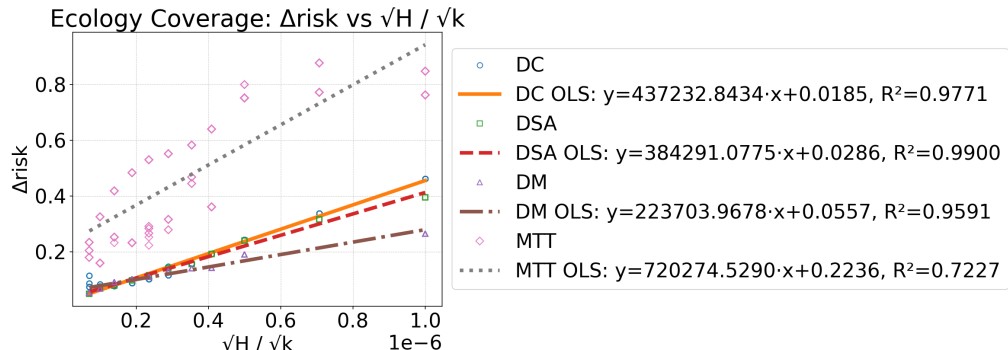

*Figure 12.* Configuration coverage law on MNIST.

method- and dataset-specific behaviors.

**Scaling-law prediction (unseen budget).** For each dataset (MNIST, CIFAR-10, CIFAR-100) and each distillation method (DC, DM, DSA, MTT), we collect test accuracies of distilled datasets at multiple budgets $k$. We treat a subset of budgets as *observed* and fit the scaling law using only these points, then *hold out* the remaining budgets as *unseen* and predict their accuracies from the fitted law. We summarize the predictive accuracy on unseen budgets using both the parity plot and the detailed error table (see Table 13).

**Coverage-law prediction (unseen configuration).** We next test the coverage law under *configuration shift*. For each fixed budget $k$, we assume access to accuracies under a finite set of *observed* training configurations and use the coverage law to predict performance under *unseen* configurations. We then obtain the *actual* unseen-configuration accuracy by training on the corresponding distilled dataset under that unseen configuration, and compare it with the predicted value. As above, evaluation is performed only on unseen configurations and summarized via parity plots and aggregate errors, with per-dataset & per-method breakdown in Table 14.

**Results and quantitative analysis.** Figure 5(a) shows the scaling-law parity plot across all datasets and methods. Unseen-budget points concentrate tightly around the identity line, indicating accurate out-of-sample prediction. The figure reports a global MAE of 0.0214, RMSE of 0.0275, and MaxAE of 0.0688 over $N = 36$ unseen-budget points. These global trends are consistent with the fine-grained statistics in Table 13, which further shows that scaling-law prediction remains stable across datasets/methods, with typically small absolute errors and bounded worst-case deviations.

Figure 5(b) reports the corresponding parity plot for the coverage-law experiment. Predictions exhibit a systematic downward shift relative to the identity line, i.e., conservative underestimation of accuracy under unseen configurations. This behavior is expected: the coverage law is constructed to estimate *worst-case* performance within the covered configuration space, prioritizing risk-aware extrapolation over unbiased point prediction. Table 14 complements this observation by quantifying the conservativeness across datasets: while absolute errors increase under configuration shift (relative to unseen-budget extrapolation), the predictions remain well-aligned with actual accuracies in terms of monotonic trends, and the degree of conservativeness varies systematically with dataset difficulty and method characteristics.

Taken together, these results demonstrate the complementary roles of the two laws: the scaling law yields accurate point predictions under unseen budgets, while the coverage law provides conservative yet reliable extrapolation under unseen training configurations, validating our theoretical framework under genuine distribution shifts.

*Table 11.* CIFAR10: regression diagnostics for the $1/\sqrt{k}$ scaling.

| Method | OLS slope ($\pm$95% CI) | $R^2$ | Spearman $\rho$ ($p$) | Break $x$ | Left slope | Right slope | $\text{AIC}_{\text{single}} \rightarrow \text{AIC}_{\text{piece}}$ | Piecewise (BIC) |
|---|---|---|---|---|---|---|---|---|
| DC | 218682.52 [198430.73, 238934.31] | 0.7361 | 0.734500 ($p = 3.026 \times 10^{-29}$) | 0 | $-145238.86$ | 223219.18 | $-1099.06 \rightarrow -1123.54$ | True |
| DSA | 310534.28 [294355.16, 326713.40] | 0.8981 | 0.995293 ($p = 3.757 \times 10^{-167}$) | 0 | 408025.98 | 143668.49 | $-1173.15 \rightarrow -1543.62$ | True |
| DM | 382698.28 [352600.32, 412796.24] | 0.7946 | 0.971444 ($p = 9.378 \times 10^{-104}$) | $1 \times 10^{-6}$ | 639800.20 | $-68151.45$ | $-968.31 \rightarrow -1222.08$ | True |
| MTT | 173469.99 [149680.36, 197259.62] | 0.5598 | 0.878770 ($p = 3.059 \times 10^{-54}$) | $1 \times 10^{-6}$ | 332818.20 | $-82910.82$ | $-1045.93 \rightarrow -1139.17$ | True |

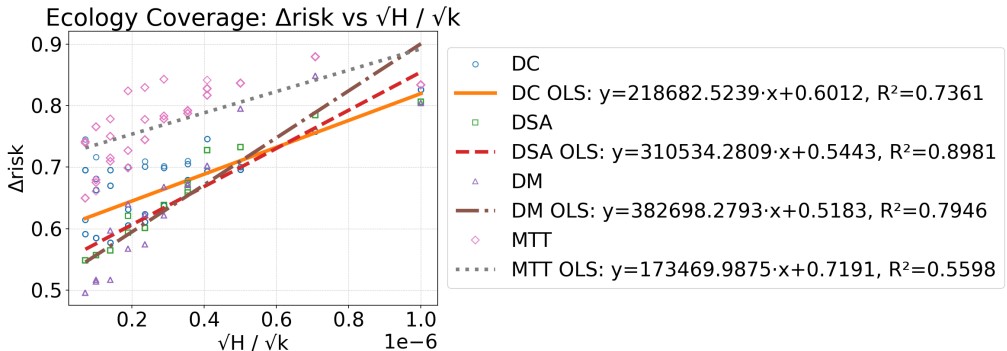

*Figure 13.* Configuration coverage law on CIFAR10.

## F.7. Coverage-Guided Source Selection

Beyond post-hoc explanation, the coverage law can guide the choice of source configurations used for distillation. We therefore ran small coverage-guided source-selection reruns in which the source subset is selected by a proxy coverage score, followed by true re-distillation and evaluation on held-out target configurations. The goal is diagnostic rather than to establish a new algorithmic benchmark. As summarized in Table 15, proxy-guided selection improves average held-out robustness in several dataset/method/source-family blocks, including MNIST–DC, MNIST–DM, CIFAR-10–DSA, and CIFAR-100–DM/DSA under broader source families. Some harder blocks remain mixed, especially on CIFAR-10/100 when worst-case performance is emphasized. Thus, the coverage proxy is operationally informative, but not a uniform dominance rule; it should be interpreted as a risk-aware guide for selecting diverse source recipes.

## F.8. Verification of Trajectory-Local Contractivity (Local PL Condition)

Our theoretical analysis relies on Assumption 4.1(iii), which requires *trajectory-locally stable dynamics*: along the realized optimization trajectory, the one-step update map admits a local contraction factor $\rho_{a,t} \in (0, 1)$. Importantly, this is a *local* requirement along the parameters visited during training, and does not assume any global PL inequality, strong convexity, or global contractivity over the entire parameter space.

**How we probe local contractivity.** Following the probing procedure in Remark 4.4, we estimate a trajectory-local contractivity coefficient $\hat{\rho}_t$ during training by measuring the sensitivity of the *one-step* update operator to small parameter perturbations. Concretely, at selected steps we perturb the current parameters by a small vector $\delta$ and compare the resulting one-step updated parameters under the same minibatch; this yields an empirical local contraction estimate $\hat{\rho}_t$. To be conservative, we summarize stability using a *post-burn-in* statistic, e.g., $\rho_{p90}$, and report the margin to the contractivity boundary, $1 - \rho_{p90}$ (log-scaled in the main plots).

**Observed two-phase behavior: burn-in vs. stable regime.** Across MNIST, CIFAR-10, and CIFAR-100 (Figures 16–18), we consistently observe a clear two-phase pattern: (i) an initial *burn-in* transient where $\hat{\rho}_t$ changes rapidly and can be noisy (reflecting early representation formation and optimizer state adaptation), followed by (ii) a *stable regime* after the burn-in divider (vertical dashed line), where $\hat{\rho}_t$ becomes highly stable and remains strictly within the contractive region. This supports the intended use of Assumption 4.1(iii), which is invoked to characterize the optimization dynamics *along the realized trajectory* rather than at initialization.

*Table 12.* CIFAR100: regression diagnostics for the $1/\sqrt{k}$ scaling.

| Method | OLS slope ($\pm$95% CI) | $R^2$ | Spearman $\rho$ ($p$) | Break $x$ | Left slope | Right slope | $\text{AIC}_{\text{single}} \rightarrow \text{AIC}_{\text{piece}}$ | Piecewise (BIC) |
|---|---|---|---|---|---|---|---|---|
| DC | 88173.79 [70698.18, 105649.40] | 0.36 | 0.597003 ($p = 2.237 \times 10^{-18}$) | 0 | 667335.69 | 77071.69 | $-1205.35 \rightarrow -1225.69$ | True |
| DSA | 149092.02 [138118.32, 160065.71] | 0.82 | 0.975961 ($p = 9.030 \times 10^{-110}$) | $1 \times 10^{-6}$ | 242359.46 | 18905.97 | $-1301.27 \rightarrow -1462.45$ | True |
| DM | 259423.76 [234560.75, 284286.76] | 0.72 | 0.945804 ($p = 1.580 \times 10^{-81}$) | 0 | 724197.60 | 127012.95 | $-1031.36 \rightarrow -1222.49$ | True |
| MTT | 57021.33 [42669.14, 71373.52] | 0.27 | 0.703224 ($p = 6.266 \times 10^{-26}$) | 0 | 395639.98 | $-12575.66$ | $-1212.69 \rightarrow -1396.78$ | True |

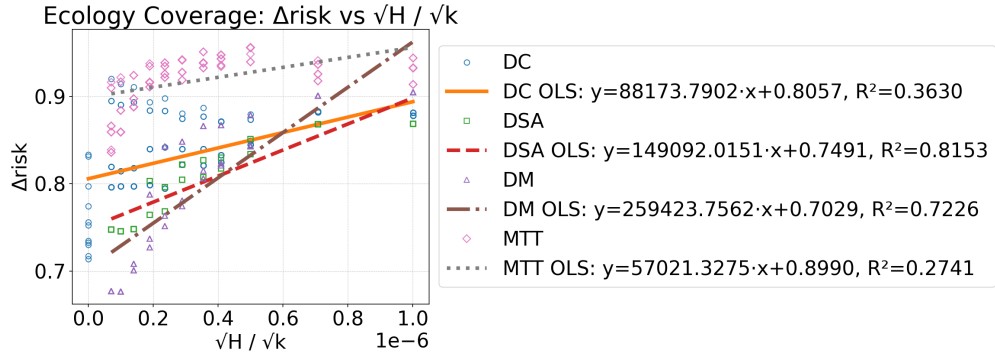

*Figure 14.* Configuration coverage law on CIFAR100.

**Robustness across distilled set size $k$ and distillation methods.** The same qualitative stability holds across a wide range of distilled set sizes $k$ (e.g., $k = 2, 8, 28, 100$ in the plotted examples). In particular, increasing $k$ does not introduce instability; instead, the post-burn-in curves remain stable and typically exhibit a non-worsening (often improving) margin to the boundary. Moreover, we observe consistent post-burn-in contractivity across multiple distillation sources (DC/DM/DSA/MTT), suggesting that the stability is not an artifact of a specific distillation method.

**Dataset difficulty and margin interpretation.** While all datasets exhibit post-burn-in contractivity, the effective margin to the boundary is dataset-dependent: CIFAR-100 (harder classification and higher diversity) shows comparatively tighter margins than MNIST, whereas MNIST typically attains the largest safety margins. This is expected: harder datasets and higher-capacity decision boundaries can induce updates that are closer to the stability boundary, yet the key point is that the dynamics remain *contractive* in the post-burn-in regime across all tested settings.

**Takeaway.** Overall, these results provide empirical evidence that the trajectory-local PL/contractivity condition required by Assumption 4.1(iii) is *practically satisfied* during training on distilled datasets. The condition is local, data-dependent, and emerges naturally from standard architectures and optimizers, rather than being a strong global regularity assumption.

## G. Limitation and Future Work

Despite providing a unified Configuration-dynamics-error framework, our study still has several limitations that we explicitly acknowledge.

**Target Configuration.** Our analysis primarily characterizes cross-configuration robustness within a controlled but practically relevant configuration family, focusing on ConvNet-style architectures and commonly used optimization and augmentation choices. While this setting captures a large fraction of current dataset distillation practice, extending the coverage law to substantially more heterogeneous model families (e.g., radically different backbone paradigms or large-scale training regimes) remains an important direction for future work.

To partially probe this regime, we include additional cross-backbone evaluations on larger foundation-model variants in Appendix F.5, which suggest that the proposed scaling and coverage trends persist qualitatively. However, a comprehensive characterization of coverage complexity across highly diverse backbone families is beyond the scope of the present study and would likely require refined coverage metrics tailored to long-horizon training dynamics.

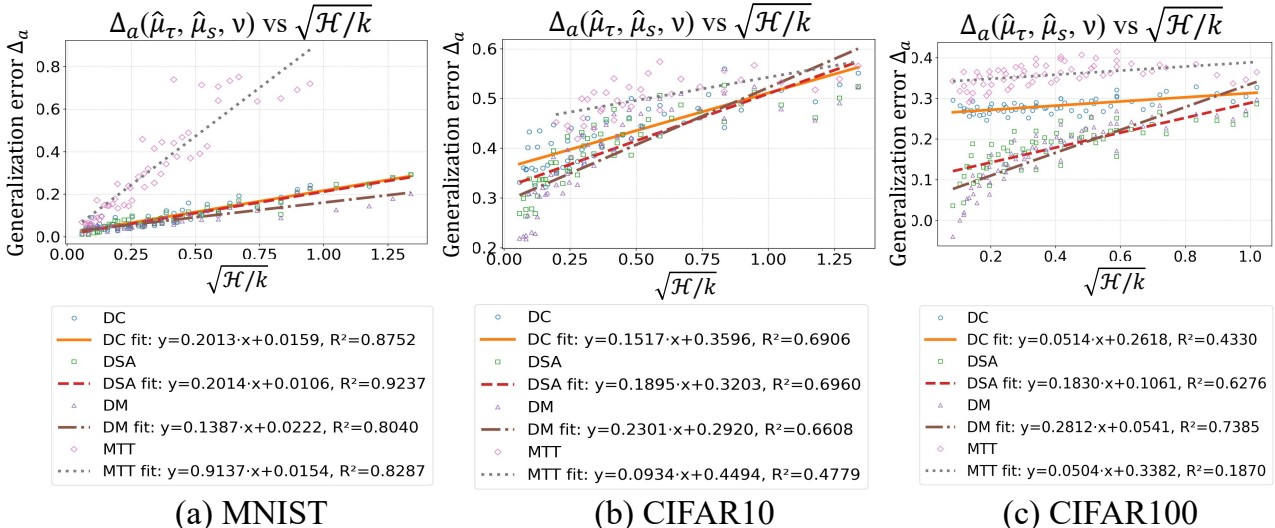

*Figure 15.* Coverage law using a coarse complexity proxy $\log M$, where $M$ is the number of target training configurations. Across datasets and distillation methods, generalization error scales approximately linearly with $\sqrt{\log M/k}$, demonstrating that the coverage law is robust to the choice of complexity proxy. Compared to $\log M$, $H_{\mathrm{cov}}$ enables finer discrimination by accounting for similarity among configurations.

**Coverage Complexity Estimation.** We instantiate configurational complexity with a practical proxy $H_{\mathrm{cov}}$ based on local training-update distances and an $r$-cover construction. This choice is motivated by computational tractability and by the fact that local update behavior is the quantity that directly enters our theoretical analysis.

We emphasize that $H_{\mathrm{cov}}$ is a surrogate rather than a unique, ground-truth notion of configuration complexity. Importantly, our empirical conclusions are not tied to this specific estimator: we obtain the same qualitative coverage scaling when replacing $H_{\mathrm{cov}}$ with a coarse alternative proxy $\log M$, where $M$ is the number of target training configurations in the configuration (Appendix F.5). This proxy replacement suggests that the coverage law reflects a genuine dependence on configurational complexity rather than an artifact of a particular estimator.

At the same time, $\log M$ captures only the cardinality of the configuration and ignores similarity structure among configurations, whereas $H_{\mathrm{cov}}$ aims to account for dispersion and redundancy via covering in the space of training dynamics. A more systematic comparison across alternative distance definitions (e.g., short-horizon trajectory distances) and covering constructions, and an understanding of when different proxies become equivalent or diverge, remains an interesting direction for strengthening the empirical instantiation of the theory.

**Stability Assumptions.** Our analysis leverages local contractivity of training trajectories, which we empirically verify in the post-burn-in regime of the studied configurations. In strongly non-contractive regimes (e.g., very large learning rates, extreme augmentations, or other sources of optimization instability), local update-based proxies may underestimate configurational complexity and the resulting predictions may degrade. Characterizing such regimes more sharply and designing estimators that remain reliable under non-contractive dynamics are promising directions for future work.

## H. Usage of LLM

In preparing this work, we made limited use of ChatGPT (OpenAI) as a supportive tool. Specifically, it was consulted in two ways:

- **Coding support**: ChatGPT-4o was occasionally used during debugging to suggest possible corrections for coding errors. All implementations were written, tested, and verified independently by the authors.

- **Language polishing**: At the final stage of manuscript preparation, ChatGPT-5 was used to polish the English expression of the appendix. The suggestions were carefully reviewed and adapted by the authors to ensure accuracy and consistency with the original technical content.

*Table 13.* Predictive validation results of scaling law on MNIST, CIFAR10 and CIFAR100.

| Dataset | Method | $MAE_\Delta$ | $RMSE_\Delta$ | $MaxAE_\Delta$ | $MAPE_\Delta$ | $MAE_{acc}$ | $RMSE_{acc}$ | $MaxAE_{acc}$ |
|---|---|---|---|---|---|---|---|---|
| MNIST | DC | 0.006743 | 0.008515 | 0.012898 | 0.162805 | 0.006743 | 0.008515 | 0.012898 |
| | DM | 0.007046 | 0.008299 | 0.012431 | 0.144498 | 0.007046 | 0.008299 | 0.012431 |
| | DSA | 0.005003 | 0.005134 | 0.006166 | 0.143334 | 0.005003 | 0.005134 | 0.006166 |
| | MTT | 0.013707 | 0.018575 | 0.031383 | 0.082897 | 0.013707 | 0.018575 | 0.031383 |
| CIFAR10 | DC | 0.005139 | 0.008363 | 0.014467 | 0.010305 | 0.005139 | 0.008363 | 0.014467 |
| | DM | 0.043863 | 0.049083 | 0.068817 | 0.077717 | 0.043863 | 0.049083 | 0.068817 |
| | DSA | 0.028151 | 0.028752 | 0.032541 | 0.051901 | 0.028151 | 0.028752 | 0.032541 |
| | MTT | 0.032106 | 0.033242 | 0.044223 | 0.047641 | 0.032106 | 0.033242 | 0.044223 |
| CIFAR100 | DC | 0.007047 | 0.007989 | 0.012258 | 0.009572 | 0.007047 | 0.007989 | 0.012258 |
| | DM | 0.038235 | 0.039995 | 0.052172 | 0.052726 | 0.038235 | 0.039995 | 0.052172 |
| | DSA | 0.032860 | 0.033002 | 0.037190 | 0.046350 | 0.032860 | 0.033002 | 0.037190 |
| | MTT | 0.036338 | 0.037043 | 0.046294 | 0.040394 | 0.036338 | 0.037043 | 0.046294 |

*Table 14.* Predictive validation results of coverage law on MNIST, CIFAR10 and CIFAR100.

| Dataset | Method | $MAE_\Delta$ | $RMSE_\Delta$ | $MaxAE_\Delta$ | $MAPE_\Delta$ | $MAE_{acc}$ | $RMSE_{acc}$ | $MaxAE_{acc}$ |
|---|---|---|---|---|---|---|---|---|
| CIFAR10 | DC | 0.159212 | 0.165302 | 0.213556 | 0.291445 | 0.159212 | 0.165302 | 0.213556 |
| | DM | 0.132588 | 0.138629 | 0.194003 | 0.245537 | 0.132588 | 0.138629 | 0.194003 |
| | DSA | 0.105963 | 0.111955 | 0.174449 | 0.199629 | 0.105963 | 0.111955 | 0.174449 |
| | MTT | 0.122425 | 0.126423 | 0.183704 | 0.184595 | 0.122425 | 0.126423 | 0.183704 |
| CIFAR100 | DC | 0.127500 | 0.139796 | 0.186213 | 0.171400 | 0.127500 | 0.139796 | 0.186213 |
| | DM | 0.068400 | 0.087921 | 0.155070 | 0.105750 | 0.068400 | 0.087921 | 0.155070 |
| | DSA | 0.066012 | 0.073004 | 0.112742 | 0.094282 | 0.066012 | 0.073004 | 0.112742 |
| | MTT | 0.046968 | 0.058021 | 0.097214 | 0.054601 | 0.046968 | 0.058021 | 0.097214 |
| MNIST | DC | 0.144863 | 0.180533 | 0.347909 | 2.366409 | 0.144863 | 0.180533 | 0.347909 |
| | DM | 0.103563 | 0.116431 | 0.219012 | 2.369760 | 0.103563 | 0.116431 | 0.219012 |
| | DSA | 0.123725 | 0.149789 | 0.305028 | 2.293224 | 0.123725 | 0.149789 | 0.305028 |
| | MTT | 0.165995 | 0.188383 | 0.375658 | 0.892966 | 0.165995 | 0.188383 | 0.375658 |

No AI tool was involved in generating research ideas, conducting experiments, or drawing conclusions. All scientific contributions are the authors' own.

*Table 15.* Coverage-guided source-selection reruns. Each row re-distills from the selected source subset and evaluates on held-out target configurations.

| Dataset | Method | Source family | IPC | Proxy avg / worst | Outcome |
|---------|--------|---------------|-----|-------------------|---------|
| MNIST | DC | arch+aug, SGD | 8,12 | 0.8923/0.8118; 0.9054/0.8062 | positive vs. random and naive |
| MNIST | DM | arch+aug, SGD | 18 | 0.9129 / 0.8306 | positive vs. random and naive |
| CIFAR-10 | DC | arch+aug, SGD | 12,18 | 0.2310/0.1802; 0.2701/0.2268 | mixed at 12, positive at 18 |
| CIFAR-10 | DSA | arch+aug, SGD | 18 | 0.3167 / 0.2685 | positive vs. random and naive |
| CIFAR-10 | DM | arch+aug, SGD | 18 | 0.2606 / 0.1949 | mixed vs. random, avg positive vs. naive |
| CIFAR-100 | DM | arch+optim, no aug | 18 | 0.1177 / 0.0790 | positive vs. random and naive |
| CIFAR-100 | DSA | arch+optim, no aug | 18 | 0.1148 / 0.0727 | positive vs. random and naive |

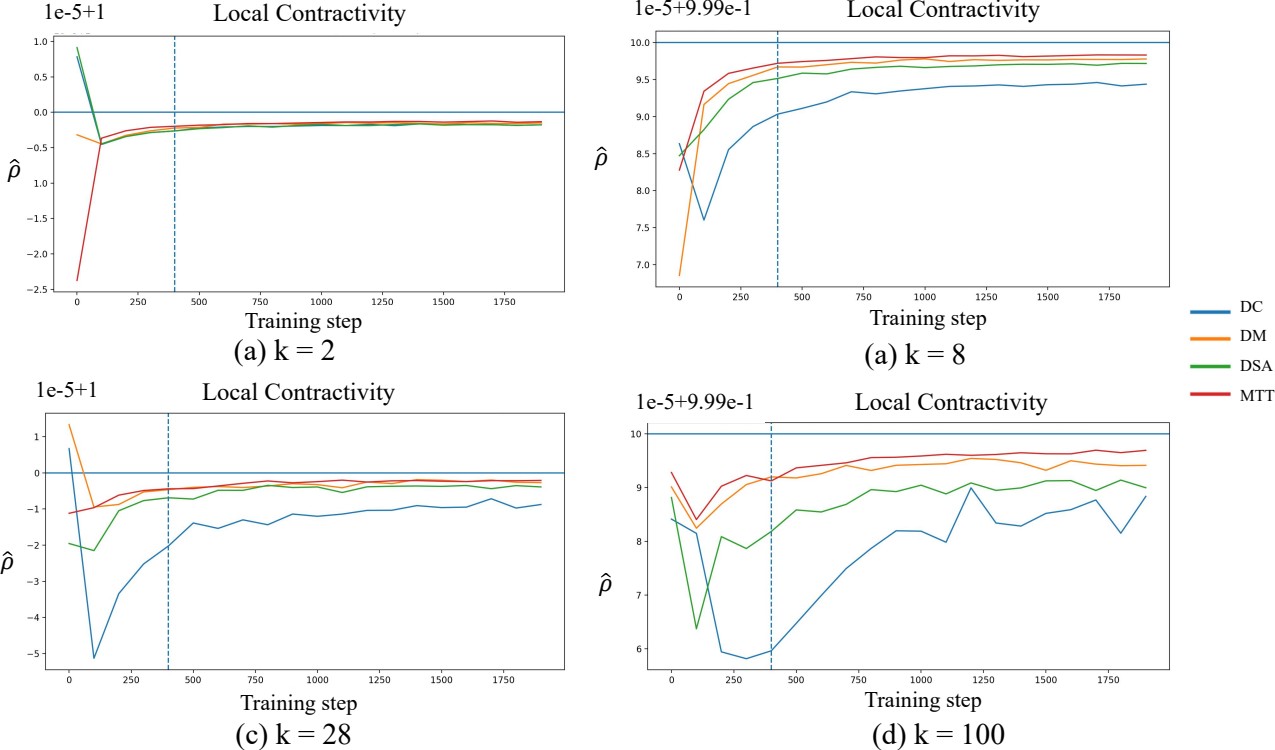

*Figure 16.* Trajectory-local contractivity probe along the optimization trajectory on MNIST (post-burn-in stability indicated by the dashed vertical line).

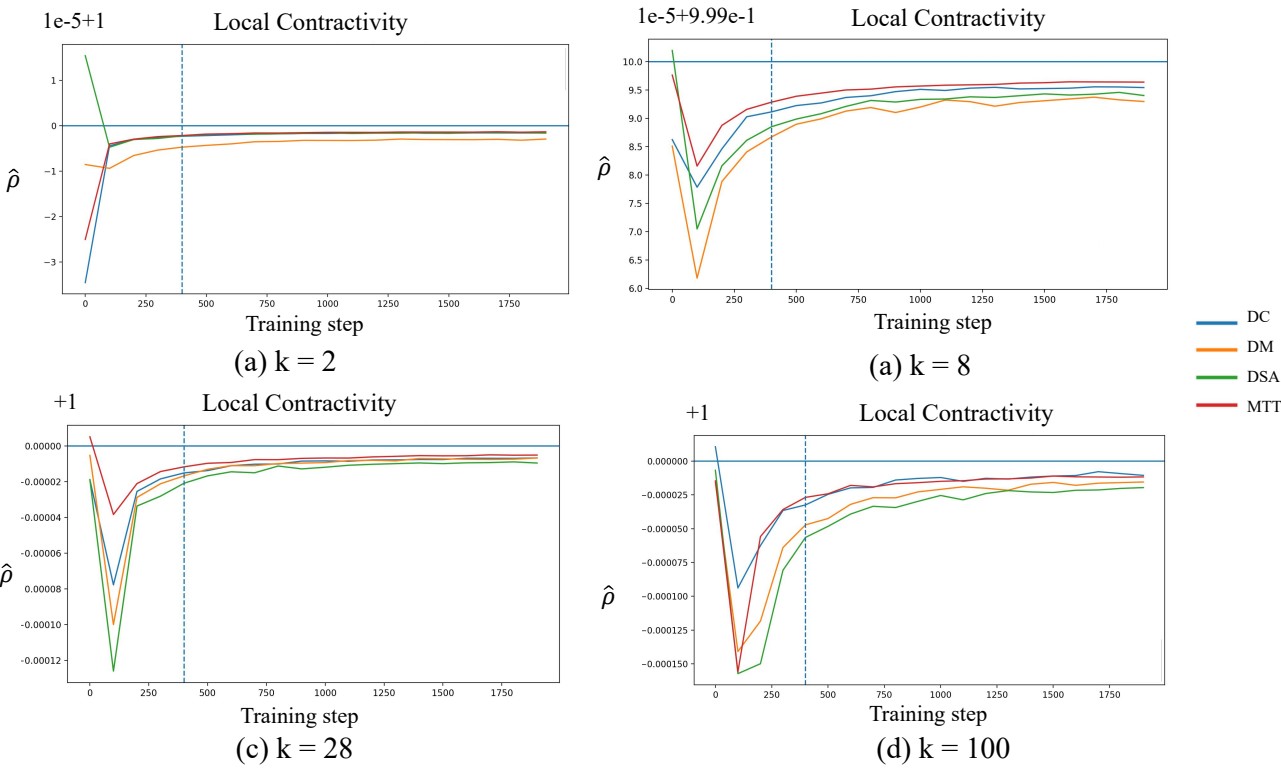

*Figure 17.* Trajectory-local contractivity probe along the optimization trajectory on CIFAR-10 (post-burn-in stability indicated by the dashed vertical line).

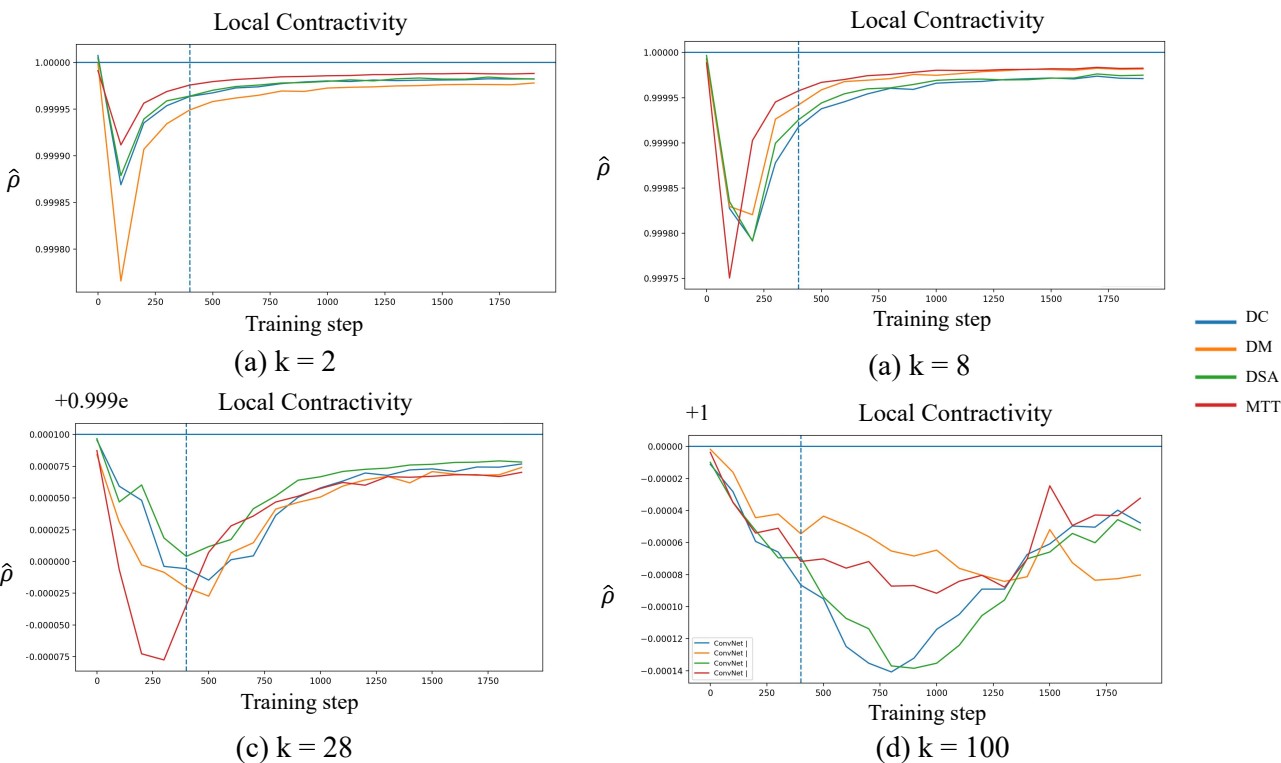

*Figure 18.* Trajectory-local contractivity probe along the optimization trajectory on CIFAR-100 (post-burn-in stability indicated by the dashed vertical line).

