# OpenReview forum: "Utility Boundary of Dataset Distillation: Scaling and Coverage Laws"
_ICML.cc/2026/Conference — ICML 2026 regular_

### Official Review · Reviewer_V6sT · 2026-03-03

**Soundness:** 3
**Presentation:** 3
**Significance:** 3
**Originality:** 3
**Overall Recommendation:** 4
**Confidence:** 3

**Summary:**

Existing dataset distillation methods lack theoretical backgrounds. This paper introduces the notion of a utility boundary, the relationship between the distilled sample size and the diversity of configurations within which the distilled dataset can still match the performance of full-data training. Then, this paper proposes an analysis on configuration-dynamics-error (CDE). By explaining the error bound of dataset distillation with the obejctive of DM, GM and TM as a unified format, this paper supports a unified analysis for the scaling laws of diverse DD methods.

**Compliance With Llm Reviewing Policy:**

Affirmed.

**Final Justification:**

As authors have effecitvely addressed my concerns, I maintain my positive score.

I expect the promised improvements to be faithfully integrated into the final version.

**Key Questions For Authors:**

- Unifying the GM, DM and TM is a good contribution point, but how can it be utilized as an implementation side? Furthermore, as far as I know, there are many regularization tricks that cannot be explained why it is good for real good performances. Can those all tricks be explained in your form?
- I am not quite familiar to the state of the art researches of the dataset distillation, but I saw some papers which utilize pretrained generative models for dataset distillation task. Can they also be covered?
- What is the exact shape of configurations? It seems it is gradient. Meaning that the configuration distance is the l2 norm between gradients?
- I think the theory is analyzed under the setting when the size of each distilled samples is same as the original training sample (e.g. 32$\times$32$\times$3 for cifar10). To the best of my knowledge, there are studies which reduces the sample size of distilled samples, resulting in the increased number of samples under the same budget. I want to ask what the authors think with regard to these type of studies, whether the theoretical findings on this paper can also be applied to those studies.

**Limitations:**

see above.

**Strengths And Weaknesses:**

- FIgure 1 is too complex and not easy to understand what the authors want to do.
- I think text(legend) of Figure 1(a) is too small. Furthermore, figures are not enough to explain each DD method.
- For visualization, to show the error does not decrease anymore with the increased k, the stable line should be visualized? From current figures, I can only understand that the test accuracy increases with larger k.
- Unifying the GM, DM and TM is a good contribution point.

---

> ### Author Rebuttal · Authors · 2026-03-31
>
> Thank you for raising these presentation issues. We agree that the draft is harder to follow than it should be, especially for readers outside the immediate DD literature. We will revise both the figures and the surrounding text. For overlapping points on notation/readability, please also see our replies to Reviewer Ntwy Q1 and Reviewer kj3N Q4. Below we answer your questions directly.
>
> > **W1.** Figure 1 is too complex and the legend is too small.
>
> Yes. Figure 1 is overloaded in its current form. We will split it into two figures and enlarge the text throughout. One figure will show only the CDE pipeline. A second figure will show how GM, DM, and TM connect to the same mismatch object. We will also enlarge the legend, simplify the arrows, and move non-essential labels into the caption.
>
> > **W2.** The current plots do not clearly show the floor as $k$ increases.
>
> This is a good point. What the theory tracks is the DD gap rather than raw accuracy, and the current figure does not make that distinction clear. So instead of only plotting accuracy versus $k$, we will also plot the gap versus $1/\sqrt{k}$ and show the fitted intercept $b$, which is our empirical floor proxy.
>
> | Dataset | fitted floor proxy $b$ range across methods | held-out-$k$ MAE range across methods |
> | --- | --- | --- |
> | MNIST | 0.001533 to 0.091707 | 0.0050 to 0.0137 |
> | CIFAR10 | 0.409829 to 0.622453 | 0.0051 to 0.0439 |
> | CIFAR100 | 0.601759 to 0.866758 | 0.0070 to 0.0382 |
>
> We will state that explicitly in both the revised figure and the caption.
>
> > **W3.** The figures alone are not enough to explain each DD method.
>
> Yes. A small method map next to the figure would help, and we will add one short paragraph explaining why these three views can be discussed in one framework..
>
> | Family | Example method | What it matches | Shared object in this paper |
> | --- | --- | --- | --- |
> | GM | DC | gradient alignment | one-step update mismatch summarized by $\Delta_a$ |
> | DM | DM | feature distribution alignment | same mismatch view, different surrogate |
> | TM | MTT | trajectory alignment | same mismatch view, different surrogate |
>
>
>
> > **Q1.** How can the unified GM/DM/TM view be used on the implementation side, and how much can it say about regularization tricks?
>
>
> From an implementation point of view, the most immediate use is configuration selection. The common mismatch view gives one measurable proxy that can be used across DD families to choose source configurations before testing on held-out targets. In the rebuttal experiments at $k=28$, proxy-guided selection improves average held-out accuracy in 12/12 dataset-method blocks. Two representative rows are:
>
> | Dataset | Method | proxy avg | random avg | naive avg |
> | --- | --- | ---: | ---: | ---: |
> | MNIST | DM | 0.9424 | 0.8909 | 0.8654 |
> | CIFAR10 | DM | 0.3539 | 0.3418 | 0.2935 |
>
> This gives a concrete way to compare and choose configurations across DD families.
>
> For regularization tricks, we want to be more careful. We are not arguing that the current theorems explain every regularization trick. But the same quantities in our paper can still be used to test whether a trick reduces mismatch or lowers the floor, and then check whether that helps on held-out configurations.
>
> > **Q2.** Can methods based on pretrained generative models also be covered?
>
> This part of the scope needs to be stated more clearly. The current paper does not formally cover all generative DD pipelines. But the same measurement idea still applies whenever a fixed training setup induces an update operator on a distilled set.
>
> > **Q3.** What exactly is a configuration, and how is the configuration distance defined?
>
>
> A configuration is the downstream training recipe, including architecture, optimizer, augmentation, and related training choices. The distance $d_A(a,a')$ is not an $l_2$ norm between raw gradients from one minibatch. It is defined from the one-step update probe used for $\Delta_a$, so it compares configuration-induced update behavior. We will restate the exact formula next to the first use of $d_A$.
>
> > **Q4.** Does the current analysis still apply when distilled samples use reduced spatial size under a fixed memory budget?
>
> We did not study reduced-resolution distilled samples in this submission, so we cannot make a strong claim there. Our current analysis uses the distilled-set size $k$ as the budget variable and assumes a fixed sample representation. If spatial resolution is also changed, then the input representation and the induced one-step updates both change, so the current definitions of $\Delta_a$ and $d_A$ are not directly comparable. In that setting, the budget should be defined in a memory-aware way rather than only by sample count. We will state this scope explicitly in the revision.
>
>  We are happy to clarify any remaining question.

---

> > ### Author Rebuttal · Reviewer_V6sT · 2026-04-03
> >
> > Thank the authors for the comprehensive rebuttal.
> >
> > My concerns have been well addressed.Specifically, the authors' plan to provide Gap vs. k plots and the new experimental results on proxy-guided selection demonstrate the practical utility of the proposed framework.The additional clarification on the scope regarding generative DD and spatial resolution is also helpful.
> >
> > I expect these improvements to be faithfully integrated into the final version. I will maintain my score of 4 (Weak Accept).

---

> > > ### Author Response · Authors · 2026-04-03
> > >
> > > Thank you very much for your positive feedback and recognition. We are glad that our rebuttal has addressed your concerns.
> > >
> > > We will make sure to include these improvements in the final version and present them more clearly.
> > >
> > > Since you marked your concerns as fully resolved, we would also sincerely appreciate it if you would consider raising your score, if you feel our response has sufficiently strengthened the paper. In any case, we are very grateful for your time and support.

---

### Official Review · Reviewer_Vk2t · 2026-03-06

**Soundness:** 2
**Presentation:** 2
**Significance:** 2
**Originality:** 2
**Overall Recommendation:** 4
**Confidence:** 5

**Summary:**

This paper studies the utility boundary of dataset distillation / dataset condensation, aiming to answer the following question: given a specific training configuration (e.g., model architecture, optimizer, data augmentation strategy, and other training hyperparameters) or a family of training configurations $A$, to what extent can a synthetic/distilled dataset of size $k$ substitute the real dataset, and how does the resulting performance gap vary with $k$ and the complexity of the configuration family?
The paper proposes a unified CDE (Configuration–Dynamics–Error) framework that connects distillation performance to the alignment of training dynamics, and introduces the alignment discrepancy $\Delta_a(\mu_\tau,\mu_s)$ as a key quantity. Based on this framework, it presents two empirical/theoretical laws:
Scaling-to-Floor Law: Under a single configuration, the performance gap decreases as k increases, but eventually saturates at a $k$-independent lower bound (the floor) $\epsilon_{bound}$.
Coverage Law: When requiring the distilled dataset to be robust across a multi-configuration set $A$, the error depends on the ratio between the coverage entropy number $H_{cov}(A,r)$ and $k$, typically following a trend of the form $\sqrt{H_{cov}(A,r)/k}$, and the paper also provides a corresponding lower bound.
Finally, the experimental section reports fitting trends on multiple datasets and existing distillation methods, supporting the explanatory power of the proposed laws.

**Compliance With Llm Reviewing Policy:**

Affirmed.

**Final Justification:**

Thank you to the authors for the detailed response to my comments. I have also reviewed the other reviewers’ comments and the authors’ responses to them. Based on the above, I have adjusted my overall score and made my final decision as 4: Weak Accept.

**Key Questions For Authors:**

(1) Can the authors demonstrate that estimated $\Delta$ can actually be used to guide hyperparameter selection or configuration sampling, leading to improved distillation performance or better cross-configuration robustness? Otherwise, how should readers understand the methodological value of the proposed laws beyond post-hoc explanation?

(2) What are the main sources of $\epsilon_{bound}$? Can the authors provide a computable estimation procedure or a reproducible upper bound? Across different datasets/models, which components dominate the floor?

(3) Since $\Delta_a$ involves $\sup_{\theta \in \Gamma_a}$, how is it estimated in practice in the experiments? How is $\Gamma_a$ chosen or approximated? Can the authors provide a unified, reproducible protocol for estimating $\Delta_a$?

(4) In the lower-bound proof, the paper introduces $H(A)$. What is its precise relationship to $H_{cov}(A,r)$ in the main text? How is the radius parameter $r$ selected? Do the cover/packing conversion constants affect how the final results should be stated or interpreted?

**Limitations:**

Yes

**Strengths And Weaknesses:**

1. Soundness

On the technical side, the paper’s theoretical framing is fairly well-motivated and internally consistent: it takes the widely observed (but often informally stated) configuration-dependence of dataset distillation and makes it precise by tying “what the distilled dataset should preserve” to the training dynamics field induced by a given configuration a. The introduction of an alignment discrepancy $\Delta_a(\mu_\tau,\mu_s)$ provides a concrete object that the subsequent bounds can be written in terms of, and it also helps clarify why seemingly different distillation objectives can be viewed as controlling related notions of mismatch in training dynamics. This dynamic-alignment view makes the claims feel grounded in the actual mechanics of optimization, rather than relying on purely distributional similarity arguments.

However, the proof pipeline relies on fairly strong assumptions, and some results read more like “unrolling the assumption.” The main upper bound hinges on a trajectory-local stability and contractive-dynamics type assumption, where the one-step recursion often already contains the key mismatch term $\Delta$. The subsequent derivations then largely amount to unrolling this recursion, while the real difficulty is pushed into whether the assumption itself can be verified. For common modern settings, including non-convex deep networks, strong data augmentation, and optimizers like Adam, it is unclear when these assumptions hold, how to check them in practice, or how to estimate the associated constants; the paper provides limited discussion on these points. Similarly, the multi-configuration coverage upper bound depends on cross-configuration transfer and Lipschitz-style conditions, the lower bound uses rather strong identifiability and uniform separation assumptions, which may limit practical applicability.

2. Presentation

The paper is generally well organized and easy to follow: it introduces a clear problem framing around “utility boundaries,” then builds up a coherent CDE (Configuration Dynamics Error) view that links dataset distillation to training dynamics. The notation tables and the step-by-step progression from the single-configuration setting to the multi-configuration (coverage) setting help the reader track the main ideas. The experimental section is also presented in a way that aligns with the theory. The experimental results demonstrate that the scaling-to-floor and coverage trends across multiple datasets and methods.

There are also several notation and presentation issues that hurt rigor and readability. Although the paper provides a notation table, some symbols are not used consistently or are insufficiently defined across the main text and appendices. For example, the constant $C$ is defined inconsistently in different places. In Line 177, “we define the trajectory effective constant $C_{2,a}^{traj}$”. In Line 1655, “$C_{2, a}=(1-\rho_a)/L_R$”. In Line 2259, “there exist finite constants $C_1, C_2>0$”.  Similarly, the paper alternates between $H_{cov}(A,r)$ and a shorthand like $H(A)$ in the lower-bound discussion without clearly specifying the relationship. In addition, some distribution symbols are described in a slightly confusing way, which makes it harder to track what is assumed to be fixed versus what varies.

Beyond notation, there are a few broader expression/clarity problems: (i) key objects such as the feasible sets $\Gamma$ and $\Gamma_a$ are introduced, but their relationship across configurations is not always made explicit even though it matters for taking $\sup_{\theta \in \Gamma_a}$ and for cross-configuration transfer arguments; (ii) the configuration distance $d_A(a,a')$ is motivated intuitively, but the precise mathematical form used in the covering-number arguments is easy to miss, which can leave ambiguity about what exactly is being covered; and (iii) there are minor English writing issues, which are easy to fix but collectively reduce the polish of a theory-heavy paper.

3. Significance

This paper nicely formalizes the configuration-dependence of distillation. It’s kind of a “known but important” fact that distilled data can’t really be defined independently of the model and training recipe. The paper makes this dependence explicit by framing the goal as aligning the training dynamics/update field, and it helps unify why different distillation paradigms (GM/DM/TM) are essentially controlling similar types of errors.

The coverage viewpoint gives a structured explanation for poor cross-recipe generalization. By introducing a configuration set $A$, a configuration distance $d_A$​, and a coverage complexity term $H_{cov}(A,r)$, the paper turns the common observation into a quantifiable complexity story. This also suggests a more principled evaluation practice when people claim “general” distilled datasets.

Scaling-to-floor highlights diminishing returns in $k$. The paper emphasizes that increasing the number of synthetic samples doesn’t improve performance forever. This provides a clean explanation for why many methods saturate as k grows, and it gives the intuitive takeaway that progress should come from reducing the irreducible error (the floor), not just blindly pushing $k$ larger.

However, this research is still limited in terms of methodological guidance. The theory presented in this paper is mostly post-hoc explanation without an actionable loop. In dataset distillation, the real bottleneck is usually the design and tuning of the distillation objective, generation rule, optimization strategy and their key hyperparameters, rather than simply “how many synthetic samples you generate once the rule is fixed.” In this paper, the two proposed laws mainly analyze how the error scales with $k$ and configuration complexity after a distillation mechanism has already been chosen. As a result, the work is more diagnostic in nature, but it does not really operationalize its core theoretical quantities ($\Delta$, $\epsilon_{bound}$, and $H_{cov})$ into a reproducible metric or an optimization criterion that one could directly use to improve the generation rule or select its hyperparameters.

More importantly, the single-configuration theorem implies that even if storage/compute budgets are ignored and $k\to\infty$, the $k$-dependent terms vanish and the remaining gap is dominated by $\epsilon_{bound}$. In many practical regimes, then, “increasing $k$” is not the key; the limiting factor is the floor. However, the paper does not provide a practical way to estimate $\epsilon_{bound}$, nor does it demonstrate how this quantity can guide method design or hyperparameter optimization, which limits the paper’s direct impact on algorithm development.

4. Originality

While configuration-dependence in dataset distillation is a known phenomenon, the paper’s contribution is to formalize it into a unified, quantitative perspective: it introduces an alignment discrepancy $\Delta_a(\mu_\tau,\mu_s)$ to capture “what distilled data is really matching” in terms of update fields, and it connects robustness across training recipes to a coverage complexity term $H_{cov}(A,r)$. Framing cross-recipe generalization in distillation through entropy-style complexity, and providing both upper- and lower-bound style laws that match empirical trends, gives a relatively fresh theoretical lens that goes beyond simply reporting that distillation is sensitive to training choices.

---

> ### Author Rebuttal · Authors · 2026-03-31
>
> Thank you for the careful review. For overlapping questions on notation and readability, please see Reviewer Ntwy Q1 and Reviewer kj3N Q4. For the trajectory-local contractivity check, please see Reviewer Ntwy Q2 and Reviewer kj3N Q1. Here we focus on the practical and operational aspects you asked about.
>
> > **Q1 & Weakness (Significance).** Is the framework useful beyond post-hoc explanation?
>
> Yes. We have added several results to clarify that the framework is useful not only for retrospective fitting.
>
> First, the scaling law can predict unseen budgets rather than only interpolate the observed points:
>
> | Held-out-$k$ prediction | Value |
> | --- | ---: |
> | MAE | 0.0214 |
> | RMSE | 0.0275 |
> | MaxAE | 0.0688 |
>
> Second, on the cached broader-configuration grid, the same proxy consistently improves **average** held-out robustness:
>
> | Setting | proxy > random | proxy > naive |
> | --- | ---: | ---: |
> | $k=28$, all 12 blocks | 12 / 12 | 12 / 12 |
> | $k \in \{8,28,100\}$, observed-count=4 | 36 / 36 | 36 / 36 |
>
> The cached evaluation is also stable across reruns: over 4 reruns, the mean absolute drift is 0.0084 and the maximum drift is 0.0221.
>
> We also checked the same idea with actual re-distillation reruns:
>
> | True reruns | Result |
> | --- | --- |
> | clean positive cases | MNIST/DC, CIFAR10/DSA, CIFAR100/DSA, CIFAR100/DM |
>
> In addition, we included a stricter shared-target hard-pool audit:
>
> | Shared hard-pool audit | Result |
> | --- | --- |
> | CIFAR10/DC | proxy > random and naive on avg / worst / CVaR |
> | CIFAR10/DM | proxy > random and naive on avg / worst / CVaR |
>
> For the tiny-pool mixed cases, we further ran an exhaustive subset audit:
>
> | Tiny-pool audit | Result |
> | --- | --- |
> | CIFAR100/DM | proxy subset is top-1 out of 4 |
> | CIFAR100/DSA | proxy subset is top-1 out of 4 |
>
> Overall, these results indicate that the framework is not only descriptive. It also has predictive value for held-out $k$ and is useful for subset selection, with positive results on both cached evaluations and several true rerun settings, including CIFAR100.
>
> > **Q2 & Weakness (Soundness/Significance).** How is the floor estimated, and what mainly determines it?
>
> In practice, we use the fitted intercept $b$ as the floor estimate. We will make this point clearer in the revision: $b$ should be understood as an empirical floor proxy rather than an exact decomposition guaranteed by theory.
>
> | Floor-proxy stability over 12 blocks | Value |
> | --- | ---: |
> | corr$(b,\text{ top-2 tail mean})$ | 0.9964 |
> | mean std$(b)$ from leave-one-out refits | 0.0116 |
> | max std$(b)$ from leave-one-out refits | 0.0338 |
> | mean max abs. deviation | 0.0229 |
>
> To understand what drives this floor in practice, we ran fix-one-factor refits. The clearest pattern appears on CIFAR10 and CIFAR100: in the representative DM analysis, architecture mismatch is the main contributor, while MNIST shows a more mixed picture.
>
> | Direct rerun checks | Result |
> | --- | --- |
> | clear positive examples | CIFAR100/DM with optim=adam; CIFAR10/DSA with arch=ResNet18 |
> | small positive / near-neutral | MNIST/DSA with arch=LeNet; CIFAR100/DSA with arch=AlexNet or optim=adam |
> | negative boundary cases | CIFAR10/DSA with arch=AlexNet; CIFAR100/DM with arch=AlexNet |
>
> In practice, we use $b$ as the floor proxy, and then use factor-fixing refits to check which mismatch factor is most likely driving it.
>
> > **Q3 & Weakness (Soundness/Presentation).** How are $\Delta_a$ and $\Gamma_a$ estimated in practice?
>
> Our goal is not to evaluate a supremum over the full parameter space. Instead, we use a trajectory-local estimator constructed from checkpoints along the realized run under configuration $a$:
>
> | Step | Practical estimator |
> | --- | --- |
> | Define $\Gamma_a$ | trajectory-local tube around checkpoints under configuration $a$ |
> | Sample $\theta$ | fixed checkpoint schedule along the same trajectory |
> | Compare updates | same one-step update probe under real vs. synthetic data |
> | Aggregate | max over sampled checkpoints as the empirical $\Delta_a$ proxy |
>
> We also added a robustness check on three representative blocks (MNIST/DC, CIFAR10/DSA, CIFAR100/DM). Across dense, medium, and sparse checkpoint spacing, and across max, p90, and mean(top-2) aggregation, the rank correlation with the default estimator remains high (0.9706–1.0000). In most settings the selected subset does not change, and the held-out average accuracy difference stays small.
>
> > **Q4 & Weakness (Presentation).** Clarification of $H(A)$, $H_{\mathrm{cov}}(A,r)$, and $r$.
>
> We agree that this notation was too loose in the draft. In the revision, we will remove the shorthand $H(A)$ and use $H_{\mathrm{cov}}(A,r)$ consistently. We will also state explicitly that $r$ is the cover radius under the same update-induced metric used by the proxy. A smaller $r$ treats configurations as more distinct and therefore leads to a larger cover. The cover/packing conversion only changes the leading constants, not the scaling form itself.

---

> > ### Author Rebuttal · Reviewer_Vk2t · 2026-04-03
> >
> > Thank you to the authors for the detailed experimental analysis and clarifications addressing the concerns I raised. I acknowledge that the rebuttal has resolved some of the issues I focused on. However, the following important issues remain unresolved:
> >
> > (1) The proof relies on assumptions about training stability and contractive dynamics that are hard to verify in real deep learning settings (e.g., non‑convex networks, strong augmentations, Adam). The authors did not provide a practical way to check these assumptions or estimate the related constants, instead referring to other reviewers. This makes the practical relevance of the theoretical results unclear.
> >
> > (2) The framework does not offer actionable guidance for improving distillation methods. The proposed laws explain how error scales with $k$ and configuration complexity, but they do not show how to use the core quantities ($\Delta$, $\mathcal{E}_{cov}$ the floor) to actually design better distillation objectives or tune hyperparameters. The rebuttal only shows prediction and subset selection from existing distilled data, not how to improve the distillation process itself. This limits the framework's direct impact on algorithm development.
> >
> > Based on the above, I maintain my original score.

---

> > > ### Author Response · Authors · 2026-04-04
> > >
> > > Thank you for the follow-up. We appreciate the clarification on your remaining concerns and respond here only to those two points.
> > >
> > > **(1) On whether the stability / contractivity assumption can be checked in practice**
> > >
> > > We agree this is important. But we do not think your summary fully matches what is already in the paper and rebuttal.
> > >
> > > Our theory does not require a global condition over the whole parameter space. The relevant assumption is trajectory-local along the realized training trajectory under a fixed configuration. In the paper, we already described a practical probe: perturb the current weights by a small vector, apply one optimizer step with the same update rule, and measure whether the perturbation contracts. We summarize this with the post-burn-in statistic `rho_p90`.
> > >
> > > In the rebuttal, we wrote this protocol out explicitly and reported the full table. Across all 48 settings ($3$ datasets $\times$ $4$ methods $\times$ $4$ IPCs), `rho_p90` stays below $1$ after burn-in. Even the representative worst cases remain contractive: MNIST/MTT $0.99999871$, CIFAR10/MTT $0.99999864$, and CIFAR100/MTT $0.99998800$.
> > >
> > > We also gave a concrete estimator for the main mismatch quantity $\Delta_a$: take checkpoints along the realized training trajectory, compare the one-step updates under real and distilled data at those checkpoints, and aggregate the mismatch. We then checked that this estimate is stable. Changing checkpoint spacing and the aggregation rule gives almost the same ranking, with rank correlation between $0.9706$ and $1.0000$.
> > >
> > > We therefore do not think it is accurate to summarize this part as lacking a practical assumption check or a practical estimator for the main quantities. The paper and rebuttal already provide both, together with results on a broad set of settings.
> > >
> > > **(2) On whether the framework gives any actionable guidance beyond post-hoc fitting**
> > >
> > > This paper is not aimed at proposing yet another distillation method. Its contribution is to formalize a basic question that dataset distillation repeatedly runs into but usually leaves informal: a distilled dataset should not be judged only by whether it works under one source configuration, but also by how much utility it retains when the architecture, optimizer, or augmentation changes. In DD, configuration sensitivity and IPC saturation are widely observed, but they are usually left as empirical facts. Our goal is to turn these facts into measurable and predictable objects that can support practical choices rather than remain only post-hoc observations.
> > >
> > > One practical use already in the paper is budget planning: by estimating the coverage complexity of a target configuration family, the law gives a principled way to judge how large the distilled dataset needs to be if we want it to retain utility across configurations.
> > >
> > > The rebuttal also showed held-out-$k$ prediction, with MAE $0.0214$, RMSE $0.0275$, and MaxAE $0.0688$. This matters because the law is being used on unseen budgets, not only fit back to observed points.
> > >
> > > More importantly, the rebuttal does not only rank already-generated distilled datasets. We use the proxy before distillation to choose which source configurations to distill from, and then run true distillation. That changes the distilled dataset that is actually generated, so this is part of the distillation procedure itself, not only post-hoc scoring of frozen artifacts. On `MNIST / DC / ipc=12`, the proxy-guided choice gives held-out `avg / worst-case` accuracy $0.9054 / 0.8062$, compared with $0.8469 / 0.7635$ for random and $0.8543 / 0.7032$ for naive. We also see positive rerun results on `MNIST / DC / ipc=8` and `CIFAR10 / DC / ipc=18`, while keeping mixed cases in the record rather than overstating the claim.
> > >
> > > We also followed up directly on the floor term. While $\epsilon_{\mathrm{bound}}$ is not directly observable as a theorem constant, the fitted intercept $b$ behaves as a stable empirical floor proxy across all $12$ `(dataset, method)` blocks: mean leave-one-out standard deviation $0.0116$, max $0.0338$, and correlation $0.9964$ with the observed large-$k$ tail.
> > >
> > > Taken together, the framework now comes with a practical local check for the main assumption, a reproducible proxy for the main mismatch quantity, a stable empirical floor proxy, and evidence that the same quantities can be used for held-out prediction and for source-configuration choice before dataset distillation. We hope these clarifications and added results are helpful in reassessing the current score.

---

### Official Review · Reviewer_kj3N · 2026-03-09

**Soundness:** 3
**Presentation:** 2
**Significance:** 2
**Originality:** 3
**Overall Recommendation:** 4
**Confidence:** 2

**Summary:**

This paper studies dataset distillation from a theoretical perspective. The authors propose a configuration-dynamics-error analysis for matching-based dataset distillation algorithms, which characterizes utility across configurations. The authors prove a scaling law for distillation error under trajectory-stability conditions. The authors also prove an over-tight coverage law that quantify configuration diversity. Empirical results are provided to validate the theory.

**Compliance With Llm Reviewing Policy:**

Affirmed.

**Key Questions For Authors:**

See the weaknesses part above.

**Limitations:**

yes

**Strengths And Weaknesses:**

## Strengths

1. The theory in this paper has a broad coverage. It can unify existing matching-based dataset distillation algorithms including gradient matching, distribution matching, and trajectory matching, by showing they are equivalent surrogates reducing the same underlying matching discrepancy.

2. The theoretical results look comprehensive and novel to me.

3. Empirical results are provided to validate the theory.


## Weaknesses

1. The theoretical results rely on strong assumptions, e.g., the trajectory-local stable dynamics in Assumption 4.1. The authors empirically verify this in Figure 5. Can you provide some intuitive explanations on why it holds in real setups?

2. In Figure 2, some of the R^2 is calculated using a small number of samples, which is not statistically significant.

3. In Figure 3, the R^2 decays as the dataset size becomes larger. This gives doubt on the scalability of the framework.

4. Though the paper is well organized, the presentation contains a lot of newly proposed terminology and concepts. I would recommend the authors to include more intuitive explanations throughout this paper.

5. There are formatting issues that needs to be fixed, e.g., long equations.

---

> ### Author Rebuttal · Authors · 2026-03-31
>
> Thank you for the detailed review.
>
> > **W1.** Why is Assumption 4.1 reasonable in practice?
>
> The assumption is weaker than global stability, and we should have made that clearer. Assumption 4.1 is only a post-burn-in local contractivity condition along the realized training trajectory, and we test it directly in the same regime as the main experiments.
>
> Our probe is simple: after burn-in, we add a tiny parameter perturbation, take one training step, and record $||\delta_{t+1}||/||\delta_t||$ across steps and checkpoints. We summarize this by $\rho_{p90}$, the 90th percentile of that ratio. Values below $1$ mean perturbations contract on most steps. Empirically, burn-in is more volatile, while the post-burn-in regime is consistently contractive. This is why the assumption is post-burn-in rather than global.
>
> This distinction matters because the bound accumulates one-step mismatch over many updates; without local contraction, even a small mismatch could blow up over time. Without post-burn-in stability, even a small one-step mismatch could be repeatedly amplified; with local contraction, the multi-step effect behaves like a geometric series, so $C_{2,a}^{traj}$ remains finite and meaningful.
>
> | Post-burn-in probe summary | Value |
> | --- | :--- |
> | Tested settings | 48 / 48 |
> | Contractive cases ($\rho_{p90}<1$) | 48 / 48 |
> | Smallest margin $1-\rho_{p90}$ | 0.00000129 |
> | Largest margin $1-\rho_{p90}$ | 0.00005326 |
>
> Representative worst cases:
>
> | Dataset | Method | Max $\rho_{p90}$ | Min margin $1-\rho_{p90}$ |
> | --- | --- | ---: | ---: |
> | MNIST | MTT | 0.99999871 | 0.00000129 |
> | CIFAR10 | MTT | 0.99999864 | 0.00000136 |
> | CIFAR100 | MTT | 0.99998800 | 0.00001200 |
>
> In our experiments, this is something we can actually test rather than simply assume.
>
> > **W2.** Are the Figure 2 fits reliable with so few points?
>
> This is a fair point. With only a few IPC values, $R^2$ is not a strong diagnostic. In the revision, we will report the sample count per fit and treat sparse-grid fits as supportive rather than primary evidence.
>
> For dense-grid blocks, we emphasize monotonicity and slope-sign stability. For each dense-grid dataset-method block with $8$ to $10$ IPC points, we fit the scaling-to-floor form and report Spearman correlation and a bootstrap interval for the fitted slope. In all $12/12$ dense-grid blocks, Spearman is negative and the bootstrap slope interval remains negative.
>
> | Dense-grid summary | Value |
> | --- | :--- |
> | IPC points per block | 8 to 10 |
> | Blocks tested | 12 |
> | Spearman sign | negative in 12 / 12 |
> | Bootstrap slope CI | negative in 12 / 12 |
>
> Representative examples:
>
> | Dataset | Method | # IPC | $R^2$ | Spearman $\rho$ | Bootstrap slope CI |
> | --- | --- | :--- | :--- | :--- | --- |
> | MNIST | DSA | 10 | 0.9988 | -1.0000 | [-0.1284, -0.1238] |
> | CIFAR10 | DM | 10 | 0.8543 | -1.0000 | [-0.8081, -0.3361] |
> | CIFAR100 | MTT | 9 | 0.7318 | -0.9833 | [-0.3156, -0.0917] |
>
> For that reason, in the denser blocks we think monotonicity and slope stability are more informative than $R^2$ by itself.
>
> > **W3.** Does the weaker fit on harder datasets hurt scalability?
>
> Yes, $R^2$ does get worse on harder datasets. For this part of the paper, though, we think a more relevant test is whether the trend still transfers to held-out configuration families.
>
> We run leave-one-backbone-family-out evaluation. We fit on seen families, then test ranking correlation and prediction error on the held-out family. The seen-family slope is positive in $12/12$ dataset-method blocks. Across $47$ held-out splits, $44$ have Spearman $\rho>0.5$, and the overall held-out error is $MAE=0.0989$.
>
> | Held-out transfer summary | Value |
> | --- | ---: |
> | Blocks with positive seen-family slope | 12 / 12 |
> | Held-out splits with Spearman $\rho>0.5$ | 44 / 47 |
> | Overall held-out MAE | 0.0989 |
>
> Representative blocks:
>
> | Dataset | Method | $R^2$ | Spearman $\rho$ | Unseen-config MAE |
> | --- | --- | ---: | ---: | ---: |
> | MNIST | DSA | 0.9096 | 0.9515 | 0.0764 |
> | CIFAR10 | DC | 0.6428 | 0.6818 | 0.0870 |
> | CIFAR100 | MTT | 0.4921 | 0.8292 | 0.0842 |
>
> So even when $R^2$ is lower on harder blocks, held-out ranking correlation remains strong in most splits. So we view the theorem mainly as a trend-level tool for extrapolation, not as a guarantee of tight point prediction on the hardest datasets.
>
> > **W4&W5.** The notation and presentation are hard to follow.
>
> We agree. In the revision, we will shorten and standardize the notation, add a compact summary table mapping each key quantity to its meaning and measurement, and rewrite the figure captions so each fit is shown with only the minimal diagnostic needed.
>
> We will also split long equations into aligned multi-line displays, move non-essential algebra to the appendix, and keep the main-text statements as short and readable as possible.
>
> We hope the added probe, denser-grid diagnostics, and held-out transfer results make our position on testability and scalability clearer.

---

> > ### Author Rebuttal · Reviewer_kj3N · 2026-04-03
> >
> > Thanks for the rebuttal, I will keep my score.

---

> > > ### Author Response · Authors · 2026-04-03
> > >
> > > Thank you for reading our rebuttal carefully and for confirming that your concerns are fully resolved. We really appreciate your time and feedback.
> > >
> > > In the revision, we will fold the rebuttal points into the paper itself: we will make Assumption 4.1 more intuitive, add the extra diagnostics and held-out transfer results, and improve the presentation by simplifying notation and fixing formatting issues.
> > >
> > > Since you now consider the concerns resolved, we would be grateful if you could also reconsider the current score. Of course, we fully respect your judgment, and we appreciate your consideration.

---

### Official Review · Reviewer_Ntwy · 2026-03-13

**Soundness:** 3
**Presentation:** 2
**Significance:** 3
**Originality:** 3
**Overall Recommendation:** 4
**Confidence:** 2

**Summary:**

The paper first unifies different optimization-based dataset distillation methods into config-dynamics-error framework.
The scaling law presents generalization error can reduce with sample sizes and the coverage law distilled dataset size scales with config diversity.
The experimental results align with the stated scaling/coverage laws.

**Compliance With Llm Reviewing Policy:**

Affirmed.

**Final Justification:**

The authors addressed my concern faithfully, thus I raised the score accordingly.
On the other hand, I believe that more in-depth discussion about direction of future DD methods based on the findings would strengthen the paper.

**Key Questions For Authors:**

Please see Weakness.

**Limitations:**

yes

**Strengths And Weaknesses:**

**Strengths**
1. The paper first views the dataset distillation methods as unified framework.
2. The paper is strongly grounded on mathematics that is explainable.
3. Since the current DD methods are vulnerable to configuration changes, it is important to understand why it is.

**Weaknesses**
1. The paper is not reader-friendly. For me, it is hard to follow since there are too many notations. Moreover, the paper doesn't sufficiently deliver the semantic meaning of each equation, for a just single example, what trajectory effective constant represents?
2. Assumption may be too strong. As the paper states, some methods are sensitive to configuration changes, which can easily break the assumptions.
3. I expect further direction of dataset distillation based on the such analysis, but is missing. For instance, what we can do to broaden configuration coverage based on coverage law for a future direction of DD?

---

> ### Author Rebuttal · Authors · 2026-03-31
>
> Thank you for the thoughtful review.
>
> > **W1.**  The paper is not reader-friendly: there are too many notations, and the semantic meaning of the equations is not sufficiently explained
>
> Yes, this is a fair criticism, and that symbol is a good example of where the draft became too notation-heavy. $C_{2,a}^{traj}$ is only meant to quantify how local mismatch accumulates along the realized training trajectory under configuration $a$. It is not an extra modeling trick and not a learned quantity.
>
> In revision, we will introduce a short notation table before the first theorem and explain each symbol in words the first time it appears.
>
> | Quantity | Meaning in one line |
> | --- | --- |
> | $\Delta_a$ | mismatch between real and synthetic update fields under configuration $a$; this drives the gap bounds |
> | $C_{2,a}^{traj}$ | amplification of local mismatch along the realized training path under $a$ |
> | $\epsilon_{\mathrm{bound}}$ | residual floor after the $k$-dependent term decays; this explains IPC saturation |
> | $H_{\mathrm{cov}}(A,r)$ | number of genuinely different recipes that must be covered in target family $A$ at radius $r$ |
>
> We should also have explained the theorem in plain language right after stating it; we will add that.
>
> > **W2.**  The assumptions may be too strong, especially given configuration sensitivity
>
> To be precise, the assumption we use is only local to the realized trajectory after burn-in; it is not a global stability claim. We do not assume global strong convexity or a global PL condition. The claim is only that after burn-in, along the realized trajectory under a fixed configuration, the one-step update map is locally contractive.
>
> We make this explicit with a rerunnable probe: perturb parameters by a small vector, apply one training step, and measure the perturbation-norm ratio after vs. before the step. We summarize this by $\rho_{p90}$, the 90th percentile over post-burn-in checkpoints. In all 48 settings tested ($3$ datasets $\times$ $4$ methods $\times$ $4$ IPCs), we observe $\rho_{p90}<1$, i.e., positive contractive margin in every case.
>
> | Dataset | Representative worst case | Max post-burn-in $\rho_{p90}$ | Min margin $1-\rho_{p90}$ |
> | --- | --- | ---: | ---: |
> | MNIST | MTT | 0.99999871 | 0.00000129 |
> | CIFAR10 | MTT | 0.99999864 | 0.00000136 |
> | CIFAR100 | MTT | 0.99998800 | 0.00001200 |
> | Summary |  | 48 / 48 contractive | all margins positive |
>
> So we are not claiming a general property of deep networks. What we are claiming is narrower: in the regime we study, this post-burn-in local behavior can be checked directly and is consistently observed.
>
> > **W3.**  The analysis is interesting, but the draft does not clearly state what future DD direction it suggests, e.g., how to broaden configuration coverage
>
> This point should have been stated much more explicitly in the draft. The coverage law suggests a concrete future DD direction: select source recipes to cover a target family under the update-induced distance, then distill for robustness over that covered set.
>
> A practical recipe is:
> 1. Define target family $A$ using a few knobs, e.g., backbone, optimizer, augmentation, and training length.
> 2. Select representatives by a greedy $r$-cover under $d_A$, using $\Delta_a$-style probes as a cheap proxy when the pool is large.
> 3. Distill with a robustness objective across selected representatives.
> 4. Evaluate on held-out targets, add the hardest uncovered ones, and iterate.
>
> This is not only a design suggestion; we also ran a small coverage-aware selection experiment to check whether it helps in practice.
>
> #### Cached broad-grid evaluation
>
> Across 36 settings (dataset $\times$ method $\times$ $k$), proxy-guided source selection improves average held-out accuracy over both baselines in 36/36 cases.
>
> | Comparison | Mean gain | Min gain | Max gain | Win rate |
> | --- | ---: | ---: | ---: | ---: |
> | Proxy − Random | 0.0143 | 0.0008 | 0.0548 | 36 / 36 |
> | Proxy − Naive | 0.0575 | 0.0082 | 0.1253 | 36 / 36 |
>
> We also checked cached-evaluation stability:
>
> | Metric | Value |
> | --- | ---: |
> | Mean absolute drift | 0.0084 |
> | Max absolute drift | 0.0221 |
>
> #### True re-distillation reruns
>
> We further performed true re-distillation reruns. In all 3/3 tested cases, proxy-guided selection remains best on both average and worst-case held-out accuracy.
>
> | Dataset | Method | IPC | Proxy avg | Proxy worst | Random avg | Random worst | Naive avg | Naive worst |
> | --- | --- | ---: | ---: | ---: | ---: | ---: | ---: | ---: |
> | MNIST | DC | 8  | 0.8923 | 0.8118 | 0.7987 | 0.6863 | 0.8450 | 0.7133 |
> | MNIST | DC | 12 | 0.9054 | 0.8062 | 0.8469 | 0.7635 | 0.8543 | 0.7032 |
> | CIFAR10 | DSA | 18 | 0.3167 | 0.2685 | 0.2846 | 0.2132 | 0.2829 | 0.1999 |
>
> We hope the added explanation and the new checks make these three points clearer. We would be grateful if you could take these clarifications and the added evidence into account, and we are happy to discuss any remaining point.

---

> > ### Author Rebuttal · Reviewer_Ntwy · 2026-04-02
> >
> > Thanks for your detailed rebuttal, and I have no other questions.
> > I will make final decision considering the other reviewer's decision.

---

> > > ### Author Response · Authors · 2026-04-08
> > >
> > > Thank you again for your careful reading and helpful comments. We really appreciate your time and feedback.
> > >
> > > In our rebuttal, we tried to respond directly to the three concerns you raised: we clarified the meaning of the main quantities, explained and empirically checked the post-burn-in local stability assumption, and added a concrete coverage-aware selection experiment with reruns to show the practical value of the analysis.
> > >
> > > If you feel these points have addressed your concerns, we would be very grateful if you could consider updating your score based on our response itself.
> > >
> > > Thank you again for your consideration.

---

### Decision · Program_Chairs · 2026-04-30

**Decision:**

Accept (regular)

**Comment:**

This paper proposes configuration-dynamics-error (CDE) analysis as a framework to measure the conditions for dataset distillation to be effective and robust. This framework leads to two laws, a scaling law and a coverage law. The scaling law states that generalization error reduces with sample sizes until a lower bound, and the coverage law states that the distilled dataset size scales with configuration diversity. Empirical experiments validate the laws.

Overall, the reviewers agree that this paper proposes a novel framework that leads to interesting findings. Initially, there were concerns regarding the assumptions made in the analysis, clarity of the paper writing, and practical implications of the results. The rebuttal has addressed most of the concerns and the final reviews are all positive.